# Widespread retreat of coastal habitat is likely at warming levels above 1.5 °C

Neil Saintilan[1,2 ✉], Benjamin Horton[3,4], Torbjörn E. Törnqvist[5], Erica L. Ashe[6], Nicole S. Khan[7], Mark Schuerch[8], Chris Perry[9], Robert E. Kopp[6], Gregory G. Garner[6], Nicholas Murray[10], Kerrylee Rogers[11], Simon Albert[12], Jeffrey Kelleway[11], Timothy A. Shaw[3], Colin D. Woodroffe[11], Catherine E. Lovelock[13], Madeline M. Goddard[14], Lindsay B. Hutley[14], Katya Kovalenko[15], Laura Feher[16] & Glenn Guntenspergen[17]

Several coastal ecosystems—most notably mangroves and tidal marshes—exhibit biogenic feedbacks that are facilitating adjustment to relative sea-level rise (RSLR), including the sequestration of carbon and the trapping of mineral sediment[1]. The stability of reef-top habitats under RSLR is similarly linked to reef-derived sediment accumulation and the vertical accretion of protective coral reefs[2]. The persistence of these ecosystems under high rates of RSLR is contested[3]. Here we show that the probability of vertical adjustment to RSLR inferred from palaeo-stratigraphic observations aligns with contemporary in situ survey measurements. A deficit between tidal marsh and mangrove adjustment and RSLR is likely at 4 mm yr⁻¹ and highly likely at 7 mm yr⁻¹ of RSLR. As rates of RSLR exceed 7 mm yr⁻¹, the probability that reef islands destabilize through increased shoreline erosion and wave over-topping increases. Increased global warming from 1.5 °C to 2.0 °C would double the area of mapped tidal marsh exposed to 4 mm yr⁻¹ of RSLR by between 2080 and 2100. With 3 °C of warming, nearly all the world's mangrove forests and coral reef islands and almost 40% of mapped tidal marshes are estimated to be exposed to RSLR of at least 7 mm yr⁻¹. Meeting the Paris agreement targets would minimize disruption to coastal ecosystems.

Coastal ecosystems have long been recognized as indispensable to the well-being and subsistence of millions of people[4]. Marine vegetation and fringing reefs attenuate wave energy, protecting coastlines while providing habitat to distinctive assemblages of species. Coral reefs are productive ecosystems of high ecological value, and reef islands—consisting of biogenic carbonate sands—are frequently inhabited by communities dependent on these resources[2]. Vegetated coastal ecosystems (mangroves, tidal marshes, seagrass meadows and kelp forests) are foundational to coastal fisheries[5] and are, in addition, well placed to contribute to CO₂ removal in efforts to maintain warming below 2 °C. Global guidelines[6] and a growing understanding of conservation and restoration opportunities are enabling an increasing number of coastal nations to account for carbon that is captured and stored in coastal and marine ecosystems ('blue carbon') while enhancing the coastal protection afforded by living shorelines[7]. Several jurisdictions have enacted or are progressing incentives for tidal wetland protection, restoration and/or creation[8] to reverse centuries of decline, slowing the rate of mangrove loss[9] and demonstrating the potential for rebuilding coastal ecosystems damaged by the human population footprint in the coastal zone[10].

These important ecosystems face an uncertain future as a result of human-induced climate change. Many of the most important coastal ecosystems show biogenic responses to RSLR that enhance their physical resilience[11]. The potential for high rates of sedimentation, productivity and organic matter preservation in mangroves and tidal marshes and the productivity of coral reefs have enabled them to grow vertically with RSLR over millennia[12]. We refer to this process as 'vertical adjustment'. Vertical adjustment can maintain a wetland above a drowning threshold, a buffer referred to as 'elevation capital'[13]. For reef island systems, vertical adjustment maintains the uppermost portions of a reef near mean sea level. Where the rate of vertical adjustment falls behind the rate of RSLR, an elevation deficit emerges, and the surface is exposed to increasing depth and duration of inundation. This change in inundation may enhance vertical adjustment[3], but if a

[1]School of Natural Sciences, Macquarie University, Sydney, New South Wales, Australia. [2]Institute of Plant Science and Microbiology, Universität Hamburg, Hamburg, Germany. [3]Earth Observatory of Singapore, Nanyang Technological University, Singapore, Singapore. [4]Asian School of the Environment, Nanyang Technological University, Singapore, Singapore. [5]Department of Earth and Environmental Sciences, Tulane University, New Orleans, LA, USA. [6]Department of Earth and Planetary Sciences and Rutgers Institute of Earth, Ocean, and Atmospheric Sciences, Rutgers University, Piscataway, NJ, USA. [7]Department of Earth Sciences, Swire Institute of Marine Science and Institute of Climate and Carbon Neutrality, University of Hong Kong, Hong Kong, Hong Kong. [8]Catchments and Coasts Research Group, Department of Geography, University of Lincoln, Lincoln, UK. [9]Geography, Faculty of Environment, Science & Economy, University of Exeter, Exeter, UK. [10]College of Science and Engineering, James Cook University, Townsville, Queensland, Australia. [11]School of Earth Atmospheric and Life Sciences and GeoQuEST Research Centre, University of Wollongong, Wollongong, New South Wales, Australia. [12]School of Civil Engineering, The University of Queensland, Brisbane, Queensland, Australia. [13]School of Biological Sciences, The University of Queensland, Brisbane, Queensland, Australia. [14]Research Institute of Environment and Livelihoods, Faculty of Science and Technology, Charles Darwin University, Darwin, Northern Territory, Australia. [15]Natural Resources Research Institute, University of Minnesota–Duluth, Duluth, MN, USA. [16]US Geological Survey, Wetland and Aquatic Research Centre, Lafayette, LA, USA. [17]US Geological Survey, Eastern Ecological Research Center, Beltsfield, MD, USA. ✉e-mail: neil.saintilan@mq.edu.au

deficit is sustained for a sufficiently long period, elevation capital is exhausted. For wetlands, retreat and a transition to open water may occur, and in reef islands, submergence of reef crests will increase wave exposure and wave over-topping frequency. Whether the areal extent of the habitat expands or contracts over time depends on the rate of loss and the rate of new habitat formation, both both of which are influenced by RSLR[14].

Contemporary observations of high accretion rates in coastal ecosystems have indicated resilience under current and projected RSLR rates, prompting reassessment of their vulnerability in modelling studies[3,11,14]. Conversely, studies emerging from the palaeo record show a comparatively high vulnerability of mangroves[12] and tidal marshes[15,16] to rates of RSLR that are anticipated in coming decades under moderate and high emissions scenarios[17]. Palaeo records show that most coral reef islands formed during the later stages of the Holocene epoch under conditions of stable or falling relative sea level[2] (RSL). The upper limits of resilience to projected RSLR remains an important knowledge gap, with wide ranging implications for coastal zone protection and management.

Here we analyse three independent lines of evidence to assess the vulnerability and exposure of coastal ecosystems to the higher rates of sea-level rise (4 mm yr$^{-1}$ to more than 10 mm yr$^{-1}$) projected under global warming scenarios. We focus on intertidal and supratidal ecosystems that undergo vertical adjustment from biogenic feedbacks, facilitating resilience to RSLR: mangroves, tidal marshes and coral reef island systems. We exclude beaches, rocky reefs and rock platforms, for which biogenic feedbacks with RSLR are largely absent, and subtidal vegetated ecosystems (seagrass meadows and kelp forests) for which thermal stress is likely to be the primary driver of change, rather than RSLR[18,19]. First, we review the behaviour of these ecosystems over the range of sea-level histories encountered following the Last Glacial Maximum 19 thousand years ago (ka), and particularly since 10 ka. Second, for mangroves and tidal marshes, we document elevation trends in relation to contemporary rates of RSLR using a global network of survey benchmarks, the surface elevation table-marker horizon (SET-MH) network. Third, we analyse the extent to which contemporary coastal ecosystems show conversion to open water (hereafter referred to as 'retreat') under a range of settings with varying rates of RSLR. From these three lines of evidence, we estimate the probability of coastal ecosystem retreat in relation to RSLR rates and model the response of the world's existing coastal ecosystems under the RSLR projections of the Intergovernmental Panel on Climate Change (IPCC) Sixth Assessment Report[17], including potential compensation through conversion of terrestrial uplands (hereafter 'landward migration'). From these analyses a picture emerges of the narrowing boundaries of the 'safe operating space'[20] for coastal ecosystems: the climate futures expected to be of low risk to existing ecosystems.

## Responses to past sea-level rise

RSL varies globally in response to both water and land vertical movement[21]. Coastlines continue to adjust to the loss of ice sheets after the previous glacial period, particularly in higher latitudes, a process called glacial isostatic adjustment[21,22] (GIA). GIA modelling provides insights into sea-level trends since the last deglaciation. These observations and models have been applied to interpret rates of RSLR associated with the timing of mangrove and tidal marsh retreat and/or advance in stratigraphic successions, with results showing broad consistency among settings[12,15].

Rapid global mean sea-level (GMSL) rise (over 10 mm yr$^{-1}$) during several periods since the Last Glacial Maximum has drowned mangrove forests and tidal marshes (Fig. 1). Periods of rapid GMSL rise include: (1) meltwater pulse 1A (14.6–14.3 ka) which drowned mangroves, tidal marshes and coral reefs, the remains of which have been found at water depths of around 90 m; and (2) a rapid rise in GMSL 11.3–11.0 ka, leaving relict features at water depths of around 50 m. Interspersed with these phases have been periods of slower GMSL rise, allowing extended periods of coastal ecosystem expansion (Fig. 1; Methods). For example, mangrove sediments associated with preserved coastal palaeo channels on the Sahul Shelf (northwest of mainland Australia) and the Sunda Shelf (western South China Sea) dating to 16.0–14.5 ka were probably drowned during meltwater pulse 1A (Methods). Mangrove forests re-established in India from around 10 ka on the former delta of the Ganges–Brahmaputra River (Fig. 1) and the northeast Australian continental shelf (Queensland). In both locations, RSLR declined to between 6 and 7 mm yr$^{-1}$, and rates of sedimentation were high. These forests were subsequently drowned during a period of more rapid RSLR of 7 to 8 mm yr$^{-1}$ between 9.0 and 8.5 ka (Methods). Widespread mangrove forest development commenced around 8.5–7.5 ka in Southeast Asia, northern and eastern Australia, South America and Africa (Fig. 1) as the rate of RSLR declined[12] below 7 mm yr$^{-1}$ (Fig. 2c), and mangroves associated with large rivers were able to maintain their intertidal position by trapping sediment and accumulating root mass.

Tidal marshes in Great Britain were 9 times more likely to retreat than advance during the Holocene when RSLR exceeded 7.1 mm yr$^{-1}$, based on more than 780 reconstructions of tidal marsh evolution[15] (Fig. 2b). In the Mississippi Delta, only short-lived and rapidly retreating fringing tidal marshes existed before 8.2 ka, with retreat occurring in approximately 50 years before RSLR slowed[16] to less than 6 to 9 mm yr$^{-1}$. Tidal marsh retreat took longer (centuries), as RSLR dropped below 6 mm yr$^{-1}$ after around 8.2 ka, but marshes did not stop retreating in the Mississippi Delta[16] until RSLR was less than 3 mm yr$^{-1}$ (Fig. 2b).

Sea level stabilized in the mid-Holocene in those parts of the world that were distant from former centres of glaciation, and many coral reefs—especially in the Pacific—reached sea level with diversification of reef habitats[23]. Subsequent fall of sea level relative to these regions resulted in emergent reef platforms, some of which became suitable habitat for mangroves, and on which it became possible for reef islands to form[24]. Infilling of estuaries resulted in development of extensive coastal plains[12], reducing intertidal areas previously covered by mangroves, including in coastal Northern Australia (South Alligator River, Fitzroy River, Ord River, Cleveland Bay and Richmond River), Thailand (Great Songkhla Lakes) and Vietnam (Mekong River and Red River).

The palaeo record therefore indicates a capacity for vertical adjustment to rates of RSLR similar to those encountered in the instrumental period. If these rates of RSLR are sustained, coastal lowlands may be re-occupied by tidal wetlands where migration is permitted, and in many places this encroachment has already commenced[25]. The potential for increased extent in these regions under higher sea level is captured in global wetland adjustment models[14,26]. However, there is consistent evidence that vertical adjustment and habitat extent are greatly reduced[12,23] as RSLR approaches 7 to 8 mm yr$^{-1}$.

## Elevation trends under current sea-level rise

Mangrove and tidal marsh accretion can increase with the rate of RSLR. Increased inundation depth and duration can facilitate both mineral deposition[27] and higher plant productivity and root mass accumulation[11]. Rates of accretion measured against artificial marker horizons and radiometric markers often correspond to high rates of RSLR encountered in settings where land is subsiding[3,28,29]. High rates of accretion (10 to 20 mm yr$^{-1}$) have been observed in contemporary mangroves and tidal marshes on active deltas[29,30], and accretion in the intertidal zone increases with increased depth and duration of inundation[3]. The assumption that accretion enables vertical adjustment to RSLR is the basis of projections of possible resilience under projected future rates[31], but the assumption requires testing against fixed elevation benchmarks.

To assess the relationship between accretion of surface sediment, vertical adjustment and sea-level rise in contemporary coastal wetlands, we used the SET-MH method (Fig. 2d; Methods). The SET-MH

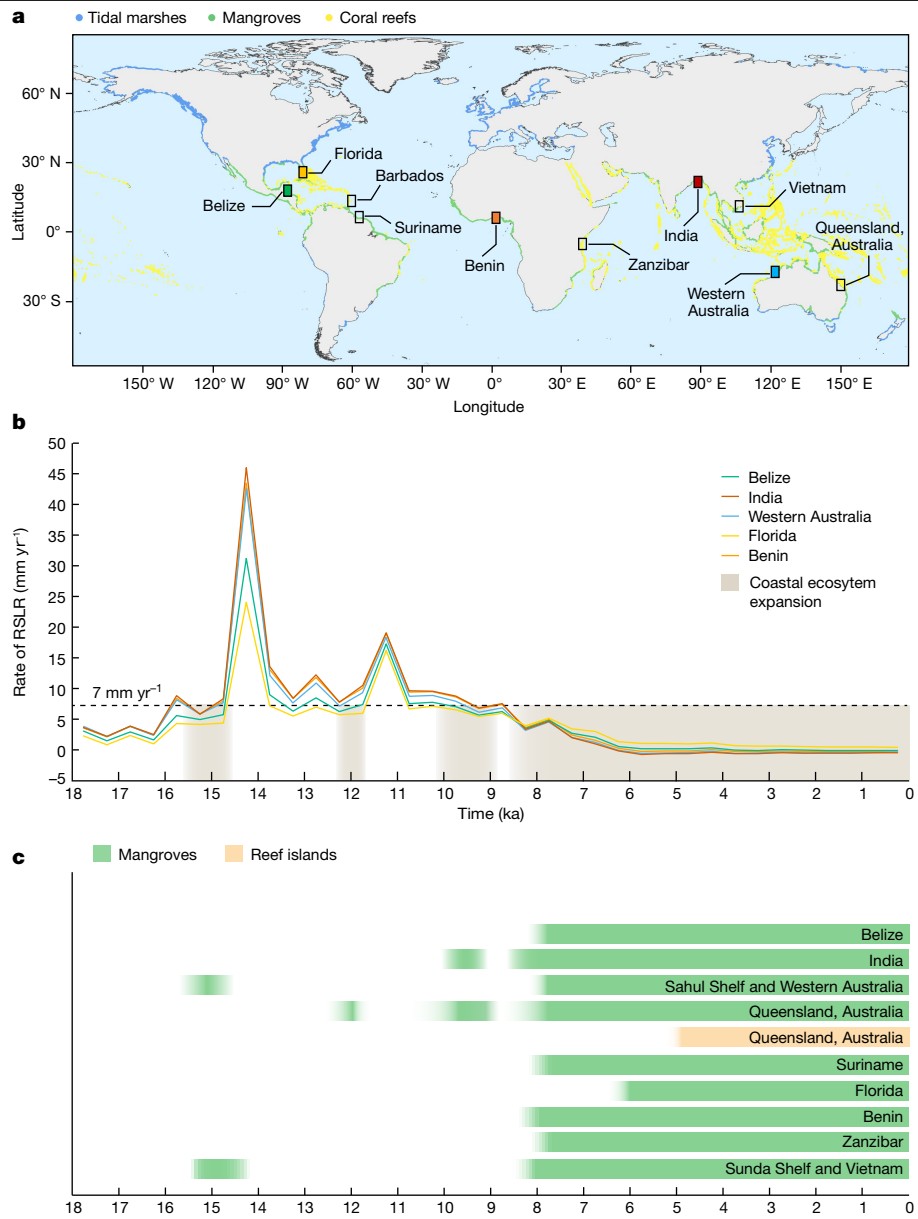

**Fig. 1 | Coastal ecosystem responses to RSLR following the Last Glacial Maximum. a**, Present-day distribution of mapped coastal ecosystems and the location of case studies highlighted in **b**,**c**. **b**, Median rates of RSLR over time derived from GIA modelling (Methods). **c**, Timing of habitat advance and retreat for a selection of locations (Methods). Mangrove, tidal marsh and coral reef island development is predominantly associated with periods of RSLR of less than 7 mm yr[-1].

uses a benchmark survey rod coupled with an introduced sediment horizon to assess the relationship between accretion and elevation gain[32]. SET-MH data in tidal marshes show that shallow subsidence (the difference between sediment accretion and elevation gain) increases with accretion rate and the rate of RSLR[28,29]. A previous analysis of a globally distributed network of 477 tidal marsh SET-MH stations showed that the increase in the subsidence rate with increasing accretion was non-linear[29]. For this reason, elevation deficits emerged under rates of RSLR similar to those inferred from the stratigraphic record[29] (Fig. 2e and Extended Data Fig. 2). We repeated this Bayesian analysis for 190 SET-MH installations in mangrove forests (Methods), estimating the cumulative probability of vertical adjustment at or exceeding the rate of RSLR at the SET-MH stations (Fig. 2f and Extended Data Fig. 2). The results were consistent with those for tidal marshes. We found that an elevation deficit at mangrove sites is very likely ($P > 0.9$) at RSLR between 7 and 8 mm yr[-1] (Fig. 2f), consistent with tidal marshes

monitored using the same method (Fig. 2e). These observations concur with the limits of tidal marsh and mangrove stability in relation to palaeo-RSLR as inferred from the stratigraphic record[29] and described previously (Fig. 2a–c).

## Habitat change under current sea-level rise

As a third line of evidence, we assessed whether changes in the extent of tidal marsh and open water were consistent with RSLR and/or the deficit between RSLR and marsh vertical adjustment (Extended Data Fig. 3; Methods). Previous surveys of contemporary North American tidal marshes in low-to-moderate tidal range settings[33] found that habitat retreat commenced at a RSLR of 4 to 6 mm yr[-1]. For example, the Maryland Eastern Shore is retreating[34] under a long-term RSLR trend of around 6 mm yr[-1]. In a comprehensive analysis of tidal marshes in the contiguous USA, gains in tidal marsh were found to be inversely related

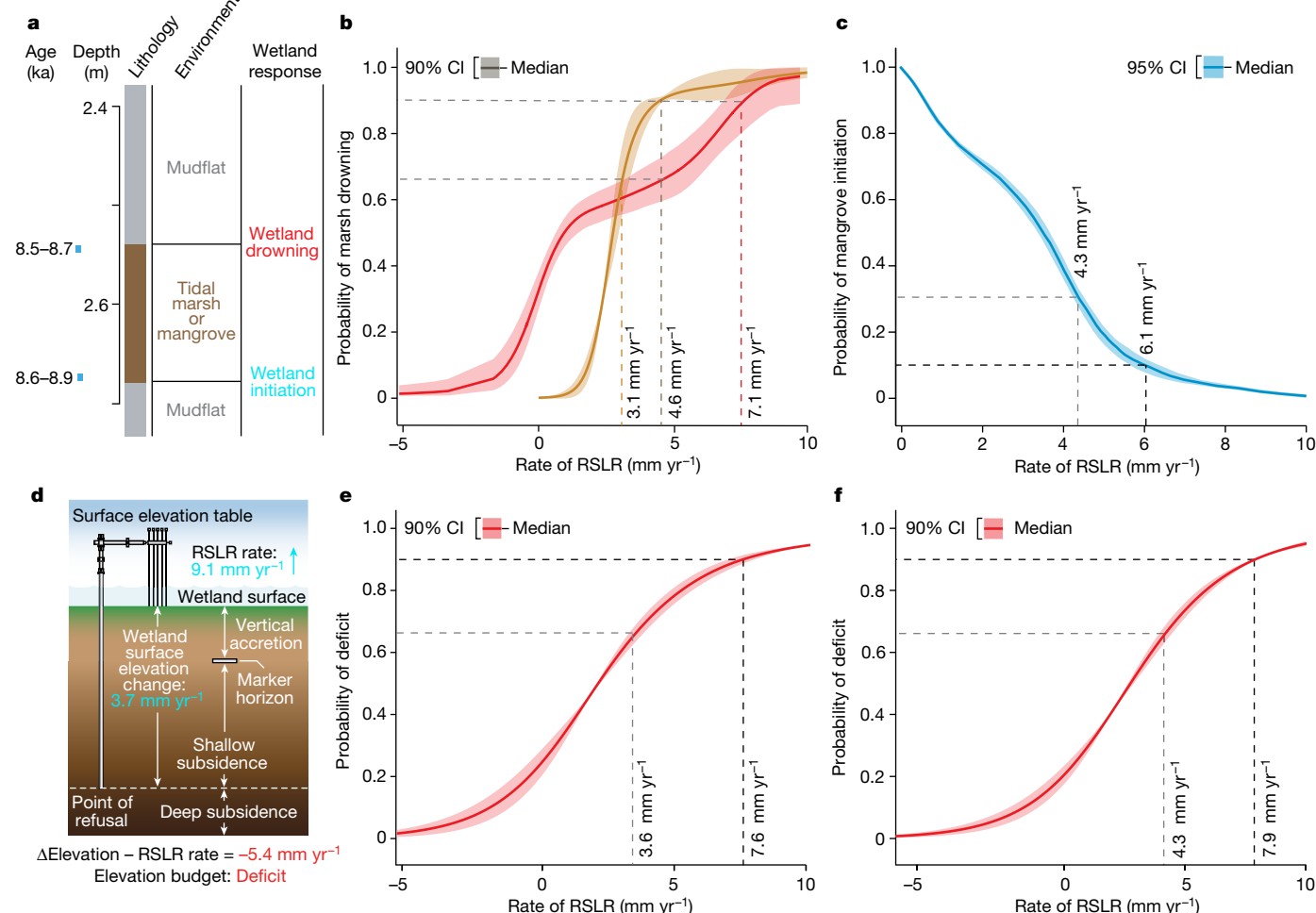

**Fig. 2 | Probability of vertical adjustment of mangrove and tidal marsh to rising sea levels. a–c**, Palaeo-stratigraphic assessments (**a**) of marsh adjustment or retreat for tidal marshes (**b**) and the probability of the initiation of sustained mangrove vertical adjustment[12] (**c**), in relation to rates of RSLR encountered over the past 10,000 years. **b**, The red line represents minerogenic UK marshes[15] and the orange line represents organic marshes of the Mississippi Delta[14]. **d–f**, Results in **a–c** are compared with vertical adjustment as assessed by the surface elevation table (**d**), analysed for the probability of a deficit between vertical adjustment and RSLR for the same period of measurement, for 477 tidal marsh SET-MH installations[29] (**e**) and 190 mangrove SET-MH installations (**f**). Adopting IPCC likelihood definitions (Methods), we indicate in each case the probability thresholds at which mangrove or marsh drowning becomes likely ($P \geq 0.66$) or very likely ($P \geq 0.90$). The corresponding histograms for each RSLR increment are shown in Extended Data Fig. 2. **b,c,e,f**, The line represents the median and the shaded region shows 90% confidence interval (CI).

to RSLR, with some marsh loss associated with short-term perturbations, notably hurricanes[34]. RSLR was also associated with reduced normalized difference vegetation index (NDVI) values for vegetation adjacent to the marsh[34], possibly resulting from saline water intrusion.

We used high-resolution global mapping of surface water change[35] and tidal wetland extent[36] in the immediate vicinity of tidal marsh SET-MH stations globally to determine the influence of contemporary RSLR, elevation capital and elevation deficit on marsh loss. Canopy cover obscured observations of surface water in mangroves. We found that tidal marsh sites were likely to show a trend towards increased presence of surface water ($P > 0.66$) once RSLR exceeded 2.3 mm yr$^{-1}$ (Extended Data Fig. 3). The frequency of surface water observations at marsh sites increased with both the rate of RSLR ($r^2 = 0.16$, $P < 0.001$; Extended Data Fig. 4) and marsh elevation deficit ($r^2 = 0.14$, $P < 0.001$; Extended Data Figs. 4 and 5). The relationship between surface water change and marsh elevation deficit was evident in lower elevation marsh sites ($r^2 = 0.20$) rather than higher elevation sites ($r^2 = 0.03$; Extended Data Fig. 5), illustrating the temporary resilience conferred by elevation capital. We also found a significant relationship between the proportion of tidal marsh conversion to open water habitat and RSLR ($P = 0.018$). Tidal marshes were as likely as not ($P = 0.5$) to be retreating as RSLR

increased above 5.4 mm yr$^{-1}$ (Extended Data Fig. 3), with relatively few marshes advancing. This estimate of retreat may be conservative because patches of interior marsh break-up may not have been identified (Methods). The ameliorating influence of elevation capital was also evident in the extent of marsh retreat. Where marshes had higher than the median elevation capital, there was no relationship between marsh retreat and RSLR ($P = 0.850$). At lower than the median elevation capital, the relationship was highly significant ($P = 0.002$).

There are relatively few data on the change to reef-top habitats. Surveys of reef island planiform change in the tropical western Pacific and Indian Oceans have shown a remarkable degree of stability under rates of RSLR up to the contemporary GMSL rate[37,38]. Our collation of existing data on reef island morphometric changes ($n = 872$) from the Indian and Pacific Oceans shows a higher probability of island contraction at rates of RSLR above the rate of contemporary GMSL rise (Methods; Extended Data Fig. 3c). Island size reduction is likely ($P \geq 0.66$) at RSLR above 6.2 mm yr$^{-1}$. The rate of RSLR in the Solomon Islands has averaged between 7 and 10 mm yr$^{-1}$ since 1994 (ref. 39), and in the exposed northern Isabel Province, five of the twenty vegetated reef islands have completely eroded, leaving dead mangrove trunks on hard coral[40]. A further six islands contracted by more than 20% in the

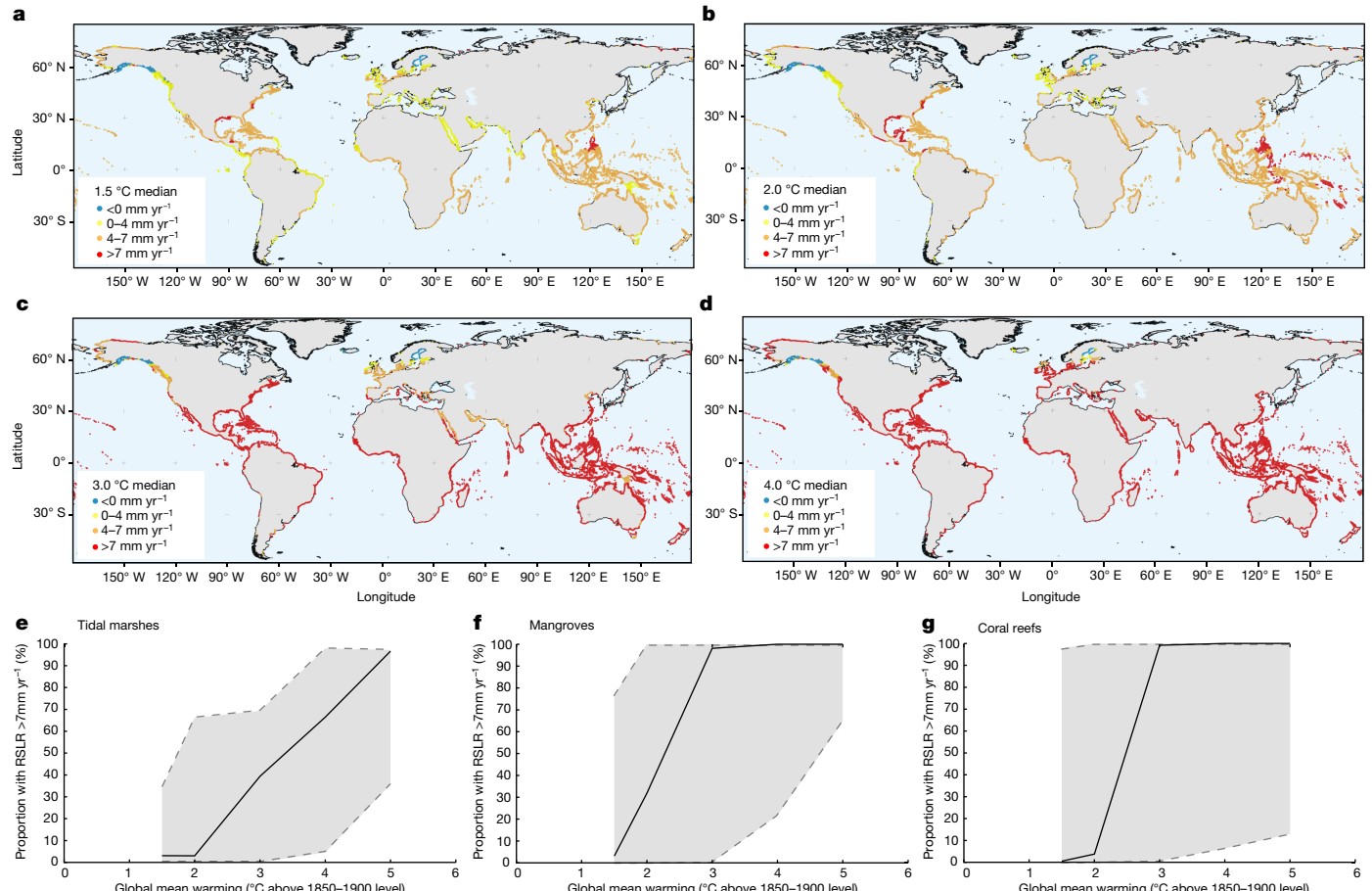

**Fig. 3 | Projected exposure of coastal ecosystems to RSLR. a–d**, Coastlines with mapped mangrove, tidal marsh or reef habitat subject to >4 mm yr⁻¹ and >7 mm yr⁻¹ RSLR over 2080–2100 under the median projections for 1.5 °C (**a**), 2.0 °C (**b**), 3.0 °C (**c**) and 4.0 °C (**d**) warming scenarios relative to 1850–1900. Note that projected rates of RLSR rely to a considerable extent on tide gauge records that may capture local anomalies (for example, due to fluid extraction)

that could produce locally higher rates. **e–g**, The proportion of global tidal marsh (**e**), mangrove (**f**) and coral reef (**g**) habitat subject to 7 mm yr⁻¹ of RSLR by 2100 in the scenarios shown in **a–d**, as well as the 5 °C scenario. Error bands show the 17–83% likely range. These projections do not take into account the possibility that ice sheet instabilities substantially increase RSLR in warming scenarios exceeding 2 °C.

period 1947–2014. These observations conform to RSLR thresholds modelled for the stability of reef islands in the Marshall Islands group based on palaeo sea-level reconstructions[41]. The reef islands in the western tropical Pacific provide insights into probable outcomes for intertidal wetland and supratidal islands on reef tops globally under conditions of accelerating RSLR projected to 2100.

## Projected response to future sea-level rise

Modelling of spatial variability in RSLR was completed in the IPCC AR6 for each warming scenario[42]. We compared regional RSLR projections to 2080–2100 with the distribution of mangroves, tidal marshes and coral reefs across the globe (Fig. 3; Methods). For each of the modelled scenarios, we determined the proportion of mangrove, tidal marsh and coral reef island habitat occurring where RSLR is projected to rise to levels for which eventual retreat of mangroves and tidal marshes is likely (4 mm yr⁻¹) or very likely (7 mm yr⁻¹), the best estimate from our combined palaeo and instrumental observations. For reef islands (a subset of mapped reefs), contraction or increasing island instability by RSLR of 7 mm yr⁻¹ is likely (Extended Data Fig. 2c), although we cannot yet specify a rate of RSLR at which contraction is highly likely, given the scarcity of contemporary observations at higher rates of RSLR, and because this threshold will vary with the rate of surrounding reef vertical growth, reef flat width, wave exposure, island size and height, and reef-derived sediment supply.

In the 1.5 °C scenario, the likely ($P \geq 0.66$) rate of GMSL rise at 2080–2100 is between 2.4 and 6.4 mm yr⁻¹. Coastlines subject to rates of RSLR of 4 to 7 mm yr⁻¹ correspond to centres of contemporary mangrove development, notably Southeast Asia and the Caribbean. Under this rate of RSLR, elevation deficits are likely ($P = 0.66$–$0.90$; Fig. 2). The probability of reaching a rate of RSLR at which elevation deficits are very likely (7 mm yr⁻¹) remains low (<11%), although coastlines subject to high rates of land subsidence—including, for example, the US Gulf Coast and Southeast Asian deltas[28,43]—are projected to exceed this rate. Median projections for the 2 °C warming scenario suggest that one third of global mangroves are subject to ≥7 mm yr⁻¹ and nearly all exposed to ≥4 mm yr⁻¹ of RSLR, although there is comparatively little change in the proportion of tidal marshes and reefs exposed to ≥7 mm yr⁻¹ of RSLR (Table 1). Under 3 °C of warming, nearly all tropical and subtropical latitude coastlines are exposed to ≥7 mm yr⁻¹ of RSLR, and these are the locations of most of the world's mangroves and coral reefs. Median RSLR projections along the world's coastlines therefore show the probability of elevation deficits in mangroves shifting from likely to very likely between 2 °C and 3 °C of global warming (Fig. 3 and Table 1).

At high latitudes, portions of coastline have declining RSLR owing to gravitational, rotational and elastic deformational effects resulting from mass loss of glaciers and the Greenland Ice Sheet offsetting GMSL rise. For this reason, proportional loss of existing tidal marsh with RSLR is expected to be lower than for mangroves with increased

**Table 1 | Median estimates (and 17–83% likely range) of the proportion of existing mangrove, tidal marsh and coral reef island vulnerable to elevation deficit and eventual loss under the five AR6 RSLR warming scenarios[17]**

| | Mangrove | | Tidal marsh | | Coral reefs |
|---|---|---|---|---|---|
| | Likely | Very likely | Likely | Very likely | Likely |
| 1.5 °C | 0.81 (0–1.0) | 0.03 (0–0.77) | 0.34 (0.03–0.69) | 0.03 (0–0.34) | 0.01 (0–0.98) |
| 2.0 °C | 0.99 (0.02–1.0) | 0.32 (0–1.0) | 0.65 (0.06–0.98) | 0.03 (0–0.66) | 0.04 (0–1.0) |
| 3.0 °C | 1.00 (0.75–1.0) | 0.98 (0–1.0) | 0.67 (0.31–0.98) | 0.39 (0–0.69) | 0.99 (0.01–1.0) |
| 4.0 °C | 1.00 (0.99–1.0) | 1.00 (0.22–1.0) | 0.70 (0.63–0.98) | 0.66 (0.04–0.98) | 1.00 (0.06–1.0) |
| 5.0 °C | 1.00 (1.0–1.0) | 1.00 (0.64–1.0) | 0.97 (0.93–0.97) | 0.97 (0.36–0.97) | 1.00 (0.13–1.0) |

For mangroves and tidal marshes, loss is the proportion of existing area exposed to 4 mm yr⁻¹ (likely loss; $P > 0.66$) and 7 mm yr⁻¹ (very likely loss; $P > 0.9$) of RSLR based on probability distributions presented in Fig. 2, and the RSLR modelling in Fig. 3. For coral reef islands, the proportion refers to numbers of reefs, and uses the conservative estimate of likely vulnerability to RSLR at 7 mm yr⁻¹ (the full dataset with uncertainties is presented as Extended Data Table 1).

warming. At 2 °C warming, the high-latitude European and North American west coasts remain below 4 mm yr⁻¹ RSLR under median estimates, and at 3 °C the Baltic Sea and Gulf of Alaska remain below 4 mm yr⁻¹. Tidal marsh habitat is likely to expand in extent in northern Siberia under higher RSLR owing to limited topographic and human development impediments (Fig. 4 and Extended Data Fig. 6). Far northern coastlines therefore emerge as important future habitats for tidal marsh—as also projected for seagrass meadows and kelp

forests[19,30,44]—under warmer temperatures and reduced ice cover and ice scour, increasing their relative contribution to blue carbon capture and storage at high latitudes.

## The influence of global change drivers

The behaviour of future ecosystems may not always be anticipated by palaeo and contemporary analogues. Processes influencing vertical adjustment of coastal wetlands and reefs to sea-level rise may be modified by climate change, though often the influence is to supress vertical adjustment. Land-use change driven by population growth may increase sediment supply by rivers, subsidizing sediment accumulation in coastal deltas[45,46]. Counteracting this is the association between economic development and dam construction, an intervention that retains sediment within catchments. Sediment yields to coastal environments in the global north are nearly half those prior to such hydrological modifications[45]. Major hydrological developments in Southeast Asian rivers have negative implications for the resilience of mangroves to sea-level rise[47].

Elevated concentrations of atmospheric $CO_2$ and associated climate change may modify biotic feedbacks to sea-level rise. Long-term field mesocosm experiments in Chesapeake Bay, USA have shown that root growth and marsh vertical adjustment was enhanced by the atmospheric $CO_2$ fertilization effect[48] and moderate warming (approximately 1.7 °C above ambient[49]). However, as observed RSLR increased above 7 mm yr⁻¹, water stress negated the benefit of elevated $CO_2$ (ref. 50), and temperatures above 1.7 °C increasingly promoted organic carbon remineralization, lowering elevation gain[50].

Ocean acidification and thermal stress will supress reef vertical growth due to impacts on coral cover, unless rapid adaptation occurs. Recent estimates identify low accretion potential (averaging 1.8 ± 2.2 mm yr⁻¹) across many tropical western Atlantic reefs[51], compared with rates derived from palaeo-reef core records[51]. Currently

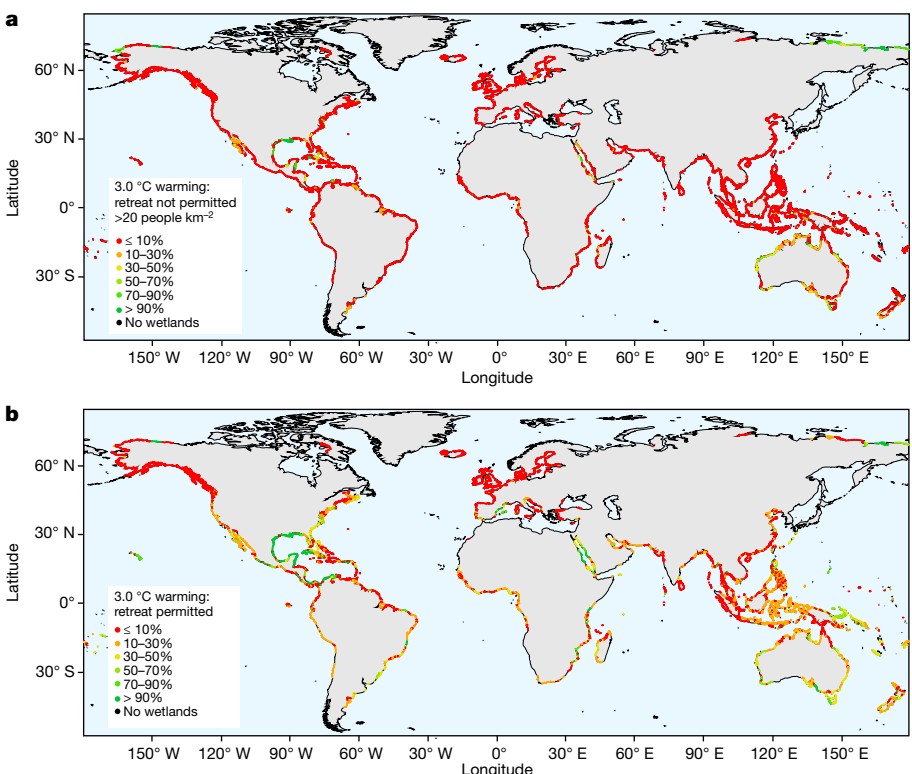

**Fig. 4 | Wetland inland retreat potential.** The percentage of the current wetland area that could potentially be compensated for via inland retreat until 2100, calculated for the 3.0 °C warming scenario (Methods). **a**, The scenario for wetland inland retreat capacity possible with a population density below 20 people per km². **b**, The scenario for wetland inland retreat unimpeded by population density (the no barriers scenario[14]; Methods). Scenarios restricting landward encroachment under lower populations density thresholds are shown in Extended Data Fig. 6.

less than half of reefs in the western Atlantic and Indian Ocean have maximum accretion potential rates matching altimetry-derived rates of sea-level rise[51]. Recent modelling of the impacts of climate change on reef accretion potential to 2100 suggest that increasingly severe and frequent bleaching events will further limit reef accretion potential[52] (even in the absence of other confounding local disturbance pressures). The potential of reef-top habitats and reef islands to accrete will therefore be influenced by increasing water depths above the surrounding fringing reefs and probable shifts in the abundance and production rates of biota from which sediment is derived. Both may negatively affect future reef-top habitats and will almost certainly impinge upon cultural use and sustainability[14].

## Implications for management

The committed loss of coastal habitats under high warming scenarios should not discourage conservation and restoration efforts. Under small elevation deficits, centuries may elapse before the elevation capital of a wetland is exhausted, and this will provide sufficient time for the supply of ecosystems services, including those critical for well-being and sustenance. Over the current century, landward migration driven by sea-level rise may compensate wetland loss, or even facilitate wetland expansion and associated carbon burial potential[53]. Extensive mangrove forest development in the mid-Holocene, coupled with high rates of vertical accretion under 4 to 7 mm yr$^{-1}$ RSLR promoted blue carbon capture and storage at a scale that may have contributed to an observed decline in global atmospheric $CO_2$ concentrations for this period[12]. In the near-term, increased GMSL potentially allows for the recolonization of these coastal floodplains, expanding mangrove area while promoting higher rates of organic carbon accumulation than currently encountered[53]. Although intensive coastal development in Asia has reduced coastal wetland extent in former biogeographic centres[36] and is likely to restrict landward retreat (Fig. 4), extensive coastal floodplains provide viable opportunities for mangrove landward migration and even aerial expansion in northern and northwestern Australia[54], the northern Gulf of Mexico[55], Siberia and—depending on opportunities for restoration in more populated areas—Central America, Colombia and the western Mediterranean (Fig. 4.). In the Gulf of Mexico and northern Australia, mangrove forest and tidal creek encroachment under higher rates of RSLR is already being observed[25,56].

The implication of the gap between the Paris Agreement aspiration (2 °C with an aim of 1.5 °C) and the pathways consistent with the implementation of current policies (2.4 °C to 3.5 °C by 2080–2100, medium confidence[57]) is profound for coastal ecosystems. Warming above 2 °C would restore the conditions faced by mangroves and tidal marshes under previous high RSLR periods and would likely expose most of the world's mangrove and two thirds of the world's tidal marsh to elevation deficits (Table 1). Warming of 3 °C by 2100 would accelerate GMSL rise to rates consistent with a high probability of eventual tidal marsh and mangrove retreat and increased reef island instability for much of their geographic extent. Once reached, these rates of RSLR are projected to persist for centuries to millennia[58]. The thermal inertia of ocean waters is likely to drive irreversible ice sheet grounding line retreat where bedrock slopes away from the coast[58], ensuring ongoing marine ice sheet instability[59]. Projected elevation deficits therefore define committed losses upon the exhaustion of elevation capital. Our analysis therefore suggests that the long-term contribution of blue carbon to climate mitigation is compromised under higher emissions scenarios. While preserving organic carbon in situ in many settings[12,53], narrower, younger and more transitional wetlands would predominate[23]. As a result, coastlines and reef islands that are currently protected will be increasingly exposed to erosion and retreat, consistent with palaeo observations[16,23].

Coastal ecosystems represent another of the numerous tipping elements for climate change impacts and rank among the more vital to human well-being and vulnerable to imminent warming levels[60]. The non-linear response to external forcing as seen in a wide range of ecosystems is closely associated with the concept of safe operating space, which promotes planetary boundaries being maintained a safe distance from critical thresholds of unacceptable environmental change[20]. Our findings demonstrate that the boundaries for a safe operating space for coastal ecosystems are approaching, and will be set by near-term emissions pathways. They also highlight the importance of mitigating against local environmental stressors (such as pollution in coral reefs) and restoring cleared and degraded wetlands to enhance resilience against climate change and coastal recession. In the face of irrevocable disruption under high rates of RSLR, the most effective means of promoting the continued survival of widespread mangrove forests, tidal marshes and coral reef islands is to achieve the Paris Agreement goal of net zero emissions by 2050. To this end, a contribution will be made by the preservation, restoration and landward accommodation of coastal blue carbon ecosystems.

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

## Methods

### Palaeo wetland response to RSLR

To estimate RSLR following the Last Glacial Maximum (Fig. 1), we use a revised numerical simulation of GIA[61], which adopts the ICE-6G global ice reconstruction from the Last Glacial Maximum to the present[62,63]. We use an ensemble of 300 combinations of rheological parameters in the GIA model to estimate RSL at 500-year time steps on a 512 × 260 global latitude × longitude grid, resulting in predictions of RSL at >130,000 points in space for each time step.

Post-glacial coastal habitat development and retreat prior to the Holocene are inferred from relict features that include the following: (1) drowned mangroves, tidal marshes and coral reefs, the remains of which have been found at around 90 m water depth[64], corresponding to meltwater pulse 1A (14.6–14.3 ka); and (2) relict features at around 50 m water depth[65,66], corresponding to a rapid rise in GMSL dating to 11.3–11.0 ka. Mangrove vertical development in relation to Holocene RSLR is based on 78 observations of the timing of the initiation of sustained mangrove peat development[12]. Evidence of post-glacial mangrove expansion is evident on the Sahul Shelf, Western Australia[67] and the Sunda Shelf, Southeast Asia[68] -12–14.5 ka, a phase ceasing during meltwater pulse 1A (Fig. 1). A relatively brief (~300-year) period of mangrove expansion and vertical development prior to 9 ka is documented in the western Ganges–Brahmaputra Delta[69], and the Queensland continental shelf[70], locations of high sediment delivery at the time[69,70]. Our GIA modelling (Fig. 1) suggests RSLR dipped to around 6 mm yr$^{-1}$ at this time. Mangroves in both sites were drowned during a period in which RSLR increased to circa 7 mm yr$^{-1}$ ~9 ka. A pan-tropical expansion in mangrove development and sustained vertical adjustment[54,71–75] commenced from ~8.5 ka as RSLR declined[12] to <6 mm yr$^{-1}$ according to GIA modelling (Fig. 1). A subset of these observations representative of the global dataset[12] are provided in Fig. 1, including Twin Cays, Belize[76]; Swan Key, FL, USA[77]; Pakhiralaya, Western Ganges, India[69]; Porto-Novo, Benin[78]; Mekong Delta, Cambodia[79]; Makoba Bay, Zanzibar[80]; Ord River, Western Australia[71]; and Mulgrave River, Queensland, Australia[81]. Reef island development commenced in the Pacific during the mid-Holocene corresponding to RSLR stabilization and fall. The example provided in Fig. 1 is Bewick Cay, Queensland, Australia[24].

Tidal marsh vulnerability to Holocene RSLR presented in Fig. 2 is based on the data from two studies utilizing multiple proxies across the UK[15] and the Mississippi Delta[16]. Holocene RSL data was compiled for 54 regions from Great Britain with the rate of RSL varying in relation to proximity to the centre of the Last Glacial Maximum British–Irish Ice Sheet[15]. RSLR rates estimated from GIA model predictions were compared to sea-level tendencies for 781 tidal marsh index points[15] (positive $n = 403$; negative $n = 360$; no tendency $n = 19$). For the Mississippi Delta, the Holocene RSL history was inferred from 72 sea-level index points, with marsh tendency assessed using 334 boreholes showing a well-defined Pleistocene–Holocene transition overlain by at least 2 m of sediment[16].

### Contemporary wetland response to RSLR

Assessment of mangrove and tidal marsh vertical adjustment was conducted using the SET-MH technique. This globally distributed network of monitoring stations[82] combines a stable benchmark rod against which measurements of elevation change are made, with an artificial marker horizon introduced at the time of benchmark rod installation, against which sediment accretion is measured (Extended Data Fig. 1). Pins extended from a portable arm (the surface elevation table) extend to the marsh surface, measuring surface elevation change in relation to the base of the benchmark rod. Comparison with elevation gain can then be made against water level changes measured at nearby tide gauges[29].

This technique was previously used[29] to test how elevation gain at 477 SET-MH monitoring stations compared to RSLR changes measured over the same period. To this analysis (presented as Fig. 2e) we have added a mangrove SET-MH network of 190 SET-MH stations (Fig. 2f), the location of which are provided in Extended Data Table 2. These data combine published rates of elevation gain with new measurements reported here (Extended Data Table 2). RSLR for the period of SET-MH measurement was extracted from tide gauge records provided by the National Oceanic and Atmospheric Administration (https://tidesandcurrents.noaa.gov/sltrends/sltrends.html) and, for Australia, the Australian Baseline Sea Level Monitoring Project (http://www.bom.gov.au/oceanography/projects/abslmp/abslmp.shtml).

### Contemporary habitat distribution

The contemporary distribution and extent of mangroves[83] (https://doi.org/10.34892/07vk-ws51), tidal marshes[84] (https://doi.org/10.34892/w2ew-m835) and coral reefs[85] (Figs. 1, 3 and 4) was accessed from the Ocean Data Viewer (https://data.unep-wcmc.org), hosted by the UN Environment World Conservation Monitoring Centre. An important caveat in relation to the representation of tidal marshes is the poor coverage of their possible extent at high northern latitudes. For Fig. 3, the coral reef dataset was complemented by additional data on the global distribution of atolls, which was sourced from the World Atolls database[86] (https://www.arcgis.com/home/item.html?id=1c18adf04d9e47669281061ff60167e1).

### Surface water and marsh change analysis

Because mangrove canopy cover obscured surface water observations, we report changes in surface water occurrence, and conversion of wetland to open water only for tidal marshes. We used two earth-observation derived global datasets to estimate tidal marsh conversion to open water across the SET-MH monitoring network. The Global Tidal Wetland Change (GTWC) dataset[36] depicts losses and gains of tidal marshes, tidal flats and mangroves (collectively termed 'tidal wetlands') at 30-m resolution over a 20-yr period (1999-2019). The data were developed through a machine learning classification of more than 1.1 million Landsat scenes acquired over the global coastal zone since 1999. GTWC data layers include tidal wetland losses, gains, and the probability of occurrence of tidal wetlands for the first (1999) and last (2019) time steps of the analysis. The Global Surface Water dataset depicts the location and temporal distribution of surface water from 1984 to 2020 at 30-m resolution[35]. The data were generated from >4.4 million Landsat scenes by individually classifying each Landsat pixel into water and non-water using an expert system. Although the two datasets are developed using Landsat data, the datasets differ in their temporal spans (2 to 4 decades), methodological approaches to mapping change dynamics, post-processing methods and minimum mapping unit. We therefore used both datasets to estimate the extent of tidal marsh conversion to open water in relation to observed RSLR (Supplementary Data 1).

To estimate net tidal wetland change and the extent of conversion to open water at each SET-MH monitoring site, we developed a buffer feature around the SET installation with an area of 5 km$^2$. For global tidal wetland change, the area of losses and gains of each tidal wetland ecosystem type (tidal marshes, tidal flats and mangroves) was computed, yielding a net change estimate of tidal wetlands associated with each SET site. For global surface water, we used the water occurrence change intensity layer, which is computed as the absolute difference in the per pixel mean water occurrence between two distinct epochs[35] (1984–1999 and 2000–2020). The average surface water change in each SET buffer feature was computed (Supplementary Data 1).

The relationships between surface water and tidal wetland change versus contemporaneous RSLR and elevation deficit were tested using multiple linear regression. Predictive variables are provided in Extended Data Table 3, and consist of climatic, hydrological and edaphic properties associated with each SET-MH station, and are sourced from ref. 39. Potential collinearity of predictors was assessed using variance

inflation factor from the car package[87]. The variance inflation factor was found to be below the level usually considered problematic (3.22). The overall relative importance of the key predictors was assessed using random forest regression analyses[88], a machine learning approach which tallies the results of small classification trees ($n = 20,000$) while retaining a bootstrapped subset of all observations for out-of-bag (internal) error testing. Analyses were performed in R version 4.1.3 and presented as Extended Data Fig. 4b.

### Island contraction and expansion

Data on island contraction or expansion (Extended Data Fig. 3) were sourced from recent assessments and reviews[37,40,89,90] (total island $n = 872$: Supplementary Data 2). We compared the proportion of islands showing areal contraction or expansion, binned at 1 mm yr$^{-1}$ RSLR increments, using the rate of RSLR cited for each reef island in the manuscripts. Islands were considered stable if change was less than 3% of the original area, following ref. 37.

### Ecosystem stability under RSLR

Contemporary marsh and mangrove resilience to RSLR was inferred from data from the globally distributed mangrove and tidal marsh SET-MH networks. The elevation surplus or deficit of each site was estimated by comparing the rate of tidal marsh surface elevation change recorded by the SET to rates of RSLR over the period of operation of the SET. RSLR was sourced from the National Oceanic and Atmospheric Administration's Laboratory for Satellite Altimetry (https://tidesand-currents.noaa.gov/sltrends/). The elevation surplus/deficit of each SET site was categorized in Extended Data Fig. 2d,e as in surplus if surface elevation change exceeded the RSLR rate by 1 mm yr$^{-1}$, stable if surface elevation change was within ±1 mm yr$^{-1}$ of the RSLR rate, or in deficit if the RSLR rate exceeded surface elevation change by 1 mm yr$^{-1}$. The stacked histograms in Extended Data Fig. 2d,e show the proportion of elevation budget categories in relation to RSLR rates (1 mm yr$^{-1}$ bin size) at each tidal marsh (Extended Data Fig. 2d) and mangrove (Extended Data Fig. 2e) SET site.

The resilience of palaeo tidal marsh to RLSR is represented in Extended Data Fig. 2a,b for the UK and Mississippi Delta, respectively. A 'negative' sea-level tendency—indicating tidal marsh advance—is identified by decreasing marine influence (that is, regressive contact), whereas a 'positive' sea-level tendency—which indicates tidal marsh retreat—is identified by increasing marine influence (that is, transgressive contact) in sediment archives. In the example core (Extended Data Fig. 2a), the contact between an intertidal mud and tidal marsh peat, which represents a negative tendency and marsh advance, was dated to ~8,439–8,956 years ago. The thin accumulation of tidal marsh peat is overlain by an intertidal mud, representing a positive tendency and marsh retreat; this event was dated to 8,501–8,959 years ago. RSLR rates were estimated for the timing of these marsh advance and retreat events recorded in the stratigraphy using a GIA model. The stacked histogram (Extended Data Fig. 2a) shows the proportion of these events from sediment archives across the UK in relation to RSLR rates (0.5 mm yr$^{-1}$ bin size). The facies succession identified in sediment cores from the Mississippi Delta (Extended Data Fig. 2b) were categorized based on the following criteria: a 'terrestrial' succession—indicating no evidence of marsh 'drowning'—is associated with the presence of terrestrial (marsh) mud or peat throughout the core and an absence of lagoonal facies; 'gradual drowning'—indicating marsh drowning that occurred over centuries—identified by at least a 30-cm-thick unit of marsh mud or peat occurring beneath lagoonal mud; 'rapid drowning'—indicating marsh drowning that occurred over about half a century—associated with less than a 30-cm-thick unit of marsh mud or peat occurring beneath lagoonal facies. The contact between marsh and lagoonal facies representing gradual or rapid marsh drowning was radiocarbon dated to determine the timing of the event, and RSLR rates at that time were estimated from an RSLR record obtained from compaction-free

basal peats from the Mississippi Delta. The proportion of each type of facies succession is shown in comparison to estimated rates of RSLR (0.5 mm yr$^{-1}$ bin size).

The 'initiation' of sustained mangrove accretion (Extended Data Fig. 2c) (at least 2 m of mangrove sediment) was radiocarbon dated and RSLR rates at that time interval were estimated from an ensemble of GIA model predictions[12]. The histogram shows the probability density (distribution) of initiation events in relation to RSLR rates (1 mm yr$^{-1}$ bins).

We summarized the probability thresholds at which marsh or mangrove elevation deficit becomes likely ($P \geq 0.66$) or very likely ($P \geq 0.90$), adopting IPCC likelihood language[91]. To estimate the probability of a negative tendency (Fig. 2b,c) or elevation deficit (Fig. 2e,f) conditional on rates of RSLR, we follow ref. 15 by modelling the elevation budget or facies successions as binary response variables (elevation deficit or drowning, 1; elevation surplus or terrestrial, 0) in a Bayesian framework. We chose the bin widths for histograms and the number of segments in the Bayesian analysis by visual inspection for best fit. Details of the probabilistic analysis used to estimate the relationship between mangrove initiation and RSLR rates (Fig. 3f) can be found in ref. 4.

### Sea-level rise projections

Sea-level rise projections (Figs. 3 and 4) were those used in the IPCC AR6 and were sourced from https://doi.org/10.5281/zenodo.5914710[42]. Sea-level rise scenarios to 2100 were converted to point shapefiles for the median, 17th and 83rd percentile projections for the following warming-level-based scenarios: 1.5 °C; 2.0 °C, 3.0 °C, 4.0 °C and 5.0 °C. The 17th–83rd percentile ranges are associated with the assessed IPCC likely range; the IPCC assessment is that there is at least a 66% chance that the true value will fall within this range. From the AR6 sea-level rise scenarios, sea-level rise rates at 2100 were converted to raster format (cell size 1 degree) for the median, 17th and 83rd percentile projections for the above-listed temperature-limited scenarios. All land-based pixels (defined as pixels where sea-level rise rates were zero for all percentiles of one temperature scenario) were converted to NoData.

### Ecosystem exposure to projected RSLR

For Fig. 3, available polygons for tidal marshes, mangroves and coral reefs were converted to point files based on each polygon's centroid coordinates. Where polygon features consisted of multiple polygons, polygon features were split into single-polygon features before converting them to centroid points. All resulting polygon centroids were merged with the available point data for mangroves, tidal marshes and reef islands into a dataset containing 1,885,466 entries. To visualize the spatial variability of wetland exposure to local sea-level rise (Fig. 3a–d), local RSLR rates (incorporating vertical land movement)[42] were extracted from the median projections of the temperature-limited scenarios 1.5 °C, 2.0 °C, 3.0 °C and 4.0 °C and classified into the following RSLR rate exposure categories: <0 mm yr$^{-1}$ (blue), 0–4 mm yr$^{-1}$ (yellow), 4–7 mm yr$^{-1}$ (orange) and >7 mm yr$^{-1}$ (red).

To calculate proportional changes of exposure to local RSLR rates for all five temperature-limited scenarios (Fig. 3e–g), only the available polygons for salt marshes, mangroves and coral reefs were utilized, as those included accurate aerial information. As above, polygon data were converted to point files, based on their centroid locations, but to preserve the accurate aerial information multi-polygon features were not split up. All local RSLR rates of each scenario (five temperature scenarios, with three percentiles each) were extracted for each ecosystem category to calculate proportional exposure to RSLR rates <0 mm yr$^{-1}$, 0–4 mm yr$^{-1}$, 4–7 mm yr$^{-1}$ and >7 mm yr$^{-1}$. For each temperature scenario, the respective uncertainty range was defined by the lower (17%) and upper (83%) percentiles respectively.

## Modelling retreat potential

AR6 RSLR data up to 2100 were utilized to model the inland retreat space of coastal wetlands available for two RSLR scenarios: the 2 °C and 3 °C warming levels (Extended Data Fig. 6). The 3 °C warming level, representing the greater potential landward retreat, is also presented in Fig. 4. This modelling relies on the global coastal wetland model, which assumes inland retreat can occur where local population densities are below a pre-defined population density threshold and the coastal topography provides sufficiently flat inland areas[14].

AR6 RSLR data (RSLR between 2020 and 2100) were retrieved from https://doi.org/10.5281/zenodo.5914710[42] with a spatial resolution of 1 degree. Trajectories of RSLR for each coastal segment, the spatial unit that the global coastal wetland model is based on ref. 92, were derived from the data point located closest (Euclidean distance) to the center of the respective coastline segment. Inland retreat space was calculated as the area additionally inundated during mean high water spring conditions, under future RSLR scenarios, and expressed as percentage of current wetland extents[14]. High water spring levels were thereby assumed to rise at the same rate as mean sea level. Local topographical profiles were calculated based on global Shuttle Radar Topography Mission data[93] and on the method first presented in ref. 94.

Taking into account the widespread obstruction that human coastal infrastructure imposes on coastal wetland inland retreat[95] and assuming that the extent of obstruction is a function of population density, wetland inland retreat was accounted for only where population densities within the local 1-in-100 year floodplain are below a threshold of 20 people per $km^2$ as a best case of these scenarios, a threshold of 5 people per $km^2$ as a worst case scenario. This range has previously been estimated to represent current conditions for the existence of barriers to coastal wetland inland retreat[14]. Meanwhile, population density has been subjected to estimated population growth following the 'middle-of-the-road' shared socio-economic pathway (SSP2)[96]. We also modelled potential landward space available for ecosystem redistribution ignoring the potential impediment of population density, the no barriers scenario (Fig. 4).

## Reporting summary

Further information on research design is available in the Nature Portfolio Reporting Summary linked to this article.

## Data availability

Data contributing to the analysis are contained in Supplementary Data 1 and Supplementary Data 2, available at https://doi.org/10.5281/zenodo.7787502[97].

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

**Acknowledgements** We thank the authors of the IPCC projection for developing and making the sea-level rise projections available, multiple funding agencies for supporting the development of the projections, and the NASA Sea-Level Change Team for developing and hosting the IPCC AR6 Sea-Level Projection Tool. N.S. was supported by an Alexander Von Humboldt Research Award. R.E.K., G.G.G. and E.L.A. were supported by awards from the US National Aeronautics and Space Administration (80NSSC17K0698, 80NSSC20K1724 and JPL task 105393.509496.02.08.13.31) and National Science Foundation (ICER-1663807, ICER-2103754, OCE-1702587 and OCE-2002437). B.H. and T.A.S. were funded by the Ministry of Education Academic Research Fund MOE2019-T3-1-004, the National Research Foundation Singapore, and the Singapore Ministry of Education, under the Research Centres of Excellence initiative and the National Sea Level Programme Funding Initiative (Award USS-IF-2020-1), administered by the National Environment Agency, Singapore and supported by the National Research Foundation, Singapore. Any opinions, findings and conclusions or recommendations expressed in this material are those of the author(s) and do not reflect the views of the NRF, MND and NEA. T.E.T. was funded by the US National Science Foundation (OCE-0601814, EAR-1349311 and OCE-1502588). M.S. has received funding from the European Union's Horizon 2020 research and innovation programme under grant agreement no. 101037097

(REST-COAST project). C.L. was funded by the Australian Research Council award FL200100133. K.R. and C.W. were funded by the Australian Research Council Award DP210100739. The authors acknowledge PALSEA (Palaeo-Constraints on Sea-Level Rise), a working group of the International Union for Quaternary Sciences (INQUA) and Past Global Changes (PAGES), which in turn received support from the Swiss Academy of Sciences and the Chinese Academy of Sciences. This article is a contribution to HOLSEA (Geographic variability of Holocene sea level) and International Geoscience Program (IGCP) Project 725, 'Forecasting Coastal Change'. This work is Earth Observatory of Singapore contribution 537.

**Author contributions** N.S. conceived the project and led the writing. G.G.G. and R.E.K. provided sea-level projections. E.L.A. conducted modelling for Figs. 1 and 2. M.S. led the modelling and analysis leading to Figs. 3 and 4. T.A.S., N.S.K. and M.S. drafted figures. N.S., B.H., T.E.T., K.R., J.K., M.M.G., C.E.L., L.B.H., L.F. and C.D.W. contributed to the assessment of tidal wetland responses. N.S., C.P., S.A., N.S.K. and C.D.W. contributed to the assessment of coral reef and reef-top habitat responses. All authors contributed to manuscript editing.

**Competing interests** The authors declare no competing interests.

**Additional information**
**Correspondence and requests for materials** should be addressed to Neil Saintilan.

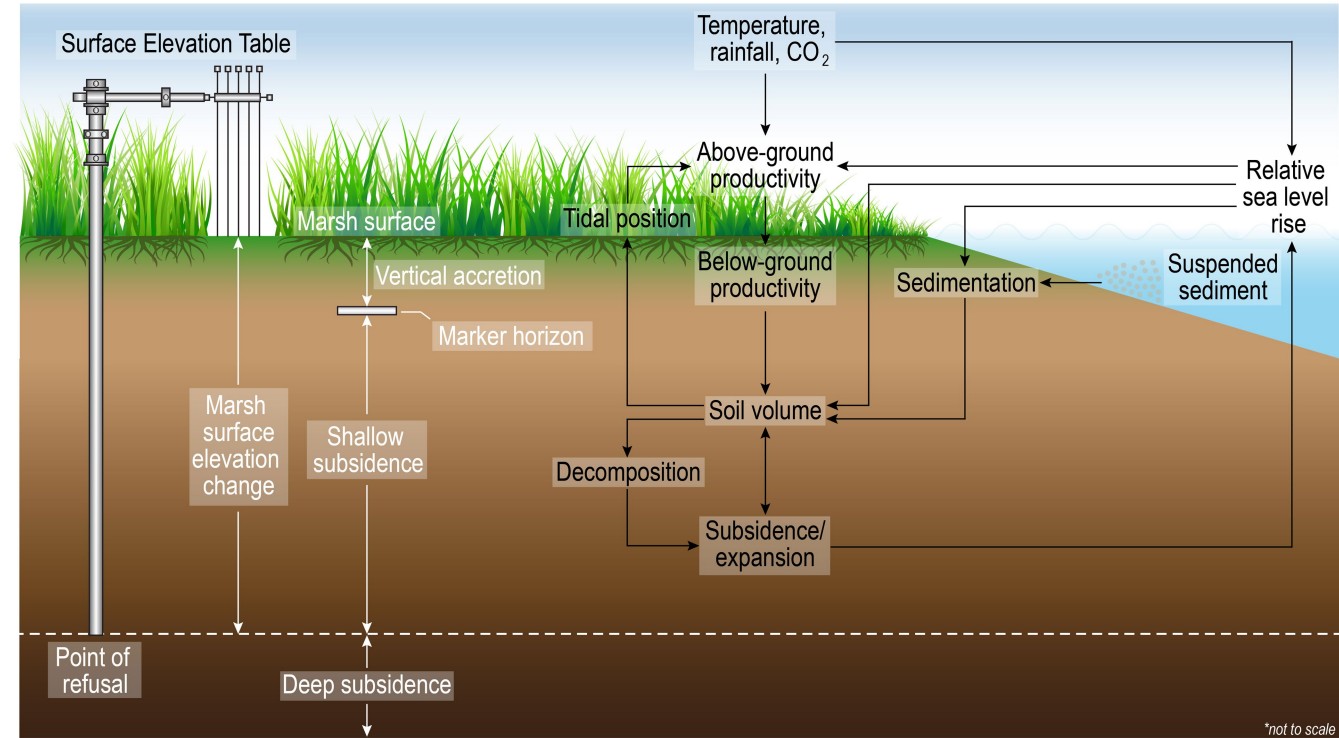

**Extended Data Fig. 1 | Components of the Surface Elevation Table - Marker Horizon system.** A benchmark rod driven to the point of refusal serves as a vertical benchmark against which tidal marsh/mangrove elevation gain or loss is measured. At the time of installation, an introduced horizon (feldspar or similar) is placed on the wetland surface, against which sediment and organic accretion is measured. These measurements allow for the inference of shallow subsidence, and the relation between elevation gain and RSLR.

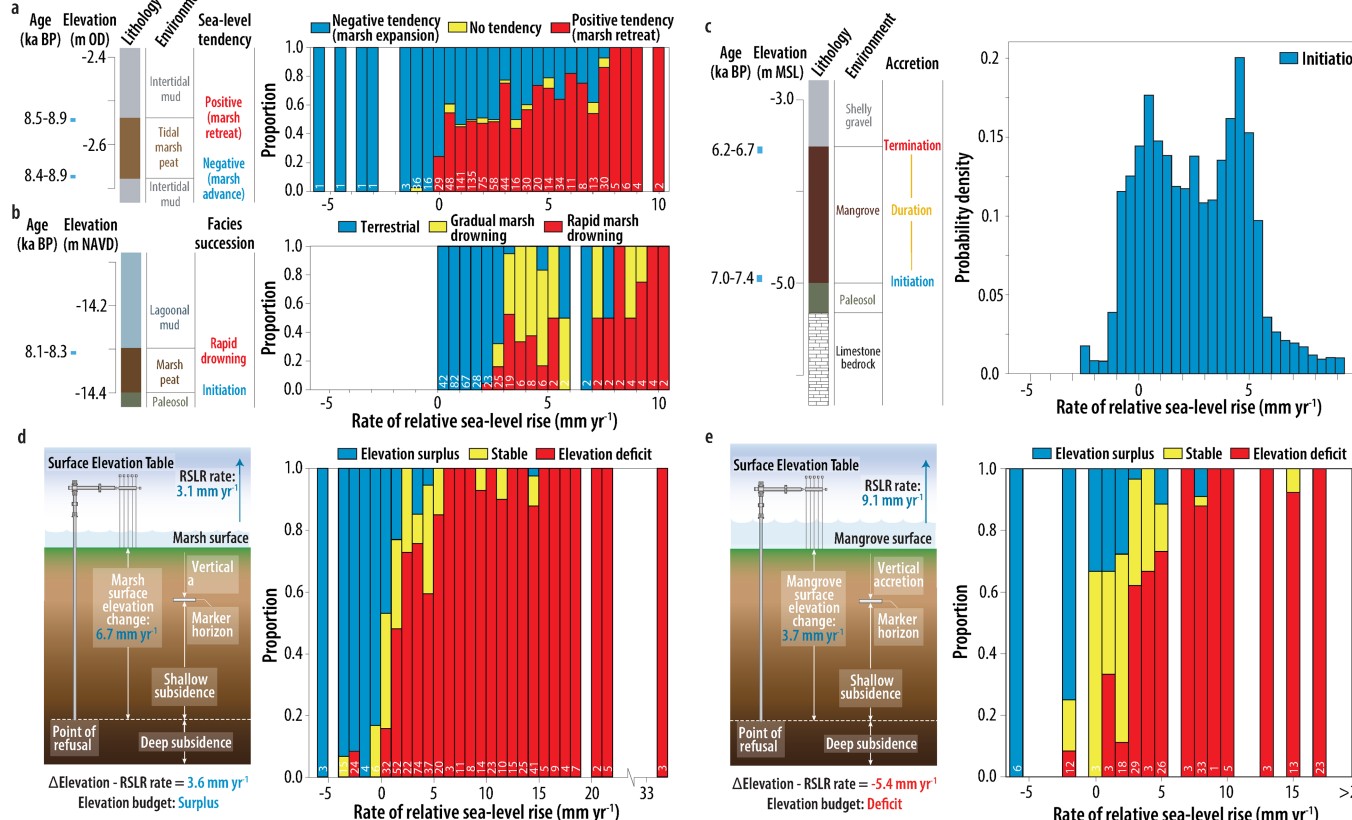

**Extended Data Fig. 2 | Details of individual studies from which the probabilities of marsh and mangrove vertical adjustment to RSLR were inferred.** (a) Analysis of tidal marsh retreat and advance recorded in sediment archives from across the United Kingdom[15]. (b) Analysis of marsh drowning recorded in sediment archives from the Mississippi Delta[16]. (c) Analysis of global mangrove accretion recorded in sediment archives from[12]. (d) Analysis of surface elevation tables (SET) from global tidal marshes[29]. (e) Analysis of surface elevation tables (SET) for mangrove SET sites in Extended Data Table 2.

For a, b, d, and e probabilistic analysis shown in Fig. 2 follows[15] and[29], where the probability of an elevation deficit/surplus, marsh retreat/drowning, or rapid drowning/terrestrial succession was modelled as a binary response variable, and the relationship of this response with rates of RSLR was estimated in a Bayesian framework. Details of the probabilistic analysis used in e to estimate the relationship between mangrove initiation and RSLR rates (c) can be found in[12]. Numbers of observations for each RSLR increment are shown at the base of each column.

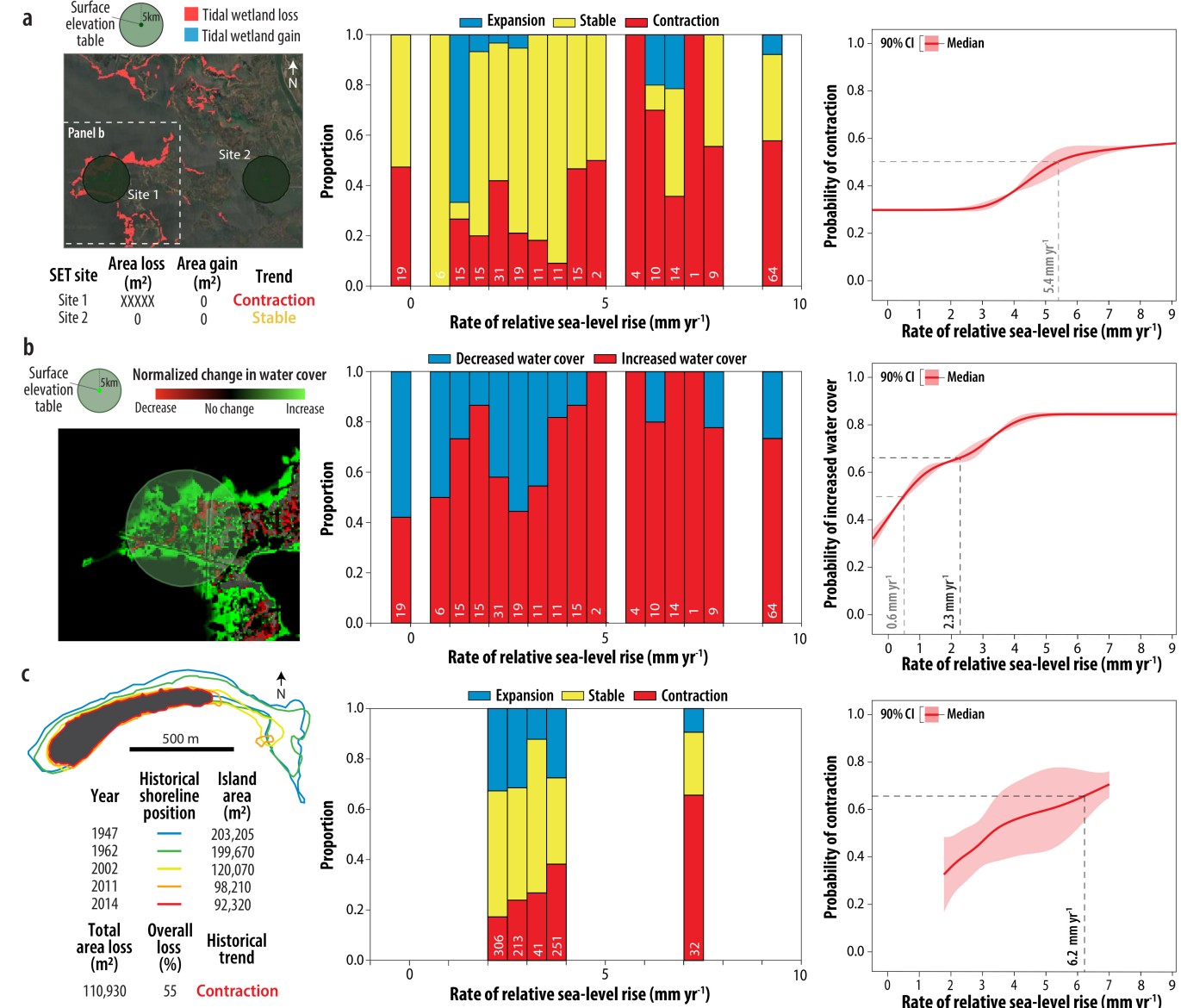

**Extended Data Fig. 3 | Probability of conversion to open water with RSLR.** Tidal marsh area change 1999-2019[36] within a 5 km² of tidal marsh SET-MH sites in relation to RSLR (a). Change in surface water occurrence at the tidal marsh SET-MH sites[29], comparing 20 years pre-2000 and post-2000[35] in relation to RSLR trends during the period of SET measurement[29] (b). Reef island planform change in relation to RSLR (c: area change and RSLR data from studies in Supplementary Information Data 2). Numbers of observations for each RSLR increment are shown at the base of each column. Image data in (a) are from Google Earth.

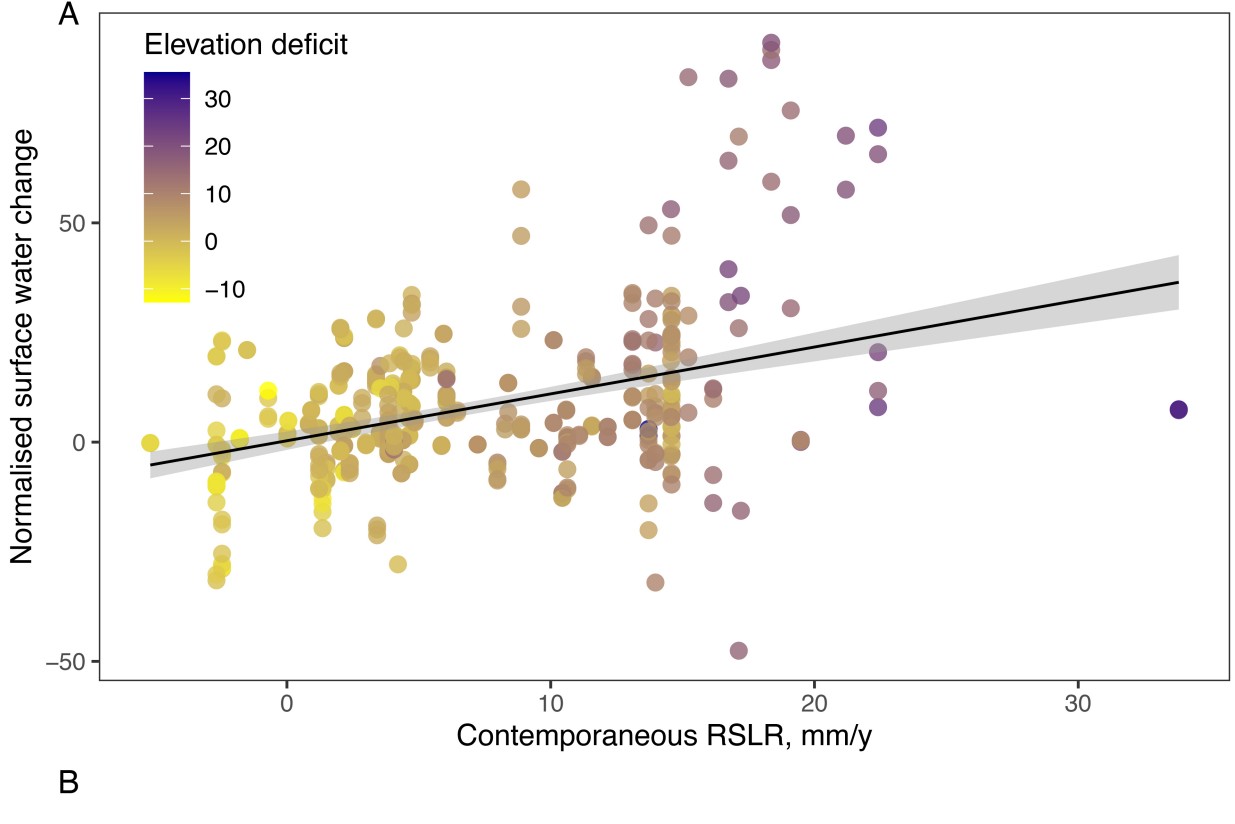

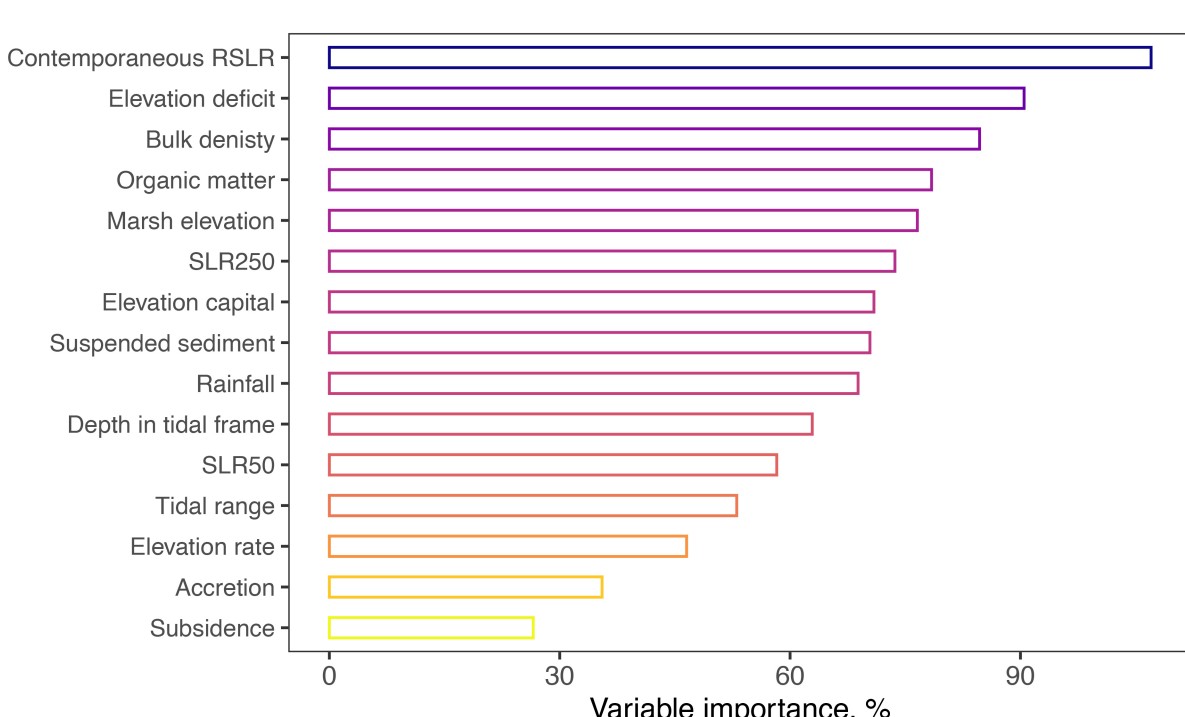

**Extended Data Fig. 4 | Surface water area in tidal marshes increases with RSLR and elevation deficit.** A: Normalised surface water change (change in the % occurrence of surface water) comparing the two decades pre-2000 and post-2000[35] at the tidal marsh SET-MH monitoring sites (n = 476)[29]. The presence of surface water increases with RLSR (r[2] = 0.16, P<0.001). Grey shading indicates the 95% confidence level interval for linear model predictions. B: Results of Random Forest analysis predicting the normalised surface water change at the SET-MH monitoring sites, based on predictive variables sourced from[29] (listed in Extended Data Table 3).

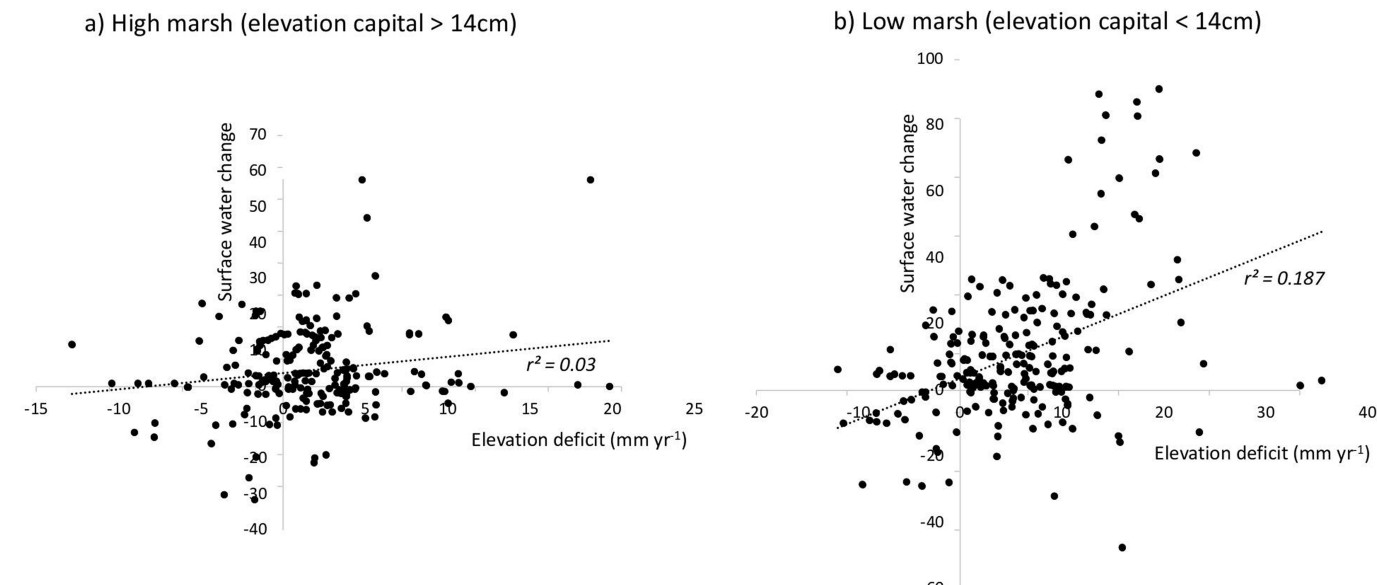

**Extended Data Fig. 5 | Marsh elevation, elevation deficit and surface water change.** Normalised surface water change (change in the % occurrence of surface water) comparing the two decades pre-2000 and post-2000[35] at the tidal marsh SET-MH monitoring sites[29], in relation to the deficit between elevation gain and RSLR. For marshes above the median marsh elevation of 14 cm above the lower limits of survival (a), the relationship is weak (linear regression $r^2$ = 0.03, P = 0.009; n = 298). For marshes below this median marsh elevation (b), normalised surface water change increases with the size of the elevation deficit (linear regression $r^2$ = 0.187, P < 0.001; n = 231). Elevation capital is the elevation of the marsh above the lower limits of survival, at which open water conversion would be expected.

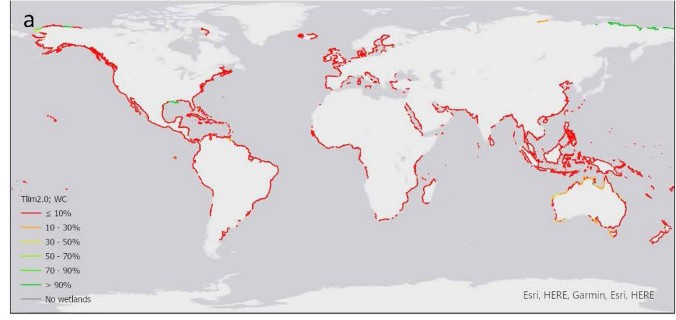

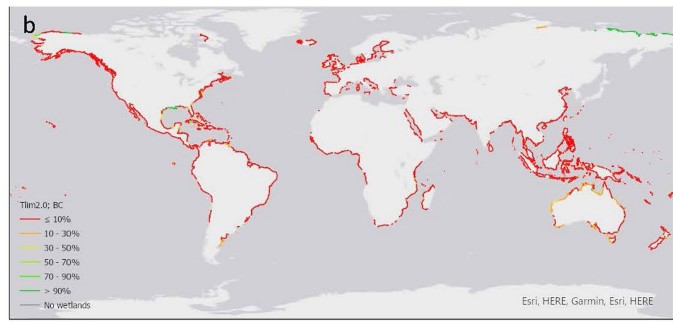

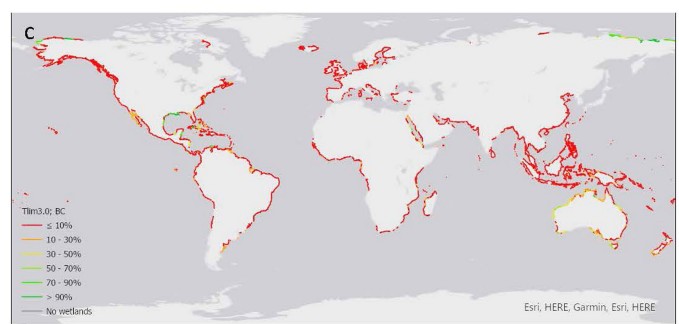

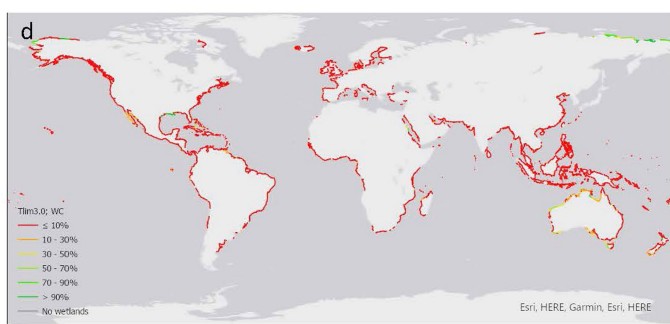

**Extended Data Fig. 6 | Inland retreat potential of existing mangrove and tidal marsh.** Percentage of the current wetland area that could potentially be compensated for via wetland inland retreat until 2100, calculated for two sea-level rise scenarios. Median projections for 2.0 °C warming, allowing wetland retreat to a population threshold of (a) 5 people km$^{-2}$ (worst case scenario, WC) and (b) 20 people km$^{-2}$ (best case scenario, BC) Median projections for 3.0 °C warming, allowing wetland retreat to a population threshold of (c) 5 people km$^{-2}$; and (d) 20 people km$^{-2}$.

**Extended Data Table 1 | Area (km², and proportion) of mangrove, tidal marsh and number of coral reefs exposed to various rates of RSLR under policy-relevant warming scenarios**

**Saltmarsh**

| SLR scenario | <0 mm/yr | 0-4 mm/yr | >4 mm/yr | >4 mm/yr proportion | Uncertainty | >7 mm/yr proportion | Uncertainty | Avg. SLR | Global SLR |
|---|---|---|---|---|---|---|---|---|---|
| 1.5°C,17% | 126590 | 89612 | 6821 | 0.0306 | 0.31 | 0.0000 | 0.03 | 0.13 | 2.4 |
| 1.5°C,50% | 67933 | 79732 | 75359 | 0.3379 | | 0.0305 | | 3.57 | 4.2 |
| 1.5°C,83% | 1082 | 67513 | 154428 | 0.6924 | 0.35 | 0.3420 | 0.32 | 7.29 | 6.4 |
| 2.0°C,17% | 74162 | 133715 | 15147 | 0.0679 | 0.58 | 0.0004 | 0.03 | 0.89 | 3.2 |
| 2.0°C,50% | 67803 | 11045 | 144175 | 0.6465 | | 0.0302 | | 4.65 | 5.4 |
| 2.0°C,83% | 829 | 4322 | 217872 | 0.9769 | 0.33 | 0.6616 | 0.63 | 8.9 | 8.2 |
| 3.0°C,17% | 82210 | 72018 | 68796 | 0.3085 | 0.36 | 0.0000 | 0.39 | 2.09 | 5.2 |
| 3.0°C,50% | 68112 | 5445 | 149467 | 0.6702 | | 0.3939 | | 6.65 | 7.8 |
| 3.0°C,83% | 2597 | 2423 | 218003 | 0.9775 | 0.31 | 0.6931 | 0.30 | 11.88 | 11.8 |
| 4.0°C,17% | 71682 | 9963 | 141379 | 0.6339 | 0.06 | 0.0462 | 0.62 | 3.68 | 7.4 |
| 4.0°C,50% | 4765 | 63213 | 155046 | 0.6952 | | 0.6647 | | 8.63 | 10.2 |
| 4.0°C,83% | 2573 | 1864 | 218587 | 0.9801 | 0.28 | 0.9786 | 0.31 | 14.42 | 14.8 |
| 5.0°C,17% | 5691 | 5743 | 145646 | 0.9272 | 0.04 | 0.3561 | 0.61 | 5.05 | 8.8 |
| 5.0°C,50% | 4765 | 273 | 152042 | 0.9679 | | 0.9665 | | 10.55 | 12.2 |
| 5.0°C,83% | 2573 | 1864 | 152643 | 0.9718 | 0.004 | 0.9718 | 0.01 | 17.2 | 17.8 |

**Mangrove**

| SLR scenario | <0 mm/yr | 0-4 mm/yr | >4 mm/yr | >4 mm/yr proportion | Uncertainty | >7 mm/yr proportion | Uncertainty | Avg. SLR | Global SLR |
|---|---|---|---|---|---|---|---|---|---|
| 1.5°C,17% | 15939 | 128246 | 369 | 0.0026 | 0.81 | 0 | 0.03 | 0.13 | 2.4 |
| 1.5°C,50% | 0 | 27377 | 117176 | 0.8106 | | 0.0311 | | 3.57 | 4.2 |
| 1.5°C,83% | 0 | 0 | 144554 | 1.0000 | 0.19 | 0.7671 | 0.74 | 7.29 | 6.4 |
| 2.0°C,17% | 6390 | 134704 | 3460 | 0.0239 | 0.97 | 0 | 0.32 | 0.89 | 3.2 |
| 2.0°C,50% | 0 | 1399 | 143154 | 0.9903 | | 0.3182 | | 4.65 | 5.4 |
| 2.0°C,83% | 0 | 0 | 144554 | 1.0000 | 0.01 | 1 | 0.68 | 8.9 | 8.2 |
| 3.0°C,17% | 0 | 36252 | 108302 | 0.7492 | 0.25 | 0.0026 | 0.98 | 2.09 | 5.2 |
| 3.0°C,50% | 0 | 0 | 144554 | 1.0000 | | 0.9816 | | 6.65 | 7.8 |
| 3.0°C,83% | 0 | 0 | 144554 | 1.0000 | 0 | 1 | 0.02 | 11.88 | 11.8 |
| 4.0°C,17% | 0 | 972 | 143582 | 0.9933 | 0.01 | 0.2161 | 0.78 | 3.68 | 7.4 |
| 4.0°C,50% | 0 | 0 | 144554 | 1.0000 | | 1 | | 8.63 | 10.2 |
| 4.0°C,83% | 0 | 0 | 144554 | 1.0000 | 0 | 1 | 0 | 14.42 | 14.8 |
| 5.0°C,17% | 0 | 561 | 143992 | 0.9961 | 0.00 | 0.6466 | 0.35 | 5.05 | 8.8 |
| 5.0°C,50% | 0 | 0 | 144554 | 1.0000 | | 1 | | 10.55 | 12.2 |
| 5.0°C,83% | 0 | 0 | 144554 | 1.0000 | 0 | 1 | 0 | 17.2 | 17.8 |

**Coral Reef**

| SLR scenario | <0 mm/yr | 0-4 mm/yr | >4 mm/yr | >4 mm/yr proportion | Uncertainty | >7 mm/yr proportion | Uncertainty | Avg. SLR | Global SLR |
|---|---|---|---|---|---|---|---|---|---|
| 1.5°C,17% | 764429 | 131102 | 2521 | 0.0028 | 0.97 | 3.3406E-06 | 0.01 | 0.13 | 2.4 |
| 1.5°C,50% | 0 | 21753 | 876300 | 0.9758 | | 0.00556204 | | 3.57 | 4.2 |
| 1.5°C,83% | 0 | 0 | 898052 | 1.0000 | 0.02 | 0.97821284 | 0.97 | 7.29 | 6.4 |
| 2.0°C,17% | 377 | 869497 | 28177 | 0.0314 | 0.97 | 0.00096097 | 0.04 | 0.89 | 3.2 |
| 2.0°C,50% | 0 | 23 | 898028 | 1.0000 | | 0.03749342 | | 4.65 | 5.4 |
| 2.0°C,83% | 0 | 0 | 898051 | 1.0000 | 0.00 | 0.99999777 | 0.96 | 8.9 | 8.2 |
| 3.0°C,17% | 121 | 69083 | 828848 | 0.9229 | 0.08 | 0.00598964 | 0.99 | 2.09 | 5.2 |
| 3.0°C,50% | 0 | 0 | 898051 | 1.0000 | | 0.99189578 | | 6.65 | 7.8 |
| 3.0°C,83% | 0 | 0 | 898051 | 1.0000 | 0 | 1 | 0.01 | 11.88 | 11.8 |
| 4.0°C,17% | 0 | 2907 | 895144 | 0.9968 | 0.00 | 0.06400527 | 0.94 | 3.68 | 7.4 |
| 4.0°C,50% | 0 | 0 | 898051 | 1.0000 | | 1 | | 8.63 | 10.2 |
| 4.0°C,83% | 0 | 0 | 898051 | 1.0000 | 0 | 1 | 0 | 14.42 | 14.8 |
| 5.0°C,17% | 0 | 3031 | 895021 | 0.9966 | 0.00 | 0.12925212 | 0.87 | 5.05 | 8.8 |
| 5.0°C,50% | 0 | 0 | 898051 | 1.0000 | | 1 | | 10.55 | 12.2 |
| 5.0°C,83% | 0 | 0 | 898051 | 1.0000 | 0 | 1 | 0 | 17.2 | 17.8 |

Scenarios represented are the 1.5°C, 2.0°C, 3.0°C, 4.0°C, and 5.0°C warming to 2080-2100 against the 1850-1900 baseline, reporting the 17th, 50th and 83rd percentile projections for each scenario. The average rate of RSLR for coastlines, and rate of global SLR are represented for each scenario as mm yr$^{-1}$.

**Extended Data Table 2 | Location and timing of mangrove SET-MH measurements, with corresponding rate of RSLR and elevation gain**

| Region | Location, source | latitude, longitude | SET (n) | Start reading | End reading | RSLR** | Mean elevation change rate mm yr⁻¹(s.d.) |
|---|---|---|---|---|---|---|---|
| Florida, USA | Big Sable Creek[105] (Feher) | 25.21, 81.14 | 1 | 1999 | 2021 | 9.04 | 0.65 (0.00) |
| Florida, USA | Biscayne Bay[105] | 25.29, 80.37 | 2 | 2011 | 2020 | 7.48 | 1.38 (0.47) |
| Florida, USA | Lostmans River[105] | 25.53, 81.18 | 1 | 1999 | 2021 | 7.48 | 5.54 (0.00) |
| Florida, USA | NE Florida Bay[105] | 25.21, 80.42 | 12 | 1998 | 2019 | 1.72 | 2.43 (1.29) |
| Florida, USA | Rookery Bay[105] | 25.58, 80.94 | 2 | 2001 | 2006 | -1.86 | 0.46 (0.35) |
| Florida, USA | Rookery Bay[105] | 25.58, 80.94 | 2 | 1997 | 2006 | -1.84 | 12.78 (3.43) |
| Florida, USA | Rookery Bay[105] | 25.58, 80.94 | 13 | 2015 | 2021 | 4.68 | 2.00 (4.77) |
| Florida, USA | Rookery Bay[105] | 25.58, 80.94 | 1 | 1993 | 2017 | 1.31 | 2.48 (0.00) |
| Florida, USA | Rookery Bay[105] | 25.58, 80.94 | 1 | 1994 | 2007 | -2.28 | 9.04 (0.00) |
| Florida, USA | Rookery Bay[105] | 25.58, 80.94 | 2 | 1993 | 2017 | 1.31 | 1.00 (1.24) |
| Florida, USA | Shark River[105] | 25.32, 81.00 | 1 | 1998 | 2018 | 1.93 | 2.76 (0.00) |
| Florida, USA | Shark River[105] | 25.32, 81.00 | 2 | 1998 | 2021 | 2.19 | 2.34 (0.71) |
| Florida, USA | Ten Thousand Is.[105] | 25.85, 81.48 | 2 | 1998 | 2018 | 2.76 | -0.82 (0.43) |
| Florida, USA | Ten Thousand Is. [105] | 25.85, 81.48 | 1 | 2012 | 2020 | 4.71 | 3.08 (0.00) |
| Louisiana, USA | Port Fouchon[106] | 29.10, 80.37 | 3 | 2006 | 2011 | 3.62 | 4.13 (0.23) |
| Belize | Twin Cays[84] | 16.82, 88.10 | 3 | 2001 | 2004 | 2.0 | -0.23 (3.94) |
| Victoria, Australia | French Island* | -38.30, 145.42 | 3 | 2000 | 2022 | 2.74 | 0.94 (0.74) |
| Victoria, Australia | Quail Island* | -38.23, 145.30 | 3 | 2000 | 2019 | 3.0 | -2.59 (1.73) |
| Victoria, Australia | Rhyll* | -38.46, 145.28 | 3 | 2000 | 2019 | 3.0 | -0.71 (0.90) |
| Victoria, Australia | Kooweerup* | -38.22, 145.41 | 2 | 2000 | 2022 | 2.74 | 1.06 (1.52) |
| NSW, Australia | Minnamurra* | -34.62, 150.84 | 3 | 2001 | 2011 | 0.47 | 1.4 (0.30) |
| NSW, Australia | Cararma Inlet* | -34.98, 150.77 | 3 | 2001 | 2020 | 3.88 | 2.18 (1.18) |
| NSW, Australia | Currambene Creek* | -35.01, 150.66 | 6 | 2001 | 2020 | 2.82 | 0.50 (0.76) |
| NSW, Australia | Homebush Bay* | -33.84, 151.07 | 6 | 2000 | 2020 | 5.4 | 2.76 (0.58) |
| NSW, Australia | Kooragang Island* | -32.84, 151.72 | 6 | 2002 | 2016 | 3.11 | 3.11 (1.09) |
| NSW, Australia | Kooragang Island[107] | -32.86, 151.71 | 3 | 2005 | 2008 | 12.68 | 3.00 (0.00) |
| NSW, Australia | Tweed River* | -28.18, 153.54 | 3 | 2000 | 2018 | 4.9 | 1.81 (0.83) |
| NSW, Australia | Berowra* | -33.62, 151.12 | 3 | 2002 | 2017 | 5.44 | 2.46 (2.10) |
| Queensland, Australia | Moreton Bay East* | -276.28,153.04 | 9 | 2007 | 2011 | 17.09 | -11.31 (2.10) |
| Queensland, Australia | Moreton Bay West* | -27.43, 153.43 | 9 | 2007 | 2011 | 17.09 | -15.37 (1.60) |
| Queensland, Australia | Daintree River* | -16.27, 145.40 | 5 | 2015 | 2021 | 17.18 | 2.98 (4.10) |
| Queensland, Australia | Daintree River* | -16.31, 145.42 | 7 | 2014 | 2021 | 15.06 | 0.92 (1.92) |
| Western Australia | Exmouth Gulf* | -22.49, 114.32 | 6 | 2011 | 2018 | -6.54 | -0.14 (0.18) |
| Northern Territory | Darwin Harbour* | -12.48, 130.91 | 6 | 2014 | 2021 | 15.13 | 7.01 (6.93) |
| Northern Territory | Darwin Harbour* | ⅃, 130.96 | 33 | 2016 | 2021 | 8.04 | 2.18 (5.97) |
| Micronesia | Yela River[108] | 5.32, 162.93 | 3 | 1997 | 2004 | 29.23 | -0.97 (1.93) |
| Micronesia | Utwe River[108] | 5.28, 162.95 | 3 | 1997 | 2004 | 30.91 | 1.90 (1.50) |
| Micronesia | Pukusruk[108] | 5.35, 163.02 | 1 | 1999 | 2004 | -1.77 | -2.33 (0.00) |
| Micronesia | Enipoas[108] | 6.82, 158.21 | 3 | 1999 | 2004 | -1.74 | 1.58 (0.65) |
| Micronesia | Sapwalap[108] | 6.87, 158.30 | 3 | 1999 | 2004 | -1.74 | -0.74 (1.17) |
| New Zealand | Firth of Thames[109] | -37.21, 175.45 | 5 | 2008 | 2015 | 9.6 | -6.10 (3.31) |

*Original SET Data; **RSLR for the period of SET-MH measurements, from the nearest tide gauge.

Extended Data Table 3 | Identifiers and Variables used in the RF regression analysis (Data[29])

| | |
|---|---|
| Accretion | Rate of accretion above the feldspar horizon (mm yr$^{-1}$) |
| elevation.rate | Rate of elevation gain from the SET record (mm yr$^{-1}$) |
| Subsidence | accretion - elevation.rate |
| SLR50 | Local sea-level trend derived from nearest tide gauge: 0-50BP linear trend (mm yr$^{-1}$) |
| RSLR.contemporaneous | RSLR for each site for the period of SET measurement. Linear trend (mm yr$^{-1}$) |
| tidal.range | Difference between MHW and MLW (m) |
| marshElevation | Elevation of the SET-MH in relation to local datum (m) |
| elevCapital | Elevation of SET-MH in relation to modelled lowest marsh limits (cm) |
| elevDeficit | Elevation Deficit, defined as RSLR period of measure minus elevation rate. (mm yr$^{-1}$) |
| bulkDensity | Bulk density of the upper 10 cm (dry, g cm$^{-3}$) |
| Organic matter | Organic matter in the upper 10cm by weight (%) |
| Rainfall | Average annual rainfall (mm) |
| TSM.2011 | MERIS-derived total suspended matter -average |

# Reporting Summary

Please do not complete any field with "not applicable" or n/a.  Refer to the help text for what text to use if an item is not relevant to your study.
For final submission: please carefully check your responses for accuracy; you will not be able to make changes later.

## Statistics

For all statistical analyses, confirm that the following items are present in the figure legend, table legend, main text, or Methods section.

| n/a | Confirmed | |
|---|---|---|
| ☐ | ☑ | The exact sample size (*n*) for each experimental group/condition, given as a discrete number and unit of measurement |
| ☐ | ☑ | A statement on whether measurements were taken from distinct samples or whether the same sample was measured repeatedly |
| ☐ | ☑ | The statistical test(s) used AND whether they are one- or two-sided *Only common tests should be described solely by name; describe more complex techniques in the Methods section.* |
| ☐ | ☑ | A description of all covariates tested |
| ☐ | ☑ | A description of any assumptions or corrections, such as tests of normality and adjustment for multiple comparisons |
| ☐ | ☑ | A full description of the statistical parameters including central tendency (e.g. means) or other basic estimates (e.g. regression coefficient) AND variation (e.g. standard deviation) or associated estimates of uncertainty (e.g. confidence intervals) |
| ☐ | ☑ | For null hypothesis testing, the test statistic (e.g. *F*, *t*, *r*) with confidence intervals, effect sizes, degrees of freedom and *P* value noted *Give P values as exact values whenever suitable.* |
| ☐ | ☑ | For Bayesian analysis, information on the choice of priors and Markov chain Monte Carlo settings |
| ☐ | ☑ | For hierarchical and complex designs, identification of the appropriate level for tests and full reporting of outcomes |
| ☐ | ☑ | Estimates of effect sizes (e.g. Cohen's *d*, Pearson's *r*), indicating how they were calculated |

*Our web collection on statistics for biologists contains articles on many of the points above.*

## Software and code

Policy information about availability of computer code

| Data collection | N/A |
|---|---|
| Data analysis | Link to code provided |

For manuscripts utilizing custom algorithms or software that are central to the research but not yet described in published literature, software must be made available to editors and reviewers. We strongly encourage code deposition in a community repository (e.g. GitHub). See the Nature Portfolio guidelines for submitting code & software for further information.

## Data

Policy information about availability of data

All manuscripts must include a data availability statement. This statement should provide the following information, where applicable:
- Accession codes, unique identifiers, or web links for publicly available datasets
- A description of any restrictions on data availability
- For clinical datasets or third party data, please ensure that the statement adheres to our policy

| Data availability statement provided |
|---|

# Research involving human participants, their data, or biological material

Policy information about studies with [human participants or human data](). See also policy information about [sex, gender (identity/presentation), and sexual orientation]() and [race, ethnicity and racism]().

| | |
|---|---|
| Reporting on sex and gender | |
| Reporting on race, ethnicity, or other socially relevant groupings | |
| Population characteristics | |
| Recruitment | |
| Ethics oversight | |

Note that full information on the approval of the study protocol must also be provided in the manuscript.

# Field-specific reporting

Please select the one below that is the best fit for your research. If you are not sure, read the appropriate sections before making your selection.

☐ Life sciences ☐ Behavioural & social sciences ☑ Ecological, evolutionary & environmental sciences

For a reference copy of the document with all sections, see [nature.com/documents/nr-reporting-summary-flat.pdf]()

# Life sciences study design

All studies must disclose on these points even when the disclosure is negative.

| | |
|---|---|
| Sample size | |
| Data exclusions | |
| Replication | |
| Randomization | |
| Blinding | |

# Behavioural & social sciences study design

All studies must disclose on these points even when the disclosure is negative.

| | |
|---|---|
| Study description | |
| Research sample | |
| Sampling strategy | |
| Data collection | |
| Timing | |
| Data exclusions | |
| Non-participation | |
| Randomization | |

# Ecological, evolutionary & environmental sciences study design

All studies must disclose on these points even when the disclosure is negative.

| | |
|---|---|
| Study description | Yes |
| Research sample | Yes |
| Sampling strategy | Yes |
| Data collection | Yes |
| Timing and spatial scale | Yes |
| Data exclusions | None |
| Reproducibility | Yes |
| Randomization | N/A |
| Blinding | N/A |

Did the study involve field work? ☐ Yes ☑ No

## Field work, collection and transport

| | |
|---|---|
| Field conditions | N/A |
| Location | N/A |
| Access & import/export | N/A |
| Disturbance | N/A |

# Reporting for specific materials, systems and methods

We require information from authors about some types of materials, experimental systems and methods used in many studies. Here, indicate whether each material, system or method listed is relevant to your study. If you are not sure if a list item applies to your research, read the appropriate section before selecting a response.

## Materials & experimental systems

| n/a | Involved in the study |
|---|---|
| ☒ | ☐ Antibodies |
| ☒ | ☐ Eukaryotic cell lines |
| ☒ | ☐ Palaeontology and archaeology |
| ☒ | ☐ Animals and other organisms |
| ☒ | ☐ Clinical data |
| ☒ | ☐ Dual use research of concern |
| ☒ | ☐ Plants |

## Methods

| n/a | Involved in the study |
|---|---|
| ☒ | ☐ ChIP-seq |
| ☒ | ☐ Flow cytometry |
| ☒ | ☐ MRI-based neuroimaging |

## Antibodies

| | |
|---|---|
| Antibodies used | |
| Validation | |

# Eukaryotic cell lines

Policy information about cell lines and Sex and Gender in Research

| | |
|---|---|
| Cell line source(s) | |
| Authentication | |
| Mycoplasma contamination | |
| Commonly misidentified lines (See ICLAC register) | |

# Palaeontology and Archaeology

| | |
|---|---|
| Specimen provenance | |
| Specimen deposition | |
| Dating methods | |

☐ Tick this box to confirm that the raw and calibrated dates are available in the paper or in Supplementary Information.

| | |
|---|---|
| Ethics oversight | |

Note that full information on the approval of the study protocol must also be provided in the manuscript.

# Animals and other research organisms

Policy information about studies involving animals; ARRIVE guidelines recommended for reporting animal research, and Sex and Gender in Research

| | |
|---|---|
| Laboratory animals | |
| Wild animals | |
| Reporting on sex | |
| Field-collected samples | |
| Ethics oversight | |

Note that full information on the approval of the study protocol must also be provided in the manuscript.

# Clinical data

Policy information about clinical studies
All manuscripts should comply with the ICMJE guidelines for publication of clinical research and a completed CONSORT checklist must be included with all submissions.

| | |
|---|---|
| Clinical trial registration | |
| Study protocol | |
| Data collection | |
| Outcomes | |

# Dual use research of concern

Policy information about dual use research of concern

## Hazards

Could the accidental, deliberate or reckless misuse of agents or technologies generated in the work, or the application of information presented in the manuscript, pose a threat to:

| No | Yes | |
|----|-----|--|
| X | ☐ | Public health |
| X | ☐ | National security |
| x | ☐ | Crops and/or livestock |
| X | ☐ | Ecosystems |
| x | ☐ | Any other significant area |

## Experiments of concern

Does the work involve any of these experiments of concern:

| No | Yes | |
|----|-----|--|
| X | ☐ | Demonstrate how to render a vaccine ineffective |
| X | ☐ | Confer resistance to therapeutically useful antibiotics or antiviral agents |
| X | ☐ | Enhance the virulence of a pathogen or render a nonpathogen virulent |
| X | ☐ | Increase transmissibility of a pathogen |
| x | ☐ | Alter the host range of a pathogen |
| X | ☐ | Enable evasion of diagnostic/detection modalities |
| x | ☐ | Enable the weaponization of a biological agent or toxin |
| x | ☐ | Any other potentially harmful combination of experiments and agents |

# Plants

| Seed stocks | N/A |
|-------------|-----|
| Novel plant genotypes | N/A |
| Authentication | N/A |

# ChIP-seq

## Data deposition

☐ Confirm that both raw and final processed data have been deposited in a public database such as GEO.

☐ Confirm that you have deposited or provided access to graph files (e.g. BED files) for the called peaks.

| Data access links
*May remain private before publication.* | |
|---|---|
| Files in database submission | |
| Genome browser session
(e.g. UCSC) | |

## Methodology

| Replicates | |
|---|---|
| Sequencing depth | |
| Antibodies | |
| Peak calling parameters | |
| Data quality | |
| Software | |

# Flow Cytometry

## Plots

Confirm that:

☐ The axis labels state the marker and fluorochrome used (e.g. CD4-FITC).

☐ The axis scales are clearly visible. Include numbers along axes only for bottom left plot of group (a 'group' is an analysis of identical markers).

☐ All plots are contour plots with outliers or pseudocolor plots.

☐ A numerical value for number of cells or percentage (with statistics) is provided.

## Methodology

Sample preparation

Instrument

Software

Cell population abundance

Gating strategy

☐ Tick this box to confirm that a figure exemplifying the gating strategy is provided in the Supplementary Information.

# Magnetic resonance imaging

## Experimental design

Design type

Design specifications

Behavioral performance measures

Imaging type(s)

Field strength

Sequence & imaging parameters

Area of acquisition

Diffusion MRI          ☐ Used          ☐ Not used

## Preprocessing

Preprocessing software

Normalization

Normalization template

Noise and artifact removal

Volume censoring

## Statistical modeling & inference

Model type and settings

Effect(s) tested

Specify type of analysis:          ☐ Whole brain          ☐ ROI-based          ☐ Both

Statistic type for inference

(See Eklund et al. 2016)

Correction

## Models & analysis

| n/a | Involved in the study |
| --- | --- |
| ☐ | ☐ Functional and/or effective connectivity |
| ☐ | ☐ Graph analysis |
| ☐ | ☐ Multivariate modeling or predictive analysis |

Functional and/or effective connectivity

Graph analysis

Multivariate modeling and predictive analysis

*Neil Saintilan*

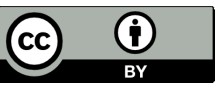

