## [Peer Review File · Nature]

Manuscript Title: Widespread retreat of coastal habitat likely at warming levels above 1.5°C

Reviewer Comments & Author Rebuttals

Reviewer Reports on the Initial Version:

Referees' comments:

Referee #1 (Remarks to the Author):

A Summary of the key results

This work reviews the threat posed by sea level rise (SLR) to a number of critical coastal ecosystems, with a comprehensive review of existing works and new models to predict likely impacts under various future scenarios

B Originality and significance: if not novel, please include reference

This is the first quantitative assessment of SLR threat run across multiple ecosystems, although some past studies have, with older data and different approaches, attempted estimates for individual ecosystems (including several of the current authors). It is an important work, serving to unify an often somewhat fractured and incomplete literature which has been dominated by partial and sometimes seemingly contradictory elements.

C Data & methodology: validity of approach, quality of data, quality of presentation

The research approach is solid. The input layers are probably the best available (but see below for mangroves), but the authors would do well to acknowledge some of these weaknesses a little more clearly. For example:

- the global map of saltmarsh, while the best global map available, is a mosaic map of very mixed quality with many gaps, a far less reliable product than the other ecosystem maps
- the coastal units of the Global Coastal Wetland Model relatively coarse summary units

It is also important that the authors give proper citations for their data layers, as instructed on the Ocean Data Viewer. This site is only serving other people's datasets, so it is not only a requirement to do this, but such information gives further detail of origins. If possible, in supplementary materials some review of the strengths and weaknesses might also be appropriate. On the Ocean Data Viewer there is also more than one layer for mangroves and so it is unclear which one was used (hopefully the latest?).

The modelling of inland migration appears to this reviewer to be overly pessimistic, although the details are hard to understand (it appears to be hidden in the Global Coastal Wetland Model). I think it is important to expand on the methods here, and it may be important to consider and express weaknesses. The best global population density maps that I am aware of will still not give the resolution to really impute the assumption that there is no space for landwards compensation. Populations are rarely evenly spread across a landscape, and 5 or 20 persons per km² could, in reality, be focused in a village with nothing nearby for several km. Especially if the population density data is simply spread by finescale administrative units or even night lights. I suspect there is considerably more room for migration than the maps imply.

The presentation of the information is generally excellent, however I feel that Figure 2 needs some improvement. For the caption greater clarity is needed. It would also help to give indication of

geographic scope of various elements and perhaps restrict text to cover only elements visible in the Figure, putting other observations in main text:

- it is not clear in 2a whether those elements that are “far from centres of glaciation (7.1 mm yr⁻¹) and Caribbean and North American locations (5.2 mm yr⁻¹)” are illustrated in some way?
- Could they indicate on the Figure that B and C are for UK and Gulf of Mex respectively?
- and likewise that E refers to the global subset of islands?

Other comments

Line69-70: currently seems to give significant precedence to fisheries values over all other, rework.

Line 138 - (illustrated in e,f). – should be d, e

Line 224-6 – feels like a somewhat throwaway comment and it would be useful to give a little more attention to the possibility of enhanced growth of coastal wetlands under high CO₂.

Lines 276,278 – further citations needed for reef degradation and disease

Line 438 – implication of extensive dieback along the coastline of half a continent – the paper refers to a very large event, but perhaps not that large

D Appropriate use of statistics and treatment of uncertainties

See earlier comments about this reviewer’s perception of uncertainty, notably in the landwards migration projections.

E Conclusions: robustness, validity, reliability

In the conclusions the paragraph from Line 486 is weak - the authors don't need to write about generalities. They hint at a significant risk, namely that we invest in blue carbon at our peril if these systems are unlikely to survive projected warming scenarios. They need to discuss this a little further: what will happen to that carbon if the ecosystems are lost or migrate (will soil carbon be stable or released?); Should blue carbon investments aim to target only the ecosystems and locations of greatest projected stability? Could blue carbon also strengthen a powerful argument for managed retreat and allowing landwards expansion of these ecosystems.

F Suggested improvements: experiments, data for possible revision

Need clarification on source data used for ecosystems (notably mangroves) and population density. There may not be any need for revision, but it is possible that improved methods might be able to strengthen the inland migration component.

G References: appropriate credit to previous work?

See earlier comment – the habitat maps are not properly referenced and need to be. Otherwise the article is comprehensively referenced.

H Clarity and context: lucidity of abstract/summary, appropriateness of abstract, introduction and conclusions

This is a well-written and highly important piece of work. I look forward to seeing it published and will be citing it when it is!

Referee #2 (Remarks to the Author):

In this manuscript, the authors set out to determine critical rates of relative sea level rise, above which it can be expected that coastal ecosystems (marshes, mangroves, coral reefs) can no longer

vertically adjust and consequently will collapse. They conclude, based on Holocene paleorecords and spatial information, that a critical rate is around 7 mm.y⁻¹, which would be reached when global temperatures would increase by more than 2 degrees. This paper forms an element in an ongoing scientific debate about the extent of resilience of coastal ecosystems against sea level rise.

The data and literature compilation by the authors leaves little doubt, if that would have existed, that the spatial extent of coastal ecosystems decreases in periods of rapid sea level rise, while they expand under moderate to low rates. In fact, this manuscript makes use of previous studies and compilations to make that point. Its originality is in bringing together results concerning different types of coastal ecosystems and trying to find generalities across those ecosystem types. However, while the authors set out to determine critical rates as tipping points beyond which a catastrophic collapse (words used in the manuscript) occurs, they fail to argue why tipping point dynamics would dominate the occurrence of coastal ecosystems at global scale. One can easily imagine that at a local scale (one m²) an on/off situation exists. Either the ecosystem is capable of realizing sufficient accretion to persist, or not. In the latter case the vegetation or reef cannot survive and a switch to an unvegetated (or coral-free) state occurs. These dynamics have a clear tipping point, and are further characterized by some positive feedbacks that also lead to alternative stable states (vegetated/unvegetated). However, at system scale the dynamics are much less straightforward. Where accretion is dominated by external sediment sources, which is the case in many marshes and mangroves, a given net influx of sediment will allow a smaller area of the coastal system to follow sea level rise upon an increase of the SLR rate, but it is unlikely that all sediment flux would suddenly remain unused. It is also unlikely that a fixed maximum rate of vertical accretion would exist - physiologically the plants may have the capacity to survive at very high accretion rates, but systemwide the flux of sediment will limit the rate. One therefore expects a narrower, faster landward shifting and younger marsh or mangrove system to persist, rather than a collapse of the system beyond a tipping point.

In purely autogenic systems, e.g. coral reefs depending on biological calcium carbonate production, a similar mechanism could occur if spatial redistribution of sediment is possible. In general, therefore, I would not expect that globally an on/off switch between presence and absence of coastal ecosystems ("collapse") will occur. Rather, I expect gradual changes that may be nonlinear but not catastrophic. The authors of this manuscript clearly take another point of view. I can, of course, be convinced but then they should provide arguments for their view.

This conceptual point is important because it determines how one looks at the paleorecord, which is the main source of information for this study. In periods when coastal ecosystems become narrower and moreover shift position faster because the shoreline moves faster, one will observe many places that switch from vegetated to unvegetated. In contrast, when sea level rise is slow, sufficient accretion may be realizable for the ecosystems to expand seaward (while also following the shoreline landward) and many places will see a switch from unvegetated to vegetated. This change in the frequency of positive and negative shifts therefore does contain information on the relative expansion/contraction of the spatial extent of the systems, but is not necessarily an indicator of their collapse. One moreover expects strong regional differences in the rate where expansion shifts into contraction, as it is highly dependent on the strength of external sediment inflow. This aspect too makes the existence of a single global tipping point SLR unlikely. However, when starting from the catastrophic concept, point observations are interpreted as local realisations of the on/off transition

and thus indicators of the likelihood of collapse. That is a fundamentally different interpretation.

With respect to the derivation of the main conclusions and data presentations in the figures 1 and 2, the manuscript heavily relies on other published or submitted studies. While I realize that it is impossible at this level to repeat all the methodological aspects of all these studies, the current state of the manuscript is deficient in the methods description. The derivation of the colour bars in Fig. 1, as an example, is so vague that it almost looks like hand waving. I am sure the authors can improve on this aspect. Likewise, the interpretation of the figures in Fig. 2 is very difficult without looking up all underlying papers. It is sometimes unclear what the data are (what is meant by 'density' in Fig. 2a for instance). It is never specified which stratigraphic records were used, where they are positioned with respect to the current shoreline and current ecosystem extent, and how the different records have been combined (or not?) into reconstructions of the landscape. The latter could include an answer to the question whether the coastal ecosystems were squeezed or catastrophically eliminated from the landscape. The discussion on the role of shallow subsidence in relation to increased accretion is another example of vague reasoning leading to strong conclusions. It is stated that with increased accretion also shallow subsidence will increase, which to a certain extent will be true. And thus, it is concluded, "For this reason there is a less than 10 percent probability of a contemporary marsh keeping pace with RSLR greater than 7 mm yr⁻¹ (Refs 72,73)". I don't know how many steps in the reasoning are skipped here, but I fail to see the logic.

The manuscript spends quite some attention to the societal aspects of carbon mitigation etc. While I do not want to dispute the importance of those implications, I wonder whether it is worth spending so much space on what essentially repeats the IPCC message. I would rather see more expansion on the question what would be lost if the coastal ecosystems are lost: what are the most essential ecosystem services, which feedback loops could operate and make things worse, and similar aspects.

In summary, the persistence of coastal ecosystems under climate change is a question of great importance, about which some controversy exists and an analysis of all available data is timely and needed. All this are arguments in favour of publishing this paper. However, I think the current analysis depends too much on the unproven and (in my opinion) unlikely assumption that coastal ecosystems are prone to collapse when sea level rises too fast. This unduly constraints the results of the analysis. It provides for a spectacular title (ecosystem collapse likely!) but is hardly substantiated in the paper. I think the whole concept of collapse can only be used in the paper and the title if there is firm evidence that this is how the dynamics will indeed work out. For now, I am not convinced and I would much prefer an analysis that tries to quantify at what rate coastal ecosystem extent and quality is lost, depending on sea level rise rate.

Referee #3 (Remarks to the Author):

This paper presents an alarming, highly pessimistic outlook for global coastal ecosystems under scenarios of warming and sea level rise, by combining palaeo, contemporary, and modelling evidence. Analysis was focused on the potential response of three major types of coastal ecosystems to sea level rise: tidal marshes, mangrove forests, and coral reefs (as well as reef islands). I recognize that this manuscript deals with a topic of broad interest and that the authors' efforts have merits. My biggest concerns are that the conclusions are overly broad, pessimistic, and not firmly supported by the evidence presented, and that the perspective largely neglected the great potential of conservation, in particular restoration, to help reverse the tides of coastal ecosystem collapse under climate change, although such potential has been highlighted in recent Nature publications (e.g., Schuerch et al. 2018, Duarte et al. 2020). I detail my major and minor concerns below.

1) Important coastal ecosystems are neglected. The authors discussed only four of the many different types of coastal ecosystems. But why they chose these four types of coastal ecosystems (tidal marshes, mangrove forests, coral reefs, and reef islands) but not others is unclear. Seagrass beds, kelp forests, tidal flats, sandy beaches, oyster reefs, and rocky shores are all broadly distributed coastal ecosystems and may respond to sea level rise and warming in very different ways. Concluding that global coastal ecosystems are very likely to collapse without considering the vast variety of other coastal ecosystems is clearly irrational.

2) A related point - The value of including coral reefs in this paper needs to be reconsidered. Although I agree that coral reefs and coastal wetlands (tidal marshes and mangrove forests) share certain similarities in their response to sea level rise (e.g., vertical adjustment as the authors highlighted) and other climate and non-climate stressors, there are more differences than similarities. I don't think it's worth highlighting the relatively few similarities while neglecting the vast differences between these systems in the same manuscript, which could lead to confusing or misleading conclusions. If coral reefs (and reef islands) have to be included, I think, at a minimum, the authors need to better discuss their similarities and differences.

3) Weak evidence for lateral responses of coastal wetlands to SLR. The evidence presented is mostly about vertical responses, rather than lateral responses, landward and seaward. But reaching the conclusion of coastal ecosystem collapse needs strong evidence for both. The authors did present evidence for the lateral response of reef islands to sea level rise, but the evidence is relatively weak and could not be used to specify a RSLR rate at which reef island contraction is highly likely as the authors noted. They also presented modelling evidence for the lateral responses of tidal marshes and mangrove forests to projected sea level rise, using the Global Coastal Wetland Model. But these modelling analyses considered only the current conditions for the existence of barrier to landward migration (i.e., two of the four scenarios used in a previous Nature paper; Schuerch et al. 2018) and neglected the other two scenarios where nature-based adaptation (i.e., restoration) has been shown to be of great potential to help mitigate coastal wetland loss under future sea level rise. Why their analysis is still novel given this previous, more complete publication is also unclear.

4) The relative and changing roles of sea level rise vs other drivers of coastal ecosystem change were overlooked. I agree that sea level rise was, is and will likely still be a key driver of coastal ecosystem

change. But not just the role of sea level rise change over time. Other factors may be even more important, at least in certain periods of time. I would say that more tidal marshes and mangrove forests have been lost to reclamation than sea level rise over the past decades or centuries. I would also say that seawater warming and related-bleaching events have driven and will likely drive more coral loss compared to sea level rise. The authors did address that these other factors may affect the capacity of coastal ecosystems to respond to sea level rise at multiple places. But the point here is that the role of some other factors may play an even more important role than sea level rise. This needs to be acknowledged in the manuscript and has key implications for how future dynamics of coastal ecosystems should be predicted and managed. For example, expansive coastal wetlands have been lost to reclamation in the past several decades, even hundreds of years, meaning that enhancing the capacity of coastal wetlands to adapt to rising sea levels would require substantial restoration. Focusing on only sea level rise and overlooking the roles of other stressors could thus be misleading.

5) The maps of coastal ecosystems used are not all complete. Although some types of coastal ecosystem (such as mangrove forests) have been well mapped, others have not. This is a critical problem to the global map of tidal marshes available on the UNEP Ocean Data Viewer, which the authors used in their analysis. Large areas of tidal marshes in Canada, Northern Russia, and other regions remain unmapped (Mcowen et al. 2017). And what have been mapped may also suffer issues such as inconsistent time periods and mapping precision. Analysis based on these maps could lead to incomplete, even erroneous conclusions. The results need to be presented and interpreted with caveats.

6) The manuscript title is a strong exaggeration, for the concerns I explained above. It is also highly pessimistic despite multiple previous publications in Nature highlighting the potential capacity of humanity to rebuild coastal ecosystems (Schuerch et al. 2018, Duarte et al. 2020). I am not opposed to making pessimistic perspectives where necessary and well demonstrated, but a Research Article supported by stronger, more rounded, more focused data and analysis may be needed. Even if such pessimistic perspectives are supported by strong data and analysis, I would say it's better to address the actions needed to ensure a positive outcome.

7) The Introduction section didn't set up the knowledge gap and the need of this research in a reasonable manner. There have been numerous studies on the responses of various coastal ecosystems to sea level rise. But what are the key knowledge gaps? What are the major uncertainties in current understanding in this field? What theories is this paper aimed to challenge or advance? The current Introduction reads like "sea level is rising faster, which affects coastal ecosystems important for people ..." I think this is too plain for a high impact journal like Nature.

8) I also feel that the writing of this manuscript is highly technical. If this manuscript is to be published at Nature, I think it needs to be written in a more comprehensible way for the general readership.

Specific comments:

L43-44, this sentence does not flow well.

L48-49, I don't think this conclusion could be made without considering restoration and other factors

as I detailed in my main comments.

L69, those listed here are not all coastal ecosystems. Somewhere early in the manuscript the authors should address why these four systems, but not others, are chosen for study and how the insights for these systems may or may not apply to other types of coastal ecosystems.

L75-81, The manuscript focuses on sea level rise, but other stressors are addressed even before the issue of sea level rise is specified...

L97-100, These details might better be placed somewhere else in the manuscript instead of the introductory paragraphs that should be used mainly to set up the key scope of this paper (as detailed in one of my main comments).

L114, When to use coastal "ecosystem" or coastal "habitat"?

L138, there is no panel f in this figure. Detailed descriptions for panels d and e are also lacking.

L167, the rate of RSLR does not really match that given in Fig. 1b. These data in the main text need to be checked against the figure.

L172-209, the logical connections of these three paragraphs describing overlapping types of ecosystems and time periods are unclear.

L238, Fig. 2a, lower case

L240-241, this sentence describing marsh response to RSLR cites Fig. 2A that shows mangrove data.

L242, this sentence about palaeo-stratigraphic assessments cites Fig. 2D, which doesn't seem to be palaeo-stratigraphic data. The data and data source are not fully explained or specified in the figure legend.

Fig. 2a, what's mangrove "initiation"? Unclear. And what's sustained mangrove "accretion"? Soil accretion or sediment accretion? Mangrove cannot accrete, per se.

L259, a comma is needed after "RSLR (d)"

L261, In Fig. 2e, the value is 6.2 mm yr⁻¹, not 7 mm yr⁻¹.

L262, the right parenthesis is missing.

L263, how was this threshold of 3% chosen?

L275-278, Placing these points here in the middle of highlighting sea level rise is a distraction.

L284-294, given that the manuscript has a specific section that discusses future projections, these should be cut from this section on contemporary dynamics. There are also similar issues in other places of this section (e.g. L321-325).

L320, moderates?

L361, this is the first place where reef islands are defined (though only partly), although reef islands have been discussed since very early in the Introduction.

L384, > 4 mm yr⁻¹? "(c)" should be added after 3.0°C.

L385-387, I would appreciate more details about this issue and others on the projection of RSLR in the Methods section, but they are completely absent.

L400, I think the data given in this Table 1 are redundant and largely overlap with Fig. 3e-g. Both 1.0 and 1.00 are used in this Table. Any difference in precision?

L417-418, Some supporting analysis and data are needed to support this statement.

L443-446, As detailed in one of my main concerns, these projections neglected the great potential of restoration. But expansive coastal wetlands have been lost to reclamation, requiring restoration.

L466, Are there any marine, but not coastal, ecosystems discussed in this section? Unclear why the authors used "marine and coastal ecosystems" in the section heading.

L448-464, This appears to be the only place where how a 5.0°C warming can affect coastal ecosystems is discussed. This seems intended to address, if not exaggerate, the negative impacts of

warming. I think it's better to discuss the responses of coral reefs to sea level rise using warming scenarios in a way consistent with those used for the other coastal ecosystems, even the impacts might not be as negative. This doesn't mean that climate warming is not a strong force for coral reef dynamics. But, as detailed in one of my main comments, warming-associated stressors other than sea level rise may play a more critical role.

L486-500, This paragraph ignores the fact that large areas of coastal wetlands have been lost to reclamation. So, restoration is not just a means to enhance carbon sequestration as a nature-based solution to climate change, but should be recognized as one of the most important strategies to enhance the resilience of coastal wetlands to climate change.

L516-518, I don't think this concluding point is sufficiently discussed in the manuscript. The current discussions on this point are mostly about coral reefs, not tidal marshes and mangrove forests.

L1000 and 1008, saltmarshes are used here, but in most places of the main text, tidal marshes are used. Be consistent with those terms. This also applies to mangrove forests and mangroves.

Referee #4 (Remarks to the Author):

A. Summary of the key results

This paper describes four stages of analysis. In the first stage a comprehensive review of the literature and further investigation is undertaken focusing on how coastal habitat has responded to relative sea level rise since the Last Glacial Maximum. In the second stage, the authors carry out a similar analysis and review but focus on changes in mangroves, tidal marshes and coral reefs using cotemporary observations. In these two stages of work the authors identify critical thresholds of 4 and 7 mm/yr for which eventual loss of mangroves and tidal marshes is likely or very likely; and 7 mm/yr for reef island contraction or instability. The third stage of analysis centres around future projections of relative sea level rise; the authors compare relative sea level rise projections out to the year 2100 with distributions of coastal habitats around the world. They determine the proportion of mangroves, tidal marshes and coral reef habitat where relative sea level rise exceeds 4 mm/yr and 7 mm/yr. The key result is that nearly all the world's mangrove forests and coral reefs and 40% of tidal marshes will be subject to relative sea level rise of greater than 7 mm/yr by 2100 under a global mean warming scenario of 3°C. In the fourth and final stage of analysis the authors model the potential for landward compensation of wetlands under relative sea level rise projections.

B. Originality and significance: if not novel, please include reference

This work is very original and significant. Previous studies on coastal ecosystem response have tended to be local/regional in focus. The novelty of this work is the global focus, and the assessment of both past and potential future changes. Another novelty is that the authors focus on all four main types of coastal ecosystems (i.e., mangroves, tidal marshes, coral reefs and reef islands) whereas previous studies have tended to focus on just mangroves and tidal marshes, or coral reefs. I believe the results will be of wide interest to scientist and practitioners across many different disciplines. The results stress the importance of limiting temperature to less than 3 degrees C by 2100.

C. Data & methodology: validity of approach, quality of data, quality of presentation

In my opinion the methodology is sound. Each stage logically follows on from the next. A large number of previous studies are reviewed. The figures are of good quality and nicely illustrate the key

results. I don't believe the manuscript has any major flaws which should prohibit its publication? However, I have some moderate concerns as follows.

First, when reading the paper there is a discontent between the literature review sections and future assessment. It is not immediately clear to me where exactly the 4 and 7 mm/yr thresholds exactly come from (on Line 360) for tidal marshes and 7 mm/yr for coral reefs. I think the authors need to explain in more detail, maybe summarising the literature review and analysis from the first two stages, describing in a paragraph exactly how they derived these two thresholds. Furthermore, the first two review/analysis sections are well written and formulated. But I wonder if it would help the reader if the review information was also summarised in a single table; which listed the period, rate of sea level rise, geographic region, associated supporting references and supporting dataset, etc. This would allow the reader to see all the review material summarised in one place.

Second, a significant concern with sea level rise is that it will extend many hundreds to thousands of years beyond 2100, even if carbon emissions are stabilised. The authors mention this briefly on line 521, but I think a slightly more detailed discussion of this should be made.

Thirdly, tidal marshes and mangroves, in particular provide a nature buffer to flood defences. As these are lost, massive investment in upgrading/improving flood defences will be required. It would be good if the authors included a short discussion of this somewhere, as I think it is relevant in the context of this paper.

Fourth, in several countries tidal marshes are being re-introduced, for example via managed realignment of the coast (<https://www.omreg.net/query-database/>). In the grand scheme of things, this is very small – but I think should at least be discussed briefly somewhere.

Fifth, I found the methodology for how the authors calculated landward wetland expansions a bit difficult to follow. I think this could be explained but, and in particularly the assumptions made, could be more clearly described. For example, it appears this approach is simply a bathy-up type filling of existing topography, and doesn't account for erosion etc.

. Appropriate use of statistics and treatment of uncertainties

I think the statistics and treatment of uncertainty is mostly handled well in this paper. However, in regard to the summary of the palaeo records, the authors should add a brief discussion of the uncertainty in rates of sea-level rise (shown in Figure 1b) as estimated by the GIA modelling approach. On pages 14 I think it would be useful to include length of coastline affect in km (e.g., line 368) along with percentage (e.g., >11%).

A. Conclusions: robustness, validity, reliability

I believe the conclusions to be robust, valid and reliable following directly from the analysis.

B. Suggested improvements: experiments, data for possible revision

I have suggested some things that could be improved in response to question C above. I have also listed some minor corrected suggestions below.

D. References: appropriate credit to previous work?

The authors refer to a comprehensive list of references.

E. Clarity and context: lucidity of abstract/summary, appropriateness of abstract, introduction and conclusions

Overall, I think the paper is very well written. The abstract and conclusions very nicely summarise the key headline finding. The introduction is well written. Small point, but the introduction could possibly be expanded slightly on the importance of coastal habitat, including for example the significance of some of these ecosystems to tourism, recreational diving, etc. Their importance goes well beyond fishing and carbon storage.

Minor Comments

Online 38 you list three things (tidal wetlands, coral reefs and reef islands) but on line 69 you list four things (mangroves, tidal marshes, coral reefs and reef islands). Minor point but I would make consistent on both lines and use the same term tidal wetland or tidal marshes, but not both.

Line 68-71: Possibly could also mention importance of some of these ecosystems to tourism, recreational diving, etc. Their importance goes well beyond fishing and carbon storage.

On lines 86 to 88 the authors state 'A further doubling in the rate of GMSL rise (to >7 mm yr⁻¹) is expected by mid-century under high emissions scenarios, and before the end of the century under mid- range emissions scenarios¹¹'. This is a bit vague. It would be useful if the authors could provide more specific details here and maybe directly refer to some of the specific scenarios and rates listed in Table 9.9 in Fox-Kemper et al. (2021).

Line 108. Maybe in brackets state the time period over which you focus on for the palaeo-record and observations; Make it explicitly clear for the reader.

Line 115: I would say 'mean sea-level' position rather than just 'sea-level' position.

Line 122: Abbreviate to GMSL, as already defined on line 83;

Lines 122 – 129. It might be useful to state the average rates of mean-sea-level rise in mm/yr for the three periods you specify to put the 7 mm/yr into direct context.

Page 5, Figure 1b: I would suggest labelling 'meltwater pulse 1A' on the figure.

Line 146-149: In brackets refer to Figure 1b and 1c here, to help guide the reader.

Line 152: Do you mean 40 mm yr⁻¹ rather than 4 mm yr⁻¹ during meltwater pulse 1A?

Line 158: 7.5 mm/yr seems very precise; would it be better to say ~ 7 mm/yr.

Line 284, abbreviate to SLR

Line 374: Add a space between 4 and mm/yr.

Small point, there is inconsistency throughout in whether you use FIG or Fig.

Line 379 should be 7 mm /yr. Missing the per year after mm.

Quality of Figure 4 is poor, and the text size of the colour labels is too small.

Line 460: you say 'water-level' increases; elsewhere in the paper you say 'sea-level'. Small point, but I would change to 'sea-level' for consistency.

Author Rebuttals to Initial Comments:

Referees' comments:

Referee #1 (Remarks to the Author):

C Data & methodology: validity of approach, quality of data, quality of presentation

The research approach is solid. The input layers are probably the best available (but see below for mangroves), but the authors would do well to acknowledge some of these weaknesses a little more clearly. For example:

- the global map of saltmarsh, while the best global map available, is a mosaic map of very mixed quality with many gaps, a far less reliable product than the other ecosystem maps - the coastal units of the Global Coastal Wetland Model relatively coarse summary units It is also important that the authors give proper citations for their data layers, as instructed on the Ocean Data Viewer. This site is only serving other people's datasets, so it is not only a requirement to do this, but such information gives further detail of origins. If possible, in supplementary materials some review of the strengths and weaknesses might also be appropriate. On the Ocean Data Viewer there is also more than one layer for mangroves and so it is unclear which one was used (hopefully the latest?).

The modelling of inland migration appears to this reviewer to be overly pessimistic, although the details are hard to understand (it appears to be hidden in the Global Coastal Wetland Model). I think it is important to expand on the methods here, and it may be important to consider and express weaknesses. The best global population density maps that I am aware of will still not give the resolution to really impute the assumption that there is no space for landwards compensation. Populations are rarely evenly spread across a landscape, and 5 or 20 persons per km² could, in reality, be focused in a village with nothing nearby for several km. Especially if the population density data is simply spread by finescale administrative units or even night lights. I suspect there is considerably more room for migration than the maps imply.

Response

Yes, the latest available mangrove layer was used. We have investigated the only rival dataset (Murray et al. 2022) and included the lead author (Nicholas Murray) in this revised submission. Our decision was to retain the Ocean Data Viewer for the extent analysis, but have expanded the methods description to explain the datasets that were used and why. We now include in Figure 4 the most optimistic of the scenarios modelled by Schuerch et al.

2018, which allows for migration purely based on topography, removing filters relating to human population density.

The presentation of the information is generally excellent, however I feel that Figure 2 needs some improvement. For the caption greater clarity is needed. It would also help to give indication of geographic scope of various elements and perhaps restrict text to cover only elements visible in the Figure, putting other observations in main text:

- it is not clear in 2a whether those elements that are “far from centres of glaciation (7.1 mm yr⁻¹) and Caribbean and North American locations (5.2 mm yr⁻¹)” are illustrated in some way?

- Could they indicate on the Figure that B and C are for UK and Gulf of Mex respectively? - and likewise that E refers to the global subset of islands?

Response

We have redrawn Figure 2, incorporating additional data but also simplifying the figure so that only cdf probabilities are shown (the histograms classifying responses as retreat/advance/stable are now shown in supplementary).

Other comments

Line69-70: currently seems to give significant precedence to fisheries values over all other, rework.

Response

This section has been rewritten

Line 138 - (illustrated in e,f). – should be d, e

Response

Revised and corrected

Line 224-6 – feels like a somewhat throwaway comment and it would be useful to give a little more attention to the possibility of enhanced growth of coastal wetlands under high CO₂.

Response

We now devote a paragraph to this issue (lines 306-312). We cite the papers from the Smithsonian’s global change wetland experiment which show enhanced growth under elevated CO₂ (e.g. Langley et al 2009). Interestingly, since our original submission, Zhu et al (2022) have shown that this enhancement is negated when RLSR reached 7mm yr⁻¹.

Lines 276,278 – further citations needed for reef degradation and disease

Response

Lines removed in rewrite

Line 438 – implication of extensive dieback along the coastline of half a continent – the paper refers to a very large event, but perhaps not that large

Response

This sentence has been deleted

E Conclusions: robustness, validity, reliability

In the conclusions the paragraph from Line 486 is weak - the authors don't need to write about generalities. They hint at a significant risk, namely that we invest in blue carbon at our peril if these systems are unlikely to survive projected warming scenarios. They need to discuss this a little further: what will happen to that carbon if the ecosystems are lost or migrate (will soil carbon be stable or released?); Should blue carbon investments aim to target only the ecosystems and locations of greatest projected stability? Could blue carbon also strengthen a powerful argument for managed retreat and allowing landwards expansion of these ecosystems.

Response

We now take up these issues in lines 362-368 of the Discussion.

Referee #2 (Remarks to the Author):

However, while the authors set out to determine critical rates as tipping points beyond which a catastrophic collapse (words used in the manuscript) occurs, they fail to argue why tipping point dynamics would dominate the occurrence of coastal ecosystems at global scale. One can easily imagine that at a local scale (one m²) an on/off situation exists. Either the ecosystem is capable of realizing sufficient accretion to persist, or not. In the latter case the vegetation or reef cannot survive and a switch to an unvegetated (or coral-free) state occurs. These dynamics have a clear tipping point, and are further characterized by some positive feedbacks that also lead to alternative stable states (vegetated/unvegetated). However, at system scale the dynamics are much less straightforward. Where accretion is dominated by external sediment sources, which is the case in many marshes and mangroves, a given net

influx of sediment will allow a smaller area of the coastal system to follow sea level rise upon an increase of the SLR rate, but it is unlikely that all sediment flux would suddenly remain unused. It is also unlikely that a fixed maximum rate of vertical accretion would exist - physiologically the plants may have the capacity to survive at very high accretion rates, but system-wide the flux of sediment will limit the rate. One therefore expects a narrower, faster landward shifting and younger marsh or mangrove system to persist, rather than a collapse of the system beyond a tipping point. In purely autogenic systems, e.g. coral reefs depending on biological calcium carbonate production, a similar mechanism could occur if spatial redistribution of sediment is possible. In general, therefore, I would not expect that globally an on/off switch between presence and absence of coastal ecosystems ("collapse") will occur. Rather, I expect gradual changes that may be nonlinear but not catastrophic. The authors of this manuscript clearly take another point of view. I can, of course, be convinced but then they should provide arguments for their view.

This conceptual point is important because it determines how one looks at the paleorecord, which is the main source of information for this study. In periods when coastal ecosystems become narrower and moreover shift position faster because the shoreline moves faster, one will observe many places that switch from vegetated to unvegetated. In contrast, when sea level rise is slow, sufficient accretion may be realizable for the ecosystems to expand seaward (while also following the shoreline landward) and many places will see a switch from unvegetated to vegetated. This change in the frequency of positive and negative shifts therefore does contain information on the relative expansion/contraction of the spatial extent of the systems, but is not necessarily an indicator of their collapse. One moreover expects strong regional differences in the rate where expansion shifts into contraction, as it is highly dependent on the strength of external sediment inflow. This aspect too makes the existence of a single global tipping point SLR unlikely. However, when starting from the catastrophic concept, point observations are interpreted as local realisations of the on/off transition and thus indicators of the likelihood of collapse. That is a fundamentally different interpretation.

With respect to the derivation of the main conclusions and data presentations in the figures 1 and 2, the manuscript heavily relies on other published or submitted studies. While I realize that it is impossible at this level to repeat all the methodological aspects of all these studies, the current state of the manuscript is deficient in the methods description. The derivation of the colour bars in Fig. 1, as an example, is so vague that it almost looks like hand waving. I am sure the authors can improve on this aspect. Likewise, the interpretation of the figures in Fig. 2 is very difficult without looking up all underlying papers. It is sometimes unclear what the data are (what is meant by 'density' in Fig. 2a for instance). It is never specified

which stratigraphic records were used, where they are positioned with respect to the current shoreline and current ecosystem extent, and how the different records have been combined (or not?) into reconstructions of the landscape. The latter could include an answer to the question whether the coastal ecosystems were squeezed or catastrophically eliminated from the landscape. The discussion on the role of shallow subsidence in relation to increased accretion is another example of vague reasoning leading to strong conclusions. It is stated that with increased accretion also shallow subsidence will increase, which to a certain extent will be true. And thus, it is concluded, "For this reason there is a less than 10 percent probability of a contemporary marsh keeping pace with RSLR greater than 7 mm yr⁻¹ (Refs 72,73)". I don't know how many steps in the reasoning are skipped here, but I fail to see the logic.

The manuscript spends quite some attention to the societal aspects of carbon mitigation etc. While I do not want to dispute the importance of those implications, I wonder whether it is worth spending so much space on what essentially repeats the IPCC message. I would rather see more expansion on the question what would be lost if the coastal ecosystems are lost: what are the most essential ecosystem services, which feedback loops could operate and make things worse, and similar aspects.

In summary, the persistence of coastal ecosystems under climate change is a question of great importance, about which some controversy exists and an analysis of all available data is timely and needed. All this are arguments in favour of publishing this paper. However, I think the current analysis depends too much on the unproven and (in my opinion) unlikely assumption that coastal ecosystems are prone to collapse when sea level rises too fast. This unduly constraints the results of the analysis. It provides for a spectacular title (ecosystem collapse likely!) but is hardly substantiated in the paper. I think the whole concept of collapse can only be used in the paper and the title if there is firm evidence that this is how the dynamics will indeed work out. For now, I am not convinced and I would much prefer an analysis that tries to quantify at what rate coastal ecosystem extent and quality is lost, depending on sea level rise rate.

Response

We agree entirely with the main thrust of these comments. We have in the revised version avoided as far as possible the suggestion of "tipping points", we remove the term "collapse" and cast the analysis and discussion entirely within a probabilistic framework (rather than "on/off" at various RSLR rates). We don't propose, for example, an upper limit of RSLR to

which wetlands may adjust. Rather, based on our data, we show that the probability of vertical adjustment decreases with RSLR, and that this decrease is broadly consistent between types of data analysed and across habitat types. We provide more detail in our methods concerning the sources of paleo data and the nature of the SET-MH observations. The tidal marsh SET-MH analysis is now published as Saintilan et al. (2022)- this paper was under review at the time of our previous submission. We add to this an analysis of a global mangrove SET-MH dataset, combining our own data with those recently published as Feher et al. (2022), to the same effect. We also note in our methods the range of settings covered by these and the paleo data: all tide ranges, organic and minerogenic marshes, and the commonly encountered bio-geomorphic settings. The use of the Murray et al (2022) data in relation to the tidal marsh SET-MH analysis provides independent verification of increased vulnerability at high RSLR. While there are significant gaps in coverage (the Arctic coast, much of Africa and South America) the species and geomorphic settings are largely encountered within our reference set of 664 surface elevation tables and 1116 paleo marsh reference points.

All these data suggest a low probability of vertical adjustment under higher RSLR (we use the nomenclature of the IPCC in referring to “unlikely” ($p < 0.34$) and “very unlikely” ($p < 0.1$) adjustment). Under high RSLR, we expect therefore that most wetlands will respond precisely as you describe: “one therefore expects a narrower, faster landward shifting and younger marsh or mangrove system to persist”. While we do not propose local extirpation (except in heavily developed landscapes), this still suggests significant disruption including the loss of most existing wetland under higher RSLR, redistribution at a global scale and, as you say, a change to the nature of the habitat and associated ecosystem services. We model the probable loss of existing habitat in relation to sea-level rise pointing to important existing refugia (e.g. high latitude tidal marshes) and emerging refugia (mid Holocene platforms in Australia, etc). We avoid revisiting Schuerch et al 2018 because our emphasis is on the fate of existing habitat.

The methods section of the paper has been greatly expanded, and includes detail on the SET-MH method, what stratigraphic cores were used in the analysis, and what global datasets were used in each component of the analysis.

Referee #3 (Remarks to the Author):

This paper presents an alarming, highly pessimistic outlook for global coastal ecosystems under scenarios of warming and sea level rise, by combining palaeo, contemporary, and modelling evidence. Analysis was focused on the potential response of three major types of coastal ecosystems to sea level rise: tidal marshes, mangrove forests, and coral reefs (as well

as reef islands). I recognize that this manuscript deals with a topic of broad interest and that the authors' efforts have merits. My biggest concerns are that the conclusions are overly broad, pessimistic, and not firmly supported by the evidence presented, and that the perspective largely neglected the great potential of conservation, in particular restoration, to help reverse the tides of coastal ecosystem collapse under climate change, although such potential has been highlighted in recent Nature publications (e.g., Schuerch et al. 2018, Duarte et al. 2020). I detail my major and minor concerns below.

1) Important coastal ecosystems are neglected. The authors discussed only four of the many different types of coastal ecosystems. But why they chose these four types of coastal ecosystems (tidal marshes, mangrove forests, coral reefs, and reef islands) but not others is unclear. Seagrass beds, kelp forests, tidal flats, sandy beaches, oyster reefs, and rocky shores are all broadly distributed coastal ecosystems and may respond to sea level rise and warming in very different ways. Concluding that global coastal ecosystems are very likely to collapse without considering the vast variety of other coastal ecosystems is clearly irrational.

Response

We make clear now our rationale for inclusion of ecosystem types: “We select habitats which show both a high sensitivity to sea-level and biogenic feedbacks that facilitate resilience: mangroves, tidal marshes, and coral reef islands.” We therefore exclude habitats which are either abiotic (sandy beaches, rocky shores), and those for which sea-level rise is likely to be a less dominant driver (kelp forests, seagrass beds). While coral reef islands are composed of abiotic material, the living reef controls both sediment supply and wave attenuation. The comparison with coastal wetland responses to sea-level rise is interesting and novel, and we have retained our discussion of this habitat.

2) A related point - The value of including coral reefs in this paper needs to be reconsidered. Although I agree that coral reefs and coastal wetlands (tidal marshes and mangrove forests) share certain similarities in their response to sea level rise (e.g., vertical adjustment as the authors highlighted) and other climate and non-climate stressors, there are more differences than similarities. I don't think it's worth highlighting the relatively few similarities while neglecting the vast differences between these systems in the same manuscript, which could lead to confusing or misleading conclusions. If coral reefs (and reef islands) have to be included, I think, at a minimum, the authors need to better discuss their similarities and differences.

Response

We agree that the vulnerabilities of coral reefs to both climate change and sea-level rise are fundamentally different to the other systems being considered. We now discuss these only in the context of our consideration of coral islands. This change has abbreviated and improved the focus of the paper.

3) Weak evidence for lateral responses of coastal wetlands to SLR. The evidence presented is mostly about vertical responses, rather than lateral responses, landward and seaward. But reaching the conclusion of coastal ecosystem collapse needs strong evidence for both. The authors did present evidence for the lateral response of reef islands to sea level rise, but the evidence is relatively weak and could not be used to specify a RSLR rate at which reef island contraction is highly likely as the authors noted. They also presented modelling evidence for the lateral responses of tidal marshes and mangrove forests to projected sea level rise, using the Global Coastal Wetland Model. But these modelling analyses considered only the current conditions for the existence of barrier to landward migration (i.e., two of the four scenarios used in a previous Nature paper; Schuerch et al. 2018) and neglected the other two scenarios where nature-based adaptation (i.e., restoration) has been shown to be of great potential to help mitigate coastal wetland loss under future sea level rise. Why their analysis is still novel given this previous, more complete publication is also unclear.

Response

Since the original submission Nicholas Murray and colleagues have published their global database on losses and gains in tidal wetlands (Murray et al. 2022, *Science* 376). Dr Murray has provided these data for the sites considered in our tidal marsh SET analysis (a global analysis in relation to RSLR was precluded by the extent of anthropogenic modification, as well as the potential for error in high intertidal environments). Because we have good control on vertical response and RSLR history at these sites, the Murray dataset provided an independent assessment of vulnerability to lateral change relating to elevation deficit in tidal marshes. This provides a “third line of evidence” strengthening our contention that tidal marshes are vulnerable to RSLR likely to be encountered globally in coming decades under mid and high emissions scenarios.

The landward accommodation model uses the same approach as Schuerch et al. 2018 but is re-parameterised using the results of our analyses. Our results therefore differ to those of Schuerch et al 2018 and are based on a more complete (data-driven) understanding of marsh and mangrove responses under higher rates of RSLR. We also use the more recent sea-level rise modelling prepared by co-authors Garner and Kopp for the Sixth Assessment report of the IPCC. We now present the most optimistic of the scenarios Schuerch et al. 2018,

removing the effect of population density altogether. The same point as in our original submission remains. For most of the world, the maximum opportunities for lateral expansion represent a small proportion of existing area due to topographic constraints.

4) The relative and changing roles of sea level rise vs other drivers of coastal ecosystem change were overlooked. I agree that sea level rise was, is and will likely still be a key driver of coastal ecosystem change. But not just the role of sea level rise change over time. Other factors may be even more important, at least in certain periods of time. I would say that more tidal marshes and mangrove forests have been lost to reclamation than sea level rise over the past decades or centuries. I would also say that seawater warming and related-bleaching events have driven and will likely drive more coral loss compared to sea level rise. The authors did address that these other factors may affect the capacity of coastal ecosystems to respond to sea level rise at multiple places. But the point here is that the role of some other factors may play an even more important role than sea level rise. This needs to be acknowledged in the manuscript and has key implications for how future dynamics of coastal ecosystems should be predicted and managed. For example, expansive coastal wetlands have been lost to reclamation in the past several decades, even hundreds of years, meaning that enhancing the capacity of coastal wetlands to adapt to rising sea levels would require substantial restoration. Focusing on only sea level rise and overlooking the roles of other stressors could thus be misleading.

Response

This is a good point and we have sought to emphasise this more fully in our discussion (eg. lines 389-390). However, the reversal of these immediate stressors does not diminish the likely dominance and ubiquity of sea-level rise as a driver of change in the long-term and on the whole decrease resilience to this key driver, as outlined in our new section “Potential CO₂ and Temperature effects” lines 305-325.

5) The maps of coastal ecosystems used are not all complete. Although some types of coastal ecosystem (such as mangrove forests) have been well mapped, others have not. This is a critical problem to the global map of tidal marshes available on the UNEP Ocean Data Viewer, which the authors used in their analysis. Large areas of tidal marshes in Canada, Northern Russia, and other regions remain unmapped (McOwen et al. 2017). And what have been mapped may also suffer issues such as inconsistent time periods and mapping precision. Analysis based on these maps could lead to incomplete, even erroneous conclusions. The results need to be presented and interpreted with caveats.

Response.

Unfortunately, these limitations bely all datasets- including the recently published work of Murray et al (2022) which does not attempt to identify tidal wetland north of 60 degrees latitude. This is a problem for our estimates of the proportion of tidal marsh resilient to projected sea-level rise because large areas of coastline in these latitudes are subject to isostatic adjustment. The appropriate caveats are now included (lines 709-710).

6) The manuscript title is a strong exaggeration, for the concerns I explained above. It is also highly pessimistic despite multiple previous publications in Nature highlighting the potential capacity of humanity to rebuild coastal ecosystems (Schuerch et al. 2018, Duarte et al. 2020). I am not opposed to making pessimistic perspectives where necessary and well demonstrated, but a Research Article supported by stronger, more rounded, more focused data and analysis may be needed. Even if such pessimistic perspectives are supported by strong data and analysis, I would say it's better to address the actions needed to ensure a positive outcome.

Response

We are sympathetic to the concern expressed here. In our rewrite, we have deliberately sought to emphasise the positive story: that existing tidal wetlands can largely survive under a low emissions future, and that in several regions opportunities exist for large-scale landward migration. These positives are now taken up in our title “A safe operating space for coastal ecosystems”.

We present the most comprehensive dataset on coastal ecosystem responses to sea-level rise yet published. These include the entire global SET-MH monitoring record, compared to the results of the two most comprehensive assessments of paleo marsh and mangrove responses to RSLR (Horton et al. 2018 and Saintilan et al 2020). In each case the probability of sustained vertical adjustment to RSLR is nearly identical, an observation which drives our global vulnerability modelling. Few of these data were available at the time Schuerch et al. 2018 was written. The potential of high emissions pathways to lead to massive disruption to coastal ecosystems cannot be ignored.

7) The Introduction section didn't set up the knowledge gap and the need of this research in a reasonable manner. There have been numerous studies on the responses of various coastal ecosystems to sea level rise. But what are the key knowledge gaps? What are the major uncertainties in current understanding in this field? What theories is this paper aimed to challenge or advance? The current Introduction reads like “sea level is rising faster, which

affects coastal ecosystems important for people ...” I think this is too plain for a high impact journal like Nature.

Response

We accept this criticism and have completely rewritten the introduction. We think the relevance of the paper and the associated knowledge gap is now more clearly established from the outset.

8) I also feel that the writing of this manuscript is highly technical. If this manuscript is to be published at Nature, I think it needs to be written in a more comprehensible way for the general readership.

Response

We have rewritten the paper with this advice in mind.

Specific comments:

L43-44, this sentence does not flow well.

Response

The sentence has been rewritten.

L48-49, I don't think this conclusion could be made without considering restoration and other factors as I detailed in my main comments.

Response

The sentence has been deleted and replaced with a greater emphasis on restoration opportunities under low emissions scenarios

L69, those listed here are not all coastal ecosystems. Somewhere early in the manuscript the authors should address why these four systems, but not others, are chosen for study and how the insights for these systems may or may not apply to other types of coastal ecosystems. Response

This is now addressed in lines 102-105 of the new introductory section

L75-81, The manuscript focuses on sea level rise, but other stressors are addressed even before the issue of sea level rise is specified...

Response

The issue of sea-level rise is introduced in the second paragraph in the newly written introduction

L97-100, These details might better be placed somewhere else in the manuscript instead of the introductory paragraphs that should be used mainly to set up the key scope of this paper (as detailed in one of my main comments).

Response

The introduction has been re-written to more clearly and directly establish the scope of the paper

L114, When to use coastal “ecosystem” or coastal “habitat”?

Response

We have revised the manuscript reserving the use of the word “habitat” to specific reference of ecosystem spatial extent.

L138, there is no panel f in this figure. Detailed descriptions for panels d and e are also lacking.

Response

The figure has been extensively revised, resolving this issue

L167, the rate of RSLR does not really match that given in Fig. 1b. These data in the main text need to be checked against the figure.

Response

This has been checked and corrected.

L172-209, the logical connections of these three paragraphs describing overlapping types of ecosystems and time periods are unclear.

Response

This section has been re-arranged with the addition of introductory and concluding text to both flag the logical direction and summarise the findings.

L238, Fig. 2a, lower case

Response

Corrected

L240-241, this sentence describing marsh response to RSLR cites Fig. 2A that shows mangrove data.

Response

Figure 2 has been extensively revised and we have checked referencing of the figure in the text.

L242, this sentence about palaeo-stratigraphic assessments cites Fig. 2D, which doesn't seem to be palaeo-stratigraphic data. The data and data source are not fully explained or specified in the figure legend.

Response

We have revised the figure to include palaeo-stratigraphic data (panels d-e), SET-MH data (panels a-c). These, and their sources, are described more fully now in the figure legend and the supplementary materials.

Fig. 2a, what's mangrove "initiation"? Unclear. And what's sustained mangrove "accretion"? Soil accretion or sediment accretion? Mangrove cannot accrete, per se.

Response

We now use the phrase *the probability of the initiation of sustained mangrove vertical peat development* which we hope is a succinct but detailed explanation. More detail is provided in the Methods section.

L259, a comma is needed after "RSLR (d)"

Response

Corrected

L261, In Fig. 2e, the value is 6.2 mm yr⁻¹, not 7 mm yr⁻¹.

Response

Relevant text removed

L262, the right parenthesis is missing.

Response

Revised figure caption has removed this sentence

L263, how was this threshold of 3% chosen?

Response

We are following here the last major review of island planiform change, Duvat et al 2019. This is now made clear.

L275-278, Placing these points here in the middle of highlighting sea level rise is a distraction.

Response

This was removed from the section in the restructuring of the manuscript

L284-294, given that the manuscript has a specific section that discusses future projections, these should be cut from this section on contemporary dynamics. There are also similar issues in other places of this section (e.g. L321-325).

Response

Sections on contemporary dynamics and projected changes are now clearly distinguished

L320, moderateS?

Response

Corrected

L361, this is the first place where reef islands are defined (though only partly), although reef islands have been discussed since very early in the Introduction.

Response

Reef islands are now defined in the opening sentences

L384, > 4 mm yr⁻¹? “(c)” should be added after 3.0°C.

Response

Corrected- many thanks

L385-387, I would appreciate more details about this issue and others on the projection of RSLR in the Methods section, but they are completely absent.

Response

We have provided lines 817-828 to describe the RSLR modelling

L400, I think the data given in this Table 1 are redundant and largely overlap with Fig. 3e-g. Both 1.0 and 1.00 are used in this Table. Any difference in precision?

Response

The table presents the results of the same analysis, but it would not be possible to derive the detail of Table 1 from Figure 3 (the intent of which is to provide a sense of geographic variation), and these details are likely to be important in various types of reporting (IPCC, etc). We have restructured Table 1 to facilitate comparisons between warming scenarios.

L417-418, Some supporting analysis and data are needed to support this statement.

Response

This text has been removed

L443-446, As detailed in one of my main concerns, these projections neglected the great potential of restoration. But expansive coastal wetlands have been lost to reclamation, requiring restoration.

Response

We have made two changes to our presentation of landward migration opportunities in Figure 4. First, we have included the most optimistic scenario by which migration is permitted even in higher population density locations. Note that previous tidal wetland and mangrove, if restored, are subject to the RLSR constraints under mid- and high-emissions scenarios as existing wetland. We now refer specifically to opportunities for landward migration and re-establishment in many regions of the globe (Northern Australia, Central America).

L466, Are there any marine, but not coastal, ecosystems discussed in this section? Unclear why the authors used “marine and coastal ecosystems” in the section heading.

Response

Reference to marine ecosystems has been removed

L448-464, This appears to be the only place where how a 5.0°C warming can affect coastal ecosystems is discussed. This seems intended to address, if not exaggerate, the negative impacts of warming. I think it’s better to discuss the responses of coral reefs to sea level rise using warming scenarios in a way consistent with those used for the other coastal ecosystems, even the impacts might not be as negative. This doesn’t mean that climate warming is not a strong force for coral reef dynamics. But, as detailed in one of my main comments, warming-associated stressors other than sea level rise may play a more critical role.

Response

This section has been removed in the revised manuscript.

L486-500, This paragraph ignores the fact that large areas of coastal wetlands have been lost to reclamation. So, restoration is not just a means to enhance carbon sequestration as a nature-based solution to climate change, but should be recognized as one of the most important strategies to enhance the resilience of coastal wetlands to climate change.

Response

We have restructured the manuscript and some of this material was moved to the introduction. We agree that restoration is an important strategy in enhancing resilience, and make specific reference to this in the penultimate sentence.

L516-518, I don't think this concluding point is sufficiently discussed in the manuscript. The current discussions on this point are mostly about coral reefs, not tidal marshes and mangrove forests.

Response

The discussion has been rewritten resolving this issue

L1000 and 1008, saltmarshes are used here, but in most places of the main text, tidal marshes are used. Be consistent with those terms. This also applies to mangrove forests and mangroves.

Response

We now use tidal marsh throughout

Referee #4 (Remarks to the Author):

First, when reading the paper there is a discontent between the literature review sections and future assessment. It is not immediately clear to me where exactly the 4 and 7 mm/yr thresholds exactly come from (on Line 360) for tidal marshes and 7 mm/yr for coral reefs. I think the authors need to explain in more detail, maybe summarising the literature review and analysis from the first two stages, describing in a paragraph exactly how they derived these two thresholds. Furthermore, the first two review/analysis sections are well written and formulated. But I wonder if it would help the reader if the review information was also summarised in a single table; which listed the period, rate of sea level rise, geographic region, associated supporting references and supporting dataset, etc. This would allow the reader to see all the review material summarised in one place.

Response

We now conclude each section with a concise statement of the implications for subsequent sections of the paper.

Second, a significant concern with sea level rise is that it will extend many hundreds to thousands of years beyond 2100, even if carbon emissions are stabilised. The authors

mention this briefly on line 521, but I think a slightly more detailed discussion of this should be made.

Response

This is central to our argument: that once rates of RSLR are reached they will likely be sustained for sufficiently long to exhaust any elevation capital. We have emphasised this in lines 334-341.

Thirdly, tidal marshes and mangroves, in particular provide a nature buffer to flood defences. As these are lost, massive investment in upgrading/improving flood defences will be required. It would be good if the authors included a short discussion of this somewhere, as I think it is relevant in the context of this paper.

Response

We now describe ecosystem service implications of transitioning from stable wetlands to migrating wetlands in the opening lines of the paper and mentioned briefly in the concluding sentences (line 390).

Fourth, in several countries tidal marshes are being re-introduced, for example via managed realignment of the coast (<https://www.omreg.net/query-database/>). In the grand scheme of things, this is very small – but I think should at least be discussed briefly somewhere.

Response

We have sought to emphasise more fully wetland restoration and the ongoing opportunities and importance of conservation and restoration. This is now covered in the introduction (lines 74-83) and the concluding paragraphs

Fifth, I found the methodology for how the authors calculated landward wetland expansions a bit difficult to follow. I think this could be explained but, and in particularly the assumptions made, could be more clearly described. For example, it appears this approach is simply a bathy-up type filling of existing topography, and doesn't account for erosion etc.

Response

We have expanded on our description of the methods used in the landward translation

D. Appropriate use of statistics and treatment of uncertainties

I think the statistics and treatment of uncertainty is mostly handled well in this paper. However, in regard to the summary of the palaeo records, the authors should add a brief discussion of the uncertainty in rates of sea-level rise (shown in Figure 1b) as estimated by

the GIA modelling approach. On pages 14 I think it would be useful to include length of coastline affect in km (e.g., line 368) along with percentage (e.g., >11%).

Response

A paragraph has been written in the Methods concerning uncertainty in the GIA modelling contributing to Fig 1B. Rather than quantifying coastline length exposed to various rates of RSLR, we have presented the area of habitat (e.g. Table 1).

Small point, but the introduction could possibly be expanded slightly on the importance of coastal habitat, including for example the significance of some of these ecosystems to tourism, recreational diving, etc. Their importance goes well beyond fishing and carbon storage.

Response

We have added content on ecosystem services both to the introduction and the discussion

Minor Comments

Online 38 you list three things (tidal wetlands, coral reefs and reef islands) but on line 69 you list four things (mangroves, tidal marshes, coral reefs and reef islands). Minor point but I would make consistent on both lines and use the same term tidal wetland or tidal marshes, but not both.

Response

We have revised this for consistency, listing mangrove, tidal marshes and reef islands throughout.

Line 68-71: Possibly could also mention importance of some of these ecosystems to tourism, recreational diving, etc. Their importance goes well beyond fishing and carbon storage. Response

In the interests of brevity we now refer to the “livelihoods and wellbeing of millions of people..”

On lines 86 to 88 the authors state ‘A further doubling in the rate of GMSL rise (to >7 mm yr-1) is expected by mid-century under high emissions scenarios, and before the end of the century under mid- range emissions scenarios¹¹’. This is a bit vague. It would be useful if the authors could provide more specific details here and maybe directly refer to some of the

specific scenarios and rates listed in Table 9.9 in Fox-Kemper et al. (2021).

Response

This section has been removed

Line 108. Maybe in brackets state the time period over which you focus on for the palaeo-record and observations; Make it explicitly clear for the reader.

Response

We have added “(after 17000 years ago, and particularly in the past 10000 years).”

Line 115: I would say ‘mean sea-level’ position rather than just ‘sea-level’ position.

Response

Corrected

Line 122: Abbreviate to GMSL, as already defined on line 83;

Response

Done

Lines 122 – 129. It might be useful to state the average rates of mean-sea-level rise in mm/yr for the three periods you specify to put the 7 mm/yr into direct context.

Response

These are represented in Fig 1

Page 5, Figure 1b: I would suggest labelling ‘meltwater pulse 1A’ on the figure.

Response

Done

Line 146-149: In brackets refer to Figure 1b and 1c here, to help guide the reader.

Response

Done

Line 152: Do you mean 40 mm yr⁻¹ rather than 4 mm yr⁻¹ during meltwater pulse 1A?

Response

Line deleted

Line 158: 7.5 mm/yr seems very precise; would it be better to say ~7 mm/yr. Response

Done

Line 284, abbreviate to SLR

Response

Done

Line 374: Add a space between 4 and mm/yr.

Response

Done

Small point, there is inconsistency throughout in whether you use FIG or Fig.

Response

Changed for consistency

Line 379 should be 7 mm /yr. Missing the per year after mm.

Response

Corrected

Quality of Figure 4 is poor, and the text size of the colour labels is too small.

Response

Fig 4 now consists of two panels, placing the remaining scenarios in supplementary

Line 460: you say 'water-level' increases; elsewhere in the paper you say 'sea-level'. Small point, but I would change to 'sea-level' for consistency.

Response

Removed

Reviewer Reports on the First Revision:

Referees' comments:

Referee #1:

Remarks to the Author:

Great paper, well done.

Here are some comments

Ln188 - missing comma

195-197 - i think sentence needs some "and" and comma alterations?

224 - Was this only done with tidal marsh? or also mangroves - the methods are a bit unclear

298 - I'm a little worried casual readers will read this as "we're likely to lose 81% of mangroves even if we achieve stability at 1.5 degrees" - could you write "...vulnerable to elevation deficit and eventual loss"

344 - I don't think this paper (your ref 13) suggests that blue carbon delivered the 5 ppmv decline? It may have played a (very) small part in that decline

351 - This sentence is about mangrove AND TIDAL MARSH unless you're suggestion mangroves in Siberia!

Figure 4 - could the lines in the legend be thickened to make them more clearly readable?

378 - Northern expansion of tidal marsh will also presumably be enabled by reduced ice-cover and ice scour along many coasts

392 - perhaps beyond the role of a reviewer but I think it would tie in with previous comments to add a third element to this sentence and mention managed retreat and allowing/encouraging the formation of new coastal wetlands landwards of current systems?

Table S2 - final column, is this a "rate" (annual) or total during the period of readings.

797 - I think you may not even be allowed to use this data without using the citations provided at this website, but even if you are it is poor practise not to acknowledge the research properly. The Ocean Data Viewer is a data library. Some readers may also want to know which version of which maps from that site you may have taken (and which year for the mangroves). I have just checked and you can click on the relevant layers and use the citations provided. Please also take this as a wider message for using this data in other publications. WCMC did not produce it and do not own it.

802 - I hadn't come across this, its interesting, but would certainly benefit from a better citation, though you may need to construct it. Looks like it was built at FIU?

807 - In title and section I am not sure if you only look at tidal marsh or also mangroves - tighten language throughout section, and if you didn't look and mangroves in this spell it out.

815 - need to cite the GSW dataset (Ref 53)

821 - is "marsh" tidal marsh or coastal wetlands

Fig 5 - poor resolution in my version so all maps are almost identical. But I THINK bottom right map should be identical to figure 4a, but doesn't look like it is? Maybe some work to clarify these maps

and description

Table S1 - I am finding this very confusing and it would benefit from a clearer explanation in the text.

Referee #2:

Remarks to the Author:

The authors have considered all referee remarks very seriously, and the present version of the manuscript is substantially improved.

In response to your explicit question, I do not see any problem with the statistical methods or with the way in which statistical results are reported.

The reworking and the replies to the comments have helped me understand that my first reading of the manuscript was incomplete and partially wrong. Yet, I think some additional changes are needed in order to prevent misunderstanding by the readers.

My misunderstanding was caused by not properly distinguishing between the terms 'retreat' and 'expansion', as used in the description of the paleo-based studies, or 'loss' in the description of the remote sensing studies, and changes in the areal extent of marshes and mangroves. In the paleo studies, e.g. of the marshes in the UK, the authors use the term 'retreat' as indication of landward lateral movement of the marsh system. Evidence for retreat can come from marsh sites turning into tidal flats, but also from terrestrial sites turning into marsh. (I think it is a pity that the two transitions are pooled into one indicator and not distinguished in the figures, as I would have been interested in how much evidence for the two types of retreat actually existed). 'Expansion' is used for evidence of seaward lateral migration, either by marsh turning into terrestrial vegetation (which is areal loss) or tidal flats turning into marsh (areal gain). Most importantly, neither 'retreat' nor 'expansion' say anything about the areal extent of the marsh. Also when interpreting remote sensing images, the results express information on changes in existing marshes. When the images show marshland being turned into water, this is areal loss at the site where marshes are presently occurring, but this loss may or may not be compensated by areal gain further inland. Information on the tendency for lateral movement of the habitats may be translated into likely changes of areal extent of the habitats, when the accommodation space landward of the present marshes is taken into consideration. The authors do this, but only at the end of the manuscript and without properly introducing the importance of adding this part of the logic.

The wording of the manuscript, also in the present version, adds to the confusion. The term 'expansion' in itself is not well chosen, as it suggests an increase of areal extent while this is not meant in the technical sense. Other examples of confused wording are: line 169: "potential for increased extent"; line 170: "vertical adjustment and habitat extent are greatly reduced"; line 224: "changes in the extent of tidal marsh" where changes in extent at the present location of occurrence is meant, not total extent of the marsh; line 233: "proportion of tidal marsh conversion" where again proportion of existing tidal marsh is meant; line 262: "eventual loss of mangroves and tidal marshes is likely" – loss of existing marsh at the present location is meant; line 283: "The probability of reaching RSLR at which loss is very likely" – I think that the evidence shows that landward migration is very likely, not necessarily loss as in loss of overall areal extent; but note at line 293 "with the greatest projected losses of existing habitat" is the correct formulation.

In practice, as is also shown by the authors, the difference between loss of habitat at the existing

location and loss overall is probably small – there are too many places where landward migration is impossible. However, for a proper understanding of the link between the evidence and the conclusions it is very important to keep the concepts clear. I would like the authors to carefully consider the wording, and to add a paragraph in the beginning of the paper where the different terms (retreat, seaward progradation, landward migration possibilities, loss or gain of areal extent) are introduced and the logic of the paper (first consider where and when landward migration is observed, then translate it into likely areal loss by also considering the possibility of encroaching current terrestrial habitat) is explained. At least for readers like myself, this would greatly help in understanding the paper.

A few minor comments:

Line 46: “destabilise”

Line 69: “The presence of marine vegetation and fringing reefs attenuate wave energy, protecting coastlines, while providing habitat to distinctive assemblages of species.” Sentence not properly structured. “presence” is subject but the verb is plural. ‘Presence’ may be left out.

Line 129: “Several periods of rapid global mean sea-level (GMSL) rise... have drowned mangrove forests and tidal marshes” Strange that periods can drown marshes

Lines 161-164: this information on how areal extent can also be lost from the landward side is very important. It could be used to re-iterate the relation between ‘expansion’ and areal extent

Line 203: “The results were consistent with those for tidal marshes.” This is not entirely true. The data shown in the supplementary material for mangrove SET sites do not show any relation at all between RSLR and rate of accretion, in contrast to what is described in the preceding paragraph for marshes.

Line 377: “Tidal marsh habitat is likely to expand in northern Siberia under higher RSLR due to a lack of topographic and human development impediments”. Here, if I understand it well, ‘expansion’ of the marsh in landward direction is meant, which is very inconsistent with the earlier use of the word (expansion = seaward migration).

Referee #3:

Remarks to the Author:

I appreciate the authors’ efforts in addressing most of all my (and the other reviewers’) comments in their manuscript. I consider that the revised manuscript provides timely findings that are now better supported by their analyses. Nonetheless, I think there is still space for improvement.

1) In response to one of my comments on the initial version of this manuscript, the authors included new analyses on how the habitat extent of tidal marshes responds to current sea level rise. The results show that the probability of tidal marsh contraction and increased water cover increase with increasing rate of RSLR (Fig. S3a, b). But it’s unclear how these results are in line with their choice of 7mm yr⁻¹ RSLR as a threshold rate above which eventual loss of tidal marshes is very likely or likely. The probability of tidal marsh contraction appears to change with RSLR by only a small extent, and even when RSLR is greater than 7mm yr⁻¹, probability of contraction remains smaller than 0.6. And unlike those shown in Fig. 2, all the responses shown in Fig. S3 are more like gradual processes, rather than threshold processes.

To predict the dynamics of coastal ecosystems under sea level rise, it’s key to understand their

extent changes (to a lesser extent, elevation changes). Elevation deficit does not necessarily go in parallel with marsh extent decline. If extent responses given in Fig. S3 are used as the main basis to define the threshold rate of RSLR and to project coastal ecosystem responses to future sea level rise, would that lead to major changes in the authors' findings and conclusions?

It's also unclear if the analysis given in Fig. S3a captures the responses of the full extent of tidal marshes to RSLR. Or it's primarily about the responses of marsh seaward edge? And how was the influence of barriers for marsh landward migration accounted for? To fully understand the results of this analysis, these points need to be clarified.

2) The manuscript was still poorly written in multiple sections. For example, although the manuscript has a section on past sea level rise, the paleo analyses in Fig. 2d-f are described only in the section on current sea level rise.

This issue is also obvious in the section "Opportunities for restoration and migration". Restoration and migration issues are discussed in the middle two paragraphs of this section, while "safe operating safe" is the key topic of the last paragraph. Transition between the two parts is abrupt, with no connections.

The "Methods" section also suffers from similar issues. I suggest the authors reorganize all the methods according to the sections (and their order) in the main text. Some figures and tables are inserted within the text, while others are put together at the end. Be consistent! There are also not a few conciseness and clarity issues.

3) The accuracy of some statements still needs to be checked. For example, L230, RSLR was given as 3.5 mm yr⁻¹, but in Fig. S3, the data value is shown as 2.3 mm yr⁻¹. L265, here Fig. 2 is cited to refer to a value of RSLR 7 mm yr⁻¹, but maybe they meant Fig. S3c (but the number given in Fig. S3c is 6.2 mm yr⁻¹).

Minor comments:

L47, I would say "mapped" tidal marshes

L114, "Stability" is a term used in multiple places in this paper, but I don't think it is analyzed per se. I would use expansion instead here.

L118, Unclear what they mean by "refugia". Refugia is not discussed anywhere else in the manuscript.

L154-156, this appears repetitive of what has been described in L144-146.

L161-162, Typically, extension of deltas will lead to expansion of intertidal areas, not reduction.

L167, RSLR rates in the instrumental period are not aforementioned.

Fig. 1 Sites in panel b and those in panel c do not match.

L183-184, rewrite

L201, I think the Bayesian analysis needs to be briefly described.

L206-208, this paleo analysis needs to be moved to the section on "past sea level rise"

L226, Fig. S3?

L291, The projection for 3°C warming scenario should be described before making the indication on tipping points.

L351, but the projections in Fig. 4 show minimal opportunities for mangrove landward expansion.

Figure 4, by "retreat", the authors meant "migration"?

L343-369, Those two paragraphs heavily focus on mangroves. More should be given to tidal marshes.

L732, clarify "512x260"

L736-750, these are not really Methods and appear repetitive of what has been described in main text.

Figure S1, The right part of this figure might better illustrate how tidal gauges measure relative sea level rise, instead of how relative sea level rise affects various marsh ecological processes.

Table S2, no data source was given for the several sites in Florida, USA

L903, delete "of"

L927, 5°C is missing?

L1021, platform?

Referee #6:

Remarks to the Author:

A safe operating space for coastal ecosystems

I am a new reviewer to this paper and hence this is the first time I have commented on it. I have seen the previous reviewer's comments and authors responses and a clean version of the revised paper.

Overall, I find the approach interesting and thought-provoking about the global response of selected coastal ecosystems – mangroves, tidal marshes and reef islands to sea-level rise and climate change. It is good to see multiple lines of evidence concerning the vertical response of wetlands considered and analysed -- palaeo-environmental data since the Last Glacial Maximum, the data from the Surface Elevation Table-Marker Horizon (SET-MH) network, and contemporary coastal ecosystems response to relative sea-level rise in terms of conversion to open water. The paper is less strong on integrating the potential for horizontal translation which could – at least in theory – offset vertical loss. This was a key conclusion the paper by Schuerch et al (2018) in Nature – the same model is used in this paper and I am unclear why there is a different outcome.

Hence, I am left wondering does the evidence presented support the conclusion of the paper that a 2 degree world is fundamentally different from a three degree world for coastal ecosystems which is what the conclusion of 2 degrees being a safe operating space and 3 degrees not being a safe operating space required? I can equally interpret from their results a world where coastal ecosystems are declining with today's sea-level rise (below a 2 degree world) and they will decline more rapidly if sea-level rise accelerates and other climate changes. However, the horizontal response can offset these losses and is 2 degrees different from 3 degrees in the area response – combined vertical and horizontal response? The horizontal response is often conditioned by how human beings behave in general and adapt in particular which was what Schuerch et al (2018) argued.

Looking at Rockstrom (2009), a safe operating space identifies and quantifies planetary boundaries that must not be transgressed to help prevent human activities from causing unacceptable environmental change. Does this paper really show that 2 degrees is qualitatively different to 3 degrees. In the vertical response maybe – but even there I am not entirely convinced. If we look at the vertical response in Figure 2, I consider the median – which I interpret as a 50:50 chance of a deficit – above the median we would seem to be in a regime of net loss – at least in a one-dimensional view of the problem. The median is associated with a RSLR of 1 to 3 mm/yr and given that global sea-level rise is now above 3 mm/yr, we are already into the territory of global decline in wetlands. In Figure 1, the big response in the palaeo-record (with exceptions) would seem to be the

decline below 3 mm/yr about 8,000 years ago? Lastly Figure S3(b) again shows the median linked to rates below 3 mm/yr. I am interested in the authors response to my interpretation.

The horizontal response is considered via the wetland inland migration potential (Figure 4), but this is not well integrated with the vertical response – which is essential to understand the stocks of wetlands. Further the results are different from the earlier paper of Schuerch et al (2018) which was rather optimistic about the future of wetlands under sea-level rise assuming the available accommodation space was available for migration. In areas such as Europe, North America and parts of Asia there are large areas where wetlands could migrate if coastal land use changed – so there is a lot of accommodation space. Why is this paper coming to a different conclusion? In one response to an earlier reviewer it says the data has been changed, but if this leads to a different view to the earlier paper this demands more explanation at least in the Supplemental Material. Other climate change drivers are mentioned although this is rather short. Other key drivers not linked to climate change such as declining sediment budgets reflecting human modifications to catchments are not mentioned at all. To be fair this is hard to quantify, but it would seem that new sediment input today is lower than in earlier times which may have important implications. Hence, I am left rather sympathetic to the remarks of reviewer #3 who was quite critical of the earlier manuscript.

Hence, my conclusion is that this needs major revision to see if the authors can address these comments – maybe by reframing the interpretation. What is a safe operating space for these coastal habitats? Is this even the right concept to frame the insights that this analysis provides? Other comments are given below.

Line 44-45 “As rates of RSLR exceed ~ 7 mm yr⁻¹, it becomes very likely ($P \geq 0.9$) that mangroves and tidal marshes eventually drown” – yes but significant drowning would be observed for smaller rises in sea level based on the data presented?

Line 50-51 “The Paris agreement therefore delineates a safe operating space whereby disruption to coastal ecosystems is minimised.” Minimised or manageable? To minimise disruption you would like a slower rate of sea-level rise than will occur under the Paris Agreement?

Line 122-123 GIA – isn't this a global process and in some places it is much more significant. Needs to be rephrased to express this point.

Line 174 Figure 1 – while there is only one case, coral islands seem to have a lower threshold than mangroves.

Line 210 Figure 2 – looking at these figures and the median – when I suppose there is a 50:50 chance of loss gives much lower thresholds of sea-level rise for loss. At 7 mm/yr you are very confident of loss, but at lower rates of rise there are still reasons for concern?

Line 273 Figure 3. What does a black coastline mean? Add to the key.

Line 305 – what about non-climate effects such as declines in sediment supply due to changes in the catchment – these already affect many catchments and the effect is likely to be enhanced by dam construction? Hence the geological analogues may be optimistic about the ability to respond as the sediment budget has changed? A paragraph on non-climate influences is needed here.

Line 352 – mangrove expansion in Siberia? Maybe temperate wetland expansion?

Line 357 Figure 4 and its discussion – results seem more pessimistic than Schuerch et al (2018) where it was argued that if the available accommodation space was available, wetland migration meant that losses would be small – and by memory gains were even possible. Why are the results so different here?

Line 392-395 “In the face of irrevocable disruption under high rates of RSLR, the most effective means of ensuring the continued survival of widespread mangrove forests, tidal marshes, and coral

reef islands is to limit global warming to less than 2°C above pre-industrial levels.” – not convinced based on the evidence presented that substantial losses can be avoided. 2°C will minimise losses but maybe we have already left a safe operating space, especially if human effects such as reduced sediment input are considered.

Author Rebuttals to First Revision:

The responses to individual comments by Referees 1, 2, 3 and 6 are provided in blue text. Many thanks again for these helpful comments

Referee #1 (Remarks to the Author):

Great paper, well done.

Here are some comments

Ln188 - missing comma

Comma inserted between the references

195-197 - i think sentence needs some "and" and comma alterations?

Good point, though we now bracket the phrase “the compaction of sediment in the upper few metres of the marsh”, which we think achieves this purpose.

224 - Was this only done with tidal marsh? or also mangroves - the methods are a bit unclear. Yes, both the change to open water and water observation analyses were only conducted for tidal marshes, because we found mangrove canopy cover to obscure observations of change in surface water occurrence. This is now made clearer in the Methods (new lines 809-810).

298 - I'm a little worried casual readers will read this as "we're likely to lose 81% of mangroves even if we achieve stability at 1.5 degrees" - could you write "...vulnerable to elevation deficit and eventual loss"

Good suggestion- altered as suggested.

344 - I don't think this paper (your ref 13) suggests that blue carbon delivered the 5 ppmv decline? It may have played a (very) small part in that decline

Changed to “may have contributed to”. The suggestion of Fig 4 in ref 13 is that the timing is highly suggestive of contribution.

351 - This sentence is about mangrove AND TIDAL MARSH unless you're suggestion mangroves in Siberia!

Thanks- now corrected to “coastal wetland”

Figure 4 - could the lines in the legend be thickened to make them more clearly readable? Done

378 - Northern expansion of tidal marsh will also presumably be enabled by reduced ice-cover and ice scour along many coasts

This is an excellent point, and we have rewritten the following sentence to read: Far northern coastlines thereby emerge as important future habitat for tidal marsh, as also projected for seagrass and kelp forests^{20,87,88}, under warmer temperatures and reduced ice cover and ice scour⁸⁹, increasing their relative contribution to “blue carbon” capture and storage. With the new reference (89) being Krause-Jensen, D. & Duarte, C. M. Expansion of vegetated coastal ecosystems in the future Arctic. *Frontiers in Marine Science* **1**, 77 (2014).

392 - perhaps beyond the role of a reviewer but I think it would tie in with previous comments to add a third element to this sentence and mention managed retreat and allowing/encouraging the formation of new coastal wetlands landwards of current systems?

We like this idea and have rewritten the concluding sentences as follows: “*In the face of irrevocable disruption under high rates of RSLR, the most effective means of ensuring the*

continued survival of widespread mangrove forests, tidal marshes, and coral reef islands is to achieve the Paris Agreement goal of net zero emissions by mid-century. To this end, a contribution will be made by the preservation, restoration and landward accommodation of coastal “blue carbon” ecosystems.” (lines 400-404)

Table S2 - final column, is this a "rate" (annual) or total during the period of readings. We have now included the unit (mm yr⁻¹) which clarifies this

797 - I think you may not even be allowed to use this data without using the citations provided at this website, but even if you are it is poor practise not to acknowledge the research properly. The Ocean Data Viewer is a data library. Some readers may also want to know which version of which maps from that site you may have taken (and which year for the mangroves). I have just checked and you can click on the relevant layers and use the citations provided. Please also take this as a wider message for using this data in other publications. WCMC did not produce it and do not own it.

All appropriate references have been included. They are as follows:

- Mcowen C, Weatherdon LV, Bochove J, Sullivan E, Blyth S, Zockler C, Stanwell-Smith D, Kingston N, Martin CS, Spalding M, Fletcher S (2017). A global map of saltmarshes (v6.1). Biodiversity Data Journal 5: e11764. Paper DOI: <https://doi.org/10.3897/BDJ.5.e11764>. Data DOI: <https://doi.org/10.34892/07vk-ws51>, last accessed: 13 September 2021.
- Spalding M, Kainuma M, Collins L (2010). World Atlas of Mangroves (version 3.1). A collaborative project of ITTO, ISME, FAO, UNEP-WCMC, UNESCO-MAB, UNU-INWEH and TNC. London (UK): Earthscan, London. 319 pp. URL: <http://www.routledge.com/books/details/9781844076574>, Data DOI: <https://doi.org/10.34892/w2ew-m835>, last accessed: 13 September 2021.
- UNEP-WCMC, WorldFish Centre, WRI, TNC (2021). Global distribution of coral reefs, compiled from multiple sources including the Millennium Coral Reef Mapping Project. Version 4.1, updated by UNEP-WCMC. Includes contributions from IMaRS-USF and IRD (2005), IMaRS-USF (2005) and Spalding et al. (2001). Cambridge (UK): UN Environment Programme World Conservation Monitoring Centre. Data DOI: <https://doi.org/10.34892/t2wk-5t34>, last accessed: 13 September 2021.
- Goldberg, WM (2016). Atolls of the world: revisiting the original checklist. Atoll Research Bulletin 610: 1-47. Data: <https://www.arcgis.com/home/item.html?id=1c18adf04d9e47669281061ff60167e1>, last accessed: 13 September 2021.

802 - I hadn't come across this, its interesting, but would certainly benefit from a better citation, though you may need to construct it. Looks like it was built at FIU?

Yes, we have included now a more appropriate citation, as above, i.e. Goldberg, WM (2016). Atolls of the world: revisiting the original checklist. Atoll Research Bulletin 610: 1-47. Data: <https://www.arcgis.com/home/item.html?id=1c18adf04d9e47669281061ff60167e1>, last accessed: 13 September 2021.

807 - In title and section I am not sure if you only look at tidal marsh or also mangroves - tighten language throughout section, and if you didn't look and mangroves in this spell it out. We have clarified this referring to “tidal wetland” throughout this section, with line 790 stating “losses and gains of tidal marshes, tidal flats and mangroves (collectively termed ‘tidal wetlands’)”

815 - need to cite the GSW dataset (Ref 53)

Now included

821 - is "marsh" tidal marsh or coastal wetlands

Tidal wetland- now clarified

Fig 5 - poor resolution in my version so all maps are almost identical. But I THINK bottom right map should be identical to figure 4a, but doesn't look like it is? Maybe some work to clarify these maps and description

(Figure S5): Given this is now supplementary, we will present these as larger panels to improve readability. Yes, figure 4a and S5d are identical

Table S1 - I am finding this very confusing and it would benefit from a clearer explanation in the text.

We have written the caption of Table S1 to read:

Table S1: Area (km², and proportion) of mangrove, tidal marsh and number of coral reefs exposed to various rates of RSLR under policy-relevant warming scenarios.

Scenarios represented are the 1.5°C, 2.0°C, 3.0°C, 4.0°C, and 5.0°C warming to 2081-2100 against the 1850-1900 baseline, reporting the 17th, 50th and 83rd percentile projections for each scenario. The average rate of RSLR, and rate of global SLR are represented for each scenario as mm yr⁻¹

Referee #2 (Remarks to the Author):

The authors have considered all referee remarks very seriously, and the present version of the manuscript is substantially improved.

In response to your explicit question, I do not see any problem with the statistical methods or with the way in which statistical results are reported.

The reworking and the replies to the comments have helped me understand that my first reading of the manuscript was incomplete and partially wrong. Yet, I think some additional changes are needed in order to prevent misunderstanding by the readers.

My misunderstanding was caused by not properly distinguishing between the terms 'retreat' and 'expansion', as used in the description of the paleo-based studies, or 'loss' in the description of the remote sensing studies, and changes in the areal extent of marshes and mangroves. In the paleo studies, e.g. of the marshes in the UK, the authors use the term 'retreat' as indication of landward lateral movement of the marsh system. Evidence for retreat can come from marsh sites turning into tidal flats, but also from terrestrial sites turning into marsh. (I think it is a pity that the two transitions are pooled into one indicator and not distinguished in the figures, as I would have been interested in how much evidence for the two types of retreat actually existed).

To some extent this is a limitation of the source paleo studies. For example, Horton et al. 2018 presents "negative index points", which can be transitions from terrestrial to tidal, or from tidal to subtidal, without distinguishing the two. We now use the term "retreat" to encompass both these transitions in the paleo record, and also the ultimate influence of elevation deficits on existing wetlands. In this context, retreat does not necessarily imply a loss of areal extent, just a net landward movement. It is difficult given the data available to determine how areal extent will change over time, except that the palaeo record suggests a narrowing under high rates of RSLR (i.e. the rate of loss at the seaward edge eventually exceeds the rate of new formation at the landward edge).

'Expansion' is used for evidence of seaward lateral migration, either by marsh turning into terrestrial vegetation (which is areal loss) or tidal flats turning into marsh (areal gain). Most importantly, neither 'retreat' nor 'expansion' say anything about the areal extent of the marsh. Also when interpreting remote sensing images, the results express information on changes in existing marshes. When the images show marshland being turned into water, this

is areal loss at the site where marshes are presently occurring, but this loss may or may not be compensated by areal gain further inland. Information on the tendency for lateral movement of the habitats may be translated into likely changes of areal extent of the habitats, when the accommodation space landward of the present marshes is taken into consideration. The authors do this, but only at the end of the manuscript and without properly introducing the importance of adding this part of the logic.

The paragraph lines 352-374 goes some way toward addressing this issue. We explore the potential for landward gains (expansion) in several locations in the short-term. The paragraph finishes with the following new sentence:

However, our analysis suggests that the long-term contribution of blue carbon to climate mitigation is compromised under higher emissions scenarios. With vertical adjustment mechanisms overwhelmed by high RSLR, the rate of seaward edge retreat is expected to exceed landward edge expansion, narrowing the coastal wetland fringes, consistent with palaeo observations{Törnqvist, et al. 2021 }.

The wording of the manuscript, also in the present version, adds to the confusion. The term ‘expansion’ in itself is not well chosen, as it suggests an increase of areal extent while this is not meant in the technical sense. Other examples of confused wording are: line 169: “potential for increased extent”; line 170: “vertical adjustment and habitat extent are greatly reduced”; line 224: “ changes in the extent of tidal marsh” where changes in extent at the present location of occurrence is meant, not total extent of the marsh; line 233: “proportion of tidal marsh conversion” where again proportion of existing tidal marsh is meant; line 262: “ eventual loss of mangroves and tidal marshes is likely” – loss of existing marsh at the present location is meant; line 283: “The probability of reaching RSLR at which loss is very likely” – I think that the evidence shows that landward migration is very likely, not necessarily loss as in loss of overall areal extent; but note at line 293 “with the greatest projected losses of existing habitat” is the correct formulation.

In practice, as is also shown by the authors, the difference between loss of habitat at the existing location and loss overall is probably small – there are too many places where landward migration is impossible. However, for a proper understanding of the link between the evidence and the conclusions it is very important to keep the concepts clear. I would like the authors to carefully consider the wording, and to add a paragraph in the beginning of the paper where the different terms (retreat, seaward progradation, landward migration possibilities, loss or gain of areal extent) are introduced and the logic of the paper (first consider where and when landward migration is observed, then translate it into likely areal loss by also considering the possibility of encroaching current terrestrial habitat) is explained. At least for readers like myself, this would greatly help in understanding the paper.

This is very helpful feedback. We have modified the second paragraph to fulfill the purpose suggested. This paragraph introduces concepts and appropriate terms. The section (lines 84-99) reads:

Human-induced climate change places these important ecosystems in an uncertain future. On the one hand, many of the most important coastal ecosystems show biogenic responses to relative sea-level rise (RSLR) that enhance their physical resilience². The potential for high rates of sedimentation, productivity, and/or organic matter preservation in mangroves, tidal marshes and coral reefs have enabled them to track past RSLR over millennia³. We refer to this process as

“vertical adjustment”. Vertical adjustment maintains a wetland above a drowning threshold, a buffer termed “elevation capital”¹⁴. For reef island systems, vertical adjustment maintains the uppermost portions of a reef at close to mean sea level. Where the rate of vertical adjustment falls behind the rate of RSLR, an “elevation deficit” emerges, and the surface is exposed to increasing depth and duration of inundation. This change in hydroperiod may enhance vertical adjustment¹⁵, but if a deficit is sustained for sufficiently long, elevation capital is exhausted. For wetlands, retreat and a transition to open water may occur, and in reef-islands submergence of reef crests will increase wave exposure and wave over-topping frequency. Whether the areal extent of the habitat expands or contracts over time depends on the rate of loss and the rate of new habitat formation, both processes being influenced by RSLR¹⁶.

We have carefully checked all references to the terms “extent”, “retreat”, “loss” and “migration” for clarity and consistency.

A few minor comments:

Line 46: “destabalise”

corrected

Line 69: “The presence of marine vegetation and fringing reefs attenuate wave energy, protecting coastlines, while providing habitat to distinctive assemblages of species.”

Sentence not properly structured. “presence” is subject but the verb is plural. ‘Presence’ may be left out.

corrected

Line 129: “Several periods of rapid global mean sea-level (GMSL) rise... have drowned mangrove forests and tidal marshes” Strange that periods can drown marshes

Restructured as follows: Rapid global mean sea-level (GMSL) rise (>10 mm yr⁻¹) over several periods since the Last Glacial Maximum has drowned mangrove forests and tidal marshes

Lines 161-164: this information on how areal extent can also be lost from the landward side is very important. It could be used to re-iterate the relation between ‘expansion’ and areal extent

Yes, we return to this in lines 373-375 and 384-386

Line 203: “The results were consistent with those for tidal marshes.” This is not entirely true. The data shown in the supplementary material for mangrove SET sites do not show any relation at all between RSLR and rate of accretion, in contrast to what is described in the preceding paragraph for marshes.

The phrase “entirely consistent” refers to the statement made in the previous sentence, i.e. “estimating the cumulative probability of vertical adjustment at or exceeding the rate of RSLR at the SET-MH stations”. The profiles of this relationship (2b for tidal marsh, 2c for mangrove) are almost exactly the same.

Line 377: “Tidal marsh habitat is likely to expand in northern Siberia under higher RSLR due to a lack of topographic and human development impediments”. Here, if I understand it well, ‘expansion’ of the marsh in landward direction is meant, which is very inconsistent with the earlier use of the word (expansion = seaward migration).

We have changed this to “expand in extent” to be consistent with our usage introduced in paragraph 2

Referee #3 (Remarks to the Author):

I appreciate the authors' efforts in addressing most of all my (and the other reviewers') comments in their manuscript. I consider that the revised manuscript provides timely findings that are now better supported by their analyses. Nonetheless, I think there is still space for improvement.

1) In response to one of my comments on the initial version of this manuscript, the authors included new analyses on how the habitat extent of tidal marshes responds to current sea level rise. The results show that the probability of tidal marsh contraction and increased water cover increase with increasing rate of RSLR (Fig. S3a, b). But it's unclear how these results are in line with their choice of 7mm yr⁻¹ RSLR as a threshold rate above which eventual loss of tidal marshes is very likely or likely. The probability of tidal marsh contraction appears to change with RSLR by only a small extent, and even when RSLR is greater than 7mm yr⁻¹, probability of contraction remains smaller than 0.6. And unlike those shown in Fig. 2, all the responses shown in Fig. S3 are more like gradual processes, rather than threshold processes. To predict the dynamics of coastal ecosystems under sea level rise, it's key to understand their extent changes (to a lesser extent, elevation changes). Elevation deficit does not necessarily go in parallel with marsh extent decline. If extent responses given in Fig. S3 are used as the main basis to define the threshold rate of RSLR and to project coastal ecosystem responses to future sea level rise, would that lead to major changes in the authors' findings and conclusions?

The problem with using extent responses as indicative of long-term behaviour of a wetland is that these responses will be lagged. Wetlands higher in the tidal frame will not show conversion to open water as soon as a deficit between elevation gain and RSLR is established, and may survive for centuries before a critical "drowning" elevation is reached. For this reason we would not expect to find a close relationship between elevation deficit and wetland loss (i.e. marsh conversion to open water) in the short term, but the fact that this relationship is significant is indicative of the long-term fate (remembering that the deficits are likely to have been established only in recent decades, and decades to centuries may be required for existing wetlands to reach a drowning elevation). We prioritise elevation trajectory in relation to RSLR in defining vulnerability because this will ultimately determine the outcome.

To illustrate this point we have introduced an additional analysis, comparing the increase in surface water, and tidal marsh extent with elevation deficit and RSLR between sites of lower than median elevation and sites of higher than median elevation. This is now presented as Extended Data Fig 5, supported by the following text (new lines 235-246):

The relationship between surface water change and marsh elevation deficit was stronger for lower elevation marsh sites ($r^2=0.20$) than higher elevation sites ($r^2 = 0.03$; Fig. S5), illustrating the temporary resilience conferred by elevation capital. ...The ameliorating influence of elevation capital was also evident in the extent of marsh loss. Where marshes had higher than the median elevation capital, there was no relationship between marsh loss and RSLR ($p=0.850$). At lower than the median elevation capital, the relationship was highly significant ($p=0.002$).

It's also unclear if the analysis given in Fig. S3a captures the responses of the full extent of tidal marshes to RSLR. Or it's primarily about the responses of marsh seaward edge? And how was the influence of barriers for marsh landward migration accounted for? To fully understand the results of this analysis, these points need to be clarified.

We have sought to clarify the extent of polygons sampled in the figure caption. They represent a 5km² buffer around each SET-MH platform (the 477 globally distributed platforms from which elevation surplus/deficits were measured).

2) The manuscript was still poorly written in multiple sections. For example, although the manuscript has a section on past sea level rise, the paleo analyses in Fig. 2d-f are described only in the section on current sea level rise.

We have extensively revised the manuscript to ensure clarity and logical progression and consistency. We have rectified this omission, referring to Fig 2e, and 2f in lines in the paragraph 141-160

This issue is also obvious in the section “Opportunities for restoration and migration”.

Restoration and migration issues are discussed in the middle two paragraphs of this section, while “safe operating safe” is the key topic of the last paragraph. Transition between the two parts is abrupt, with no connections.

We have provided two linking sentences here which improves the flow of the logic, emphasising the potential short-term wetland resilience

The “Methods” section also suffers from similar issues. I suggest the authors reorganize all the methods according to the sections (and their order) in the main text. Some figures and tables are inserted within the text, while others are put together at the end. Be consistent!

There are also not a few conciseness and clarity issues.

We have carefully reviewed the Methods section for clarity and consistency and made several edits. All figures and tables now appear separately to the text

3) The accuracy of some statements still needs to be checked. For example, L230, RSLR was given as 3.5 mm yr⁻¹, but in Fig. S3, the data value is shown as 2.3 mm yr⁻¹. L265, here Fig. 2 is cited to refer to a value of RSLR 7 mm yr⁻¹, but maybe they meant Fig. S3c (but the number given in Fig. S3c is 6.2 mm yr⁻¹).

We have thoroughly checked all quantitative statements in the text. The quoted rate on L230 has been corrected, as has the reference to Fig S2c rather than Fig 2. We use RSLR of 7mm yr⁻¹ as a reporting threshold across scenarios, and so corrected the wording of L271 to “island instability by RLSR of 7 mm yr⁻¹ is likely...”, noting that the specific value of 6.2 mm yr⁻¹ is provided in L252. L240 was corrected from 5.5 mm yr⁻¹ to 5.4 mm yr⁻¹.

Minor comments:

L47, I would say “mapped” tidal marshes
done

L114, “Stability” is a term used in multiple places in this paper, but I don’t think it is analyzed per se. I would use expansion instead here.

done

L118, Unclear what they mean by “refugia”. Refugia is not discussed anywhere else in the manuscript.

This phrase has been deleted

L154-156, this appears repetitive of what has been described in L144-146.

We have consolidated L154-156 with L144-146 (now L164-168). Together they describe a chronology of marsh response to slowing RSLR.

L161-162, Typically, extension of deltas will lead to expansion of intertidal areas, not reduction.

We removed “extension of deltas”

L167, RSLR rates in the instrumental period are not aforementioned.

We have included (<5 mm yr⁻¹)

Fig. 1 Sites in panel b and those in panel c do not match.

We used a representative subset of sites for panel b- including all would obfuscated the figure L183-184, rewrite

Changed to: *Mangrove and tidal marsh accretion can increase with the rate of RSLR. Increased inundation depth and duration may facilitate both mineral deposition⁴⁷ and higher plant productivity and root mass accumulation¹².*

L201, I think the Bayesian analysis needs to be briefly described.

We now refer to the Methods where the Bayesian analysis is described (lines 834-887)

L206-208, this paleo analysis needs to be moved to the section on “past sea level rise”

The point we make here is that the relationships described in lines 150-162 are nearly identical to that inferred from the SET-MH record. We now make clearer in L158 that we are referring to the data represented in Fig 2e,f.

L226, Fig. S3?

No, just Fig S2. These are the histograms from which the profiles in Figure 2 are derived.

L291, The projection for 3°C warming scenario should be described before making the indication on tipping points.

Good point- the paragraph has been reorganised accordingly

L351, but the projections in Fig. 4 show minimal opportunities for mangrove landward expansion.

Here we are drawing attention to those areas with >90% of current area available

Figure 4, by “retreat”, the authors meant “migration”?

Yes, we have clarified our terminology in this regard, as also requested by Referee 2. Retreat is used consistently.

L343-369, Those two paragraphs heavily focus on mangroves. More should be given to tidal marshes.

This paragraph is now immediately followed by a paragraph devoted to tidal marsh projected changes. In the previous version, a paragraph separated these two, and the restructuring of this section now provides clarity.

L732, clarify “512x260”

Changed to “500-year periods on a 512 x 260 global latitude x longitude grid, resulting in predictions of GIA at > 130,000 points in space for each time period”.

L736-750, these are not really Methods and appear repetitive of what has been described in main text.

This section of the Methods is provided in response to Referee 2’s request in the initial review round that more information be provided in supplementary materials relating to the derivation of Figure 1, which this provides. Lines 736-741 describe the GIA modelling (a key method), while lines 727-743 provide clarity concerning the source studies for Figure 1.

Figure S1, The right part of this figure might better illustrate how tidal gauges measure relative sea level rise, instead of how relative sea level rise affects various marsh ecological processes.

We think the left and right parts match well- in that the right-hand representation of processes helps the reader interpret the variables measured in the SET-MH technique.

Table S2, no data source was given for the several sites in Florida, USA

The source is now included (Feher et al. 2022).

L903, delete “of”

done

L927, 5°C is missing?

We have now included 5°C in Table 1

L1021, platform?

No, “planiform” as in “the vertical orthographic projection of an object on a horizontal plane”

Referee #6

Overall, I find the approach interesting and thought-provoking about the global response of selected coastal ecosystems – mangroves, tidal marshes and reef islands to sea-level rise and climate change. It is good to see multiple lines of evidence concerning the vertical response of wetlands considered and analysed -- palaeo-environmental data since the Last Glacial Maximum, the data from the Surface Elevation Table-Marker Horizon (SET-MH) network, and contemporary coastal ecosystems response to relative sea-level rise in terms of conversion to open water.

The paper is less strong on integrating the potential for horizontal translation which could – at least in theory – offset vertical loss. This was a key conclusion the paper by Schuerch et al (2018) in Nature – the same model is used in this paper and I am unclear why there is a different outcome.

The simplest answer to this question is that the results are largely the same but represented differently. We show in Figure 4 the potential landward migration area represented as a proportion of current area. It is similar in this sense to Schuerch et al Fig 2b, the relative change in wetland extent to 2100, but we did not seek to model projected total wetland extent in 2100 (that is, we do not model seaward edge retreat). There were two reasons for this (i) current wetlands are an artefact of a 6000-year stable sea-level history, and their short-term resilience relies on elevation capital accumulated over that time. That is, projecting to 2100 and showing an expansion in overall extent gives an overly optimistic sense of the fate of wetlands under RSLR. (ii) the modelling of seaward edge retreat in Schuerch et al 2018 is based on theoretical relationships of Kirwan et al. 2016 which our data dispute.

The main difference in parameterisation related to our use of the most recent sea-level rise projections developed for AR6, which differ in some aspects to previous sea-level rise modelling used in Schuerch et al. 2018.

Hence, I am left wondering does the evidence presented support the conclusion of the paper that a 2 degree world is fundamentally different from a three degree world for coastal ecosystems which is what the conclusion of 2 degrees being a safe operating space and 3 degrees not being a safe operating space required? I can equally interpret from their results a world where coastal ecosystems are declining with today's sea-level rise (below a 2 degree world) and they will decline more rapidly if sea-level rise accelerates and other climate changes. However, the horizontal response can offset these losses and is 2 degrees different from 3 degrees in the area response – combined vertical and horizontal response? The horizontal response is often conditioned by how human beings behave in general and adapt in particular which was what Schuerch et al (2018) argued.

Paleo analogues consistently show that rates of RSLR consistent with 2-3 degrees of warming lead to transient, rapidly transitioning wetlands, and we review the evidence for this. Tornqvist et al 2021 provide a detailed treatment of this issue. Under constant sediment delivery and expanding accommodation (both lateral and vertical), intertidal lateral width must decline, as defined by the conservation of mass. The palaeo record appears consistent with this principle. We have not modelled total wetland extent for the reasons outlined above. Instead, we make the case that warming associated with emissions over the coming few decades will likely seal the fate of much of the world's existing tidal wetlands and coral

islands as we know them. What survives is likely to be transient wetlands with very different properties.

Looking at Rockstrom (2009), a safe operating space identifies and quantifies planetary boundaries that must not be transgressed to help prevent human activities from causing unacceptable environmental change. Does this paper really show that 2 degrees is qualitatively different to 3 degrees. In the vertical response maybe – but even there I am not entirely convinced. If we look at the vertical response in Figure 2, I consider the median – which I interpret as a 50:50 chance of a deficit – above the median we would seem to be in a regime of net loss – at least in a one-dimensional view of the problem. The median is associated with a RSLR of 1 to 3 mm/yr and given that global sea-level rise is now above 3 mm/yr, we are already into the territory of global decline in wetlands. In Figure 1, the big response in the palaeo-record (with exceptions) would seem to be the decline below 3 mm/yr about 8,000 years ago? Lastly Figure S3(b) again shows the median linked to rates below 3 mm/yr. I am interested in the authors response to my interpretation.

You have raised an important issue here, and much of our most recent revision has sought to address this concern. Earlier versions of our paper were too focussed on 7mm yr^{-1} of RSLR as an upper threshold beyond which wetlands struggled to keep up. In subsequent revisions we have rephrased this more accurately as the rate at which elevation deficits are “highly likely” ($p=0.9$, to use the IPCC nomenclature). Fig 3 shows most of the mid to low latitude coastlines crossing into this category between 2 and 3 degrees of warming. However, your point about lower rates is entirely valid. At 4 mm per year, wetlands are “likely” ($P=0.66$) to show an elevation deficit, according to both the paleo and SET data. This risk is not sufficiently expressed in the version you reviewed, and we have sought in the current version to rectify this. We now more strongly emphasize the risks associated with the 4-7 mm per year RSLR in lines 268-310.

The horizontal response is considered via the wetland inland migration potential (Figure 4), but this is not well integrated with the vertical response – which is essential to understand the stocks of wetlands.

As suggested above, the 2100 time-horizon gives an overly optimistic view of the fate of the world’s wetlands under current and projected rates of warming because of the short-term influence of elevation capital. In future work we would seek to run the global wetland model to 2300, an exercise that would have allowed for both the exhaustion of elevation capital in nearly all existing wetlands, and also the influence of higher rates of RSLR in narrowing the width of transgressing wetland. This is not currently possible given data and model limitations. Instead, we show the proportion of habitat types exposed to elevation deficits under different warming scenarios, making the point that these deficits are “locked in” (i.e. eventual drowning is inevitable).

Further the results are different from the earlier paper of Schuerch et al (2018) which was rather optimistic about the future of wetlands under sea-level rise assuming the available accommodation space was available for migration. In areas such as Europe, North America and parts of Asia there are large areas where wetlands could migrate if coastal land use changed – so there is a lot of accommodation space. Why is this paper coming to a different conclusion? In one response to an earlier reviewer it says the data has been changed, but if

this leads to a different view to the earlier paper this demands more explanation at least in the Supplemental Material.

The modelling of Kirwan et al (2016), upon which Schuerch et al (2018) is based, suggests the potential for landward migration (lateral expansion) while existing wetland potentially remains in place, due to potentially high vertical accretion under sufficient sediment inputs. Even so, under high RSLR the Kirwan et al. (2016) model shows few wetlands keeping pace, due to a lack of sufficient sediment delivery. More importantly, Tornqvist et al (2021) point out, the notion of wetlands keeping pace with increasing RSLR while accommodation expands does not allow for the conservation of mass (an increase in area and depth of accommodation would require a proportionate increase in sediment input). It is inevitable that wetlands narrow unless there is a substantial increase in sediment inputs. Our analyses (as explored also in Saintilan et al. 2022) show that vertical development is further constrained by the influence of subsidence on allochthonous contributions. Even if more sediment were available, much would be lost to elevation gain because of a higher rate of subsidence. In this, the paleo and observation data are in agreement.

Other climate change drivers are mentioned although this is rather short. Other key drivers not linked to climate change such as declining sediment budgets reflecting human modifications to catchments are not mentioned at all. To be fair this is hard to quantify, but it would seem that new sediment input today is lower than in earlier times which may have important implications. Hence, I am left rather sympathetic to the remarks of reviewer #3 who was quite critical of the earlier manuscript.

We have introduced a paragraph under the heading “Interactions with future sedimentation, CO₂ and temperature” as follows:

Processes influencing the adjustment of coastal wetlands and reefs to sea-level rise may be modified by climate change, though often the influence is to suppress vertical adjustment. Catchment land-use change driven by population growth may increase sediment transport by rivers, subsidising sediment accumulation in coastal deltas (Dethier et al. 2022, Nienhuis et al. 2023). Counteracting this is the association between economic development and dam construction, an intervention which retains sediment within catchments. Sediment yields to coastal environments in the global north are nearly half those prior to hydrological modifications (Dethier et al. 2022). Major hydrological developments in SE Asian rivers have implications for the resilience of mangroves to sea-level rise (Lovelock et al. 2015).

Hence, my conclusion is that this needs major revision to see if the authors can address these comments – maybe by reframing the interpretation. What is a safe operating space for these coastal habitats? Is this even the right concept to frame the insights that this analysis provides?

We agree that the revisions are best completed as a re-framing of the interpretation. We have sought to provide the right balance between a recognition of “committed loss”, even under lower temperature increments, and the opportunities for conservation and restoration where elevation deficits are low.

Other comments are given below.

Line 44-45 “As rates of RSLR exceed ~7 mm yr⁻¹, it becomes very likely (P³0.9) that

mangroves and tidal marshes eventually drown” – yes but significant drowning would be observed for smaller rises in sea level based on the data presented?

Good point- this section has been rewritten as follows:

A deficit between tidal marsh and mangrove adjustment and RSLR is likely ($P=0.67$) at 4mm yr^{-1} of RSLR, and very likely ($P=0.9$) at 7mm yr^{-1} of RSLR. As rates of RSLR exceed $\sim 7\text{mm yr}^{-1}$, it also becomes likely that reef islands destabilise as reef crests are over-topped by waves. Exposure of habitats to these rates of RSLR is highly sensitive to warming levels. Increased warming from $1.5\text{ }^{\circ}\text{C}$ to $2.0\text{ }^{\circ}\text{C}$ would double the area of mapped tidal marsh exposed to 4mm yr^{-1} of RSLR, to 65%. Nearly all the world's mangrove forests and coral reef islands and 40% of mapped tidal marshes are estimated be subject to $\text{RSLR} \geq 7\text{mm yr}^{-1}$ by 2100 under a climate warming scenario of $3\text{ }^{\circ}\text{C}$.

Line 50-51 “The Paris agreement therefore delineates a safe operating space whereby disruption to coastal ecosystems is minimised.” Minimised or manageable? To minimise disruption you would like a slower rate of sea-level rise than will occur under the Paris Agreement?

This sentence has been deleted.

Line 122-123 GIA – isn't this a global process and in some places it is much more significant. Needs to be rephrased to express this point.

Line 174 Figure 1 – while there is only one case, coral islands seem to have a lower threshold than mangroves.

They certainly formed later during the high-stand. This however may be due to a lag between sea-level stabilisation, coral reef growth to the surface and the accumulation of sediment.

Line 210 Figure 2 – looking at these figures and the median – when I suppose there is a 50:50 chance of loss gives much lower thresholds of sea-level rise for loss. At 7mm/yr you are very confident of loss, but at lower rates of rise there are still reasons for concern?

Yes. There is “likely” elevation deficit at 4mm yr^{-1} . We include these levels in the probability distributions also.

Line 273 Figure 3. What does a black coastline mean? Add to the key.

Caption now reads “Coastlines with mapped mangrove, tidal marsh or reef habitat subject to...”

Line 305 – what about non-climate effects such as declines in sediment supply due to changes in the catchment – these already affect many catchments and the effect is likely to be enhanced by dam construction? Hence the geological analogues may be optimistic about the ability to respond as the sediment budget has changed? A paragraph on non-climate influences is needed here.

A paragraph has been added, as previously mentioned.

Line 352 – mangrove expansion in Siberia? Maybe temperate wetland expansion?

This has been corrected- Referee 1 also spotted this.

Line 357 Figure 4 and its discussion – results seem more pessimistic than Schuerch et al (2018) where it was argued that if the available accommodation space was available, wetland migration meant that losses would be small – and by memory gains were even possible. Why are the results so different here?

Schuerch et al. (2018) concludes that gains are only possible if coastal management strategies are implemented to allow for them. In the current situation losses of up to 30% are estimated by 2100. In our paper we were reluctant to talk about overall gains or losses to 2100 for the reasons outlined above. Contemporary wetlands are an artefact of a 6000-year history of stable sea-level. Their loss to RSLR will be lagged because of existing elevation capital. Gains to 2100 are possible because in some locations increases at the landward edge will not be matched by recession at the seaward edge (though note in Schuerch et al 2018 Fig 2b that many coastlines show declines relative to current extent). If our suggestion is right that contemporary wetlands are already on a trajectory toward drowning, the important factor is how much landward accommodation is possible. Our Fig 4 shows that for most of the world's coastlines, this is fairly small (at least to 2100) compared to contemporary wetland extent.

Line 392-395 “In the face of irrevocable disruption under high rates of RSLR, the most effective means of ensuring the continued survival of widespread mangrove forests, tidal marshes, and coral reef islands is to limit global warming to less than 2°C above preindustrial levels.” – not convinced based on the evidence presented that substantial losses can be avoided. 2°C will minimise losses but maybe we have already left a safe operating space, especially if human effects such as reduced sediment input are considered.

This is a good point and we have rewritten this section. We are seeking to strike the right balance between the concerning implication of elevation deficits and the need for ongoing conservation and restoration of coastal ecosystems.

References cited

Dethier, E. N., Renshaw, C. E., & Magilligan, F. J. (2022). Rapid changes to global river suspended sediment flux by humans. *Science*, 376(6600), 1447-1452.

Kirwan, M. L., Temmerman, S., Skeeahan, E. E., Guntenspergen, G. R., & Fagherazzi, S. (2016). Overestimation of marsh vulnerability to sea level rise. *Nature Climate Change*, 6(3), 253-260.

Lovelock, C. E., Cahoon, D. R., Friess, D. A., Guntenspergen, G. R., Krauss, K. W., Reef, R., ... & Triet, T. (2015). The vulnerability of Indo-Pacific mangrove forests to sea-level rise. *Nature*, 526(7574), 559-563.

Nienhuis, J. *et al.* Reply to: Concerns about data linking delta land gain to human action. *Nature* **614**, E26-E28 (2023).

Saintilan, N., Kovalenko, K. E., Guntenspergen, G., Rogers, K., Lynch, J. C., Cahoon, D. R., ... & Khan, N. (2022). Constraints on the adjustment of tidal marshes to accelerating sea level rise. *Science*, 377(6605), 523-527.

Schuerch, M., Spencer, T., Temmerman, S., Kirwan, M. L., Wolff, C., Lincke, D., ... & Brown, S. (2018). Future response of global coastal wetlands to sea-level rise. *Nature*, *561*(7722), 231-234.

Törnqvist, T. E., Cahoon, D. R., Morris, J. T., & Day, J. W. (2021). Coastal wetland resilience, accelerated sea-level rise, and the importance of timescale. *AGU Advances*, *2*(1), e2020AV000334.

Reviewer Reports on the Second Revision:

Referees' comments:

Referee #3 (Remarks to the Author):

This manuscript has been significantly improved again. It's amazing to see how it took shape step by step. I have only a few relatively minor suggestions on this version. I look forward to its publication.

1) The results given in Fig. 4 are not sufficiently described in the main text. Two scenarios of wetland retreat are modeled here: retreat possible with population density below 20 people km⁻², and retreat unimpeded by population density. As the authors explain in Methods, the first scenario (retreat possible with population density below 20 people km⁻²) represents a best-case scenario. But isn't "retreat unimpeded by population density" also a best-case scenario? Maybe better include an average scenario or a worst-case scenario in the main figure for reference?

2) There is still some space to improve the supplementary Methods section. For example, the authors used multiple linear regression to test the relationships between surface water and tidal wetland change vs. contemporaneous RSLR and elevation deficit. But there are no details about how multiple linear regression was conducted, what were the predictor variables, and what were the results. The many predictors given in Fig. 4B and Extended Data Table 3 were not mentioned nor explained in the Methods or main text.

The subsection "Ecosystem stability under RSLR - Bayesian modelling" includes many analyses other than Bayesian modeling. So this is not an appropriate heading. Again, I think stability is not really analyzed in this manuscript. So I would avoid the use of "stability" wherever possible. The authors also used the concepts "vulnerability" and "resilience". How to distinguish these terms? What's the difference from stability?

This subsection also includes analyses on both paleo and contemporary sea level rise. I suggest that the authors reorganize the methods according to the order of these analyses given in the main text. Descriptions of the programs and any codes the authors used to conduct all these data analyses are completely lacking.

3) The authors did not respond, in a straightforward way, to one of my questions from last round of review: It's also unclear if the analysis given in Fig. S3a captures the responses of the full extent of tidal marshes to RSLR. Or it's primarily about the responses of marsh seaward edge? And how was the influence of barriers for marsh landward migration accounted for? To fully understand the results of this analysis, these points need to be clarified.

What's the representativeness of the 5km² buffer around each SET-MH platform? Does it cover barriers for marsh landward migration? It's not a problem if within the 5 km² buffers were all marshes or open water. But this might differ between a small and a big marsh.

Line 160, should this be "as the rate of RSLR declined below 4 mm yr⁻¹"? It's not 7mm yr⁻¹.

Line 180, strange citation format

Line 207, "gain), increases", I don't think you need the comma here.

Line 250-252, a supplementary figure would better illustrate these trends.

Line 255, planiform?

Line 262-265, better move this sentence to the end of this paragraph?

Line 326, but land-use change described later in this paragraph is not "climate change"

Line 361, well-being?

Extended Data Figure 4: What does the GSW in Fig. 4A stand for? This was not defined anywhere. A positive value indicates deficit? A negative value indicates surplus? The use of the term "elevation deficit" does not seem to match the signs of the numbers.

Extended Data Table 3: Do you mean predictors? Any difference between identifiers and variables? Identifiers were never mentioned in main text and the Methods section. So, they came out of nowhere.

Referee #6 (Remarks to the Author):

Thank you for your responses to my review which largely address my concerns. I agree that the sea-level rise associated with the Paris Agreement are associated with much smaller risks to coastal ecosystems than higher warming scenarios. But even for current levels of sea-level rise the risks of significant decline are very real. I am happy for this manuscript to be published as presented here.

Author Rebuttals to Second Revision:

Referee #3 (Remarks to the Author):

This manuscript has been significantly improved again. It's amazing to see how it took shape step by step. I have only a few relatively minor suggestions on this version. I look forward to its publication.

1) The results given in Fig. 4 are not sufficiently described in the main text. Two scenarios of wetland retreat are modeled here: retreat possible with population density below 20 people km⁻², and retreat unimpeded by population density. As the authors explain in Methods, the first scenario (retreat possible with population density below 20 people km⁻²) represents a best-case scenario. But isn't "retreat unimpeded by population density" also a best-case scenario? Maybe better include an average scenario or a worst-case scenario in the main figure for reference?

We are following here the terminology used in Schuerch et al 2018, using the highly optimistic "no barriers" scenario and "best case" scenario, which we clarify now in the text as referring to the best of the scenarios incorporating population restrictions to landward encroachment (now "*..the best-case of these scenarios*"). The worst case scenario is included in Extended Data Figure 6, and reference to this figure is now made in the caption of Fig 4. Our reason for not including this in Figure 4 is that our purpose is to show that even the best case scenarios have significant limitations on landward encroachment in the short term compared to the extent of existing wetland threatened by RSLR.

2) There is still some space to improve the supplementary Methods section. For example, the authors used multiple linear regression to test the relationships between surface water and tidal wetland change vs. contemporaneous RSLR and elevation deficit. But there are no details about how multiple linear regression was conducted, what were the predictor variables, and what were the results.

The relevant section of the Methods now reads:

The relationships between surface water and tidal wetland change vs. contemporaneous RSLR and elevation deficit were tested using multiple linear regression. Predictive variables are provided in Extended Data Table 3, and consist of climatic, hydrological and edaphic properties associated with each SET-MH station, and are sourced from Saintilan, et al. 2022. Potential collinearity of predictors was assessed using variance inflation factor (VIF) from the car package⁹⁵. VIF was found to be below the level usually considered problematic (3.22). The overall relative importance of the key predictors was assessed using Random Forest (RF) regression analyses⁹⁶, a machine learning approach which tallies the results of small classification trees (n = 20,000) while retaining a bootstrapped subset of all observations for out-of-bag (internal) error testing. Analyses were performed in R version 4.1.3, and presented as Extended Data Fig 4a.

The many predictors given in Fig. 4B and Extended Data Table 3 were not mentioned nor explained in the Methods or main text.

Good point. We now include the line

Predictive variables are provided in Extended Data Table 3, and consist of climatic, hydrological and edaphic properties associated with each SET-MH station, and are sourced from ref⁴⁹.

The subsection “Ecosystem stability under RSLR - Bayesian modelling” includes many analyses other than Bayesian modeling. So this is not an appropriate heading.

The term Bayesian modelling has been deleted from the title of the subsection

Again, I think stability is not really analyzed in this manuscript. So I would avoid the use of “stability” wherever possible. The authors also used the concepts “vulnerability” and “resilience”. How to distinguish these terms? What’s the difference from stability?

This subsection also includes analyses on both paleo and contemporary sea level rise. I suggest that the authors reorganize the methods according to the order of these analyses given in the main text.

We have removed reference to stability in relation to tidal marsh and mangrove behaviour, and from the title in Methods section 4. We have retained the term in relation to coral island planiform responses to RSLR, because it captures the contrast between islands with little morphometric change and those (under RSLR) and substantial alterations.

Descriptions of the programs and any codes the authors used to conduct all these data analyses are completely lacking.

The closing paragraph of this section is now written as follows:

The relationships between surface water and tidal wetland change vs. contemporaneous RSLR and elevation deficit were tested using multiple linear regression. Predictive variables are provided in Extended Data Table 3, and consist of climatic, hydrological and edaphic properties associated with each SET-MH station, and are sourced from ref³⁸.

Potential collinearity of predictors was assessed using variance inflation factor (VIF) from the car package⁹⁶. VIF was found to be below the level usually considered problematic (3.22). The overall relative importance of the key predictors was assessed using Random Forest (RF) regression analyses⁹⁷, a machine learning approach which tallies the results of small classification trees ($n = 20,000$) while retaining a bootstrapped subset of all observations for out-of-bag (internal) error testing. Analyses were performed in R version 4.1.3 and presented as Extended Data Fig 4b.

3) The authors did not respond, in a straightforward way, to one of my questions from last round of review: It's also unclear if the analysis given in Fig. S3a captures the responses of the full extent of tidal marshes to RSLR. Or it's primarily about the responses of marsh seaward edge? And how was the influence of barriers for marsh landward migration accounted for? To fully understand the results of this analysis, these points need to be clarified. What's the representativeness of the 5km² buffer around each SET-MH platform? Does it cover barriers for marsh landward migration? It's not a problem if within the 5 km² buffers were all marshes or open water. But this might differ between a small and a big marsh.

The analysis considers change in tidal marsh extent in the immediate vicinity of the SET platforms. The method incorporated change at the seaward edge and landward edges, and interior marsh breakup (with the caveat that a minimum change size threshold is applied). We experimented with a few buffer sizes before settling on 5km². Smaller buffers were relatively insensitive to marsh breakup clearly visible in the imagery. Larger buffers would, as you suggest, extend beyond the marsh and possibly introduce biases in relation to encroachment opportunity.

Line 160, should this be "as the rate of RSLR declined below 4 mm yr⁻¹"? It's not 7mm yr⁻¹. Note that we say widespread mangrove forest development "commenced" below 7mm yr⁻¹.
¹ Ref 13 provides a fuller treatment on this point. It was more like 4 mm yr⁻¹ in many settings (including most of the new world, for instance), but the large expansionary phase of mangrove development commenced at 7mm yr⁻¹ of RSLR.

Line 180, strange citation format

Corrected. Most of these references were moved to the Methods section.

Line 207, "gain), increases", I don't think you need the comma here.

Yes- comma now removed.

Line 250-252, a supplementary figure would better illustrate these trends.

Though the relationship is significant the r² is not high. We constructed the figure but decided it added little.

Line 255, planiform?

Yes. Corrected

Line 262-265, better move this sentence to the end of this paragraph? Agreed- moved as suggested.

Line 326, but land-use change described later in this paragraph is not “climate change” The title is changed to “*The influence of global change drivers*”

Line 361, well-being?

Yes- thank-you. Now corrected

Extended Data Figure 4: What does the GSW in Fig. 4A stand for? This was not defined anywhere. A positive value indicates deficit? A negative value indicates surplus? The use of the term "elevation deficit" does not seem to match the signs of the numbers. The title of the Y-axis has been changed to “Normalised Surface Water Change”. We have reversed the shading representing elevation deficit, so that the deeper blue shades correspond to a higher elevation deficit (the deficit between RSLR and elevation gain, (so yes, positive is a deficit, leading to greater – more positive- surface water).

Extended Data Table 3: Do you mean predictors? Any difference between identifiers and variables? Identifiers were never mentioned in main text and the Methods section. So, they came out of nowhere.

We have included now in the Methods section the sentence:

Predictive variables are provided in Extended Data Table 3, and consist of climatic, hydrological and edaphic properties associated with each SET-MH station, and are sourced from ref⁴⁹.